# UTILITY: Utilizing Explainable Reinforcement Learning to Improve Reinforcement Learning

**Shicheng Liu & Minghui Zhu**
Department of Electrical Engineering
Pennsylvania State University
University Park, PA 16802, USA
{sfl5539,muz16}@psu.edu

## Abstract

Reinforcement learning (RL) faces two challenges: (1) The RL agent lacks explainability. (2) The trained RL agent is, in many cases, non-optimal and even far from optimal. To address the first challenge, explainable reinforcement learning (XRL) is proposed to explain the decision-making of the RL agent. In this paper, we demonstrate that XRL can also be used to address the second challenge, i.e., improve RL performance. Our method has two parts. The first part provides a two-level explanation for why the RL agent is not optimal by identifying the mistakes made by the RL agent. Since this explanation includes the mistakes of the RL agent, it has the potential to help correct the mistakes and thus improve RL performance. The second part formulates a constrained bi-level optimization problem to learn how to best utilize the two-level explanation to improve RL performance. In specific, the upper level learns how to use the high-level explanation to shape the reward so that the corresponding policy can maximize the cumulative ground truth reward, and the lower level learns the corresponding policy by solving a constrained RL problem formulated using the low-level explanation. We propose a novel algorithm to solve this constrained bi-level optimization problem, and theoretically guarantee that the algorithm attains global optimality. We use MuJoCo experiments to show that our method outperforms state-of-the-art baselines.

## 1 Introduction

While reinforcement learning (RL) has been implemented in a wide range of applications, it faces two significant challenges: (1) The RL agent lacks transparency due to its black-box nature. (2) It has been widely observed in the RL community (Haarnoja et al., 2018; Henderson et al., 2018; Dulac-Arnold et al., 2019; Cheng et al., 2024) that, in many cases, the trained RL agent does not achieve maximum cumulative reward (i.e., non-optimal and even far from optimal). These two challenges motivate the need to improve the transparency and performance of the RL agent.

To address the first challenge, explainable reinforcement learning (XRL) methods are proposed to explain the decision-making of the RL agents, including learning an interpretable policy (Bastani et al., 2018; Bewley & Lawry, 2021), pinpointing regions in the observations that are critical for choosing certain actions (Atrey et al., 2019; Puri et al., 2019), learning the reward function that is actually maximized (Xie et al., 2022), and identifying the critical states influential to the cumulative reward (Guo et al., 2021; Cheng et al., 2023). These XRL methods generate various explanations that improve the transparency of the RL agent and help people build trust in the RL agent.

This paper demonstrates that XRL can also be used to address the second challenge of RL, i.e., RL improvement. Given a non-optimal RL agent, we use XRL to explain why this RL agent is not optimal by finding the mistakes made by the RL agent. Since our explanation provides insights into the RL agent's mistakes, it has the potential to help correct the mistakes and thus improve performance. Some recent works (Guo et al., 2021; Cheng et al., 2023; 2024) also use XRL to improve the RL performance. In specific, they propose to first identify the critical states that are

most influential to the cumulative reward as an explanation, and then perturb the actions (Guo et al., 2021) or fine-tune the policy (Cheng et al., 2023; 2024) at those critical states. However, they do not explain why the RL agent does not maximize the cumulative reward. This paper proposes a novel framework that first explains why the RL agent is not optimal, and then learns how to utilize the generated explanations to improve RL performance. We summarize our contributions as follows:

**Contribution statement**. This paper proposes an optimization-based framework that aims to learn how to best utilize XRL to improve RL. We refer to this framework as "**ut**ilizing expla**i**nable R**L** to **i**mprove reinforcemen**t** learning efficac**y**" (UTILITY). Our contributions are threefold:

First, we provide a two-level explanation for why the RL agent is not optimal. The high-level explanation learns a reward function to which the RL agent is actually optimal, and then explains why the RL agent is not optimal by comparing this learned reward function to the ground truth reward function. The low-level explanation identifies the state-action pairs that lead the RL agent to be non-optimal. We refer to these state-action pairs as "misleading" state-action pairs, and rigorously derive a mathematical metric to identify the "misleading" state-action pairs as the low-level explanation.

Second, we formalize the problem of utilizing the two-level explanations to improve the performance as a constrained bi-level optimization problem. The upper-level problem aims to learn how to use the high-level explanation (i.e., the learned reward function) to shape the ground truth reward function to help the corresponding policy maximize the cumulative (ground truth) reward. The lower-level problem learns the corresponding policy by solving a constrained RL problem where the objective is to maximize the cumulative shaping reward and the constraint is to discourage from visiting the low-level explanation (i.e., the "misleading" state-action pairs). A key insight is that the formulated optimization problem facilitates policy improvement. Current constrained bi-level optimization can only deal with the case where the lower-level problem is strongly convex. However, in our case, both the objective function and constraint in the lower-level problem are highly non-convex. Therefore, a novel theoretical framework is desired to solve this constrained bi-level optimization problem.

Third, we develop a novel theoretical framework and thereby an algorithm to solve the constrained bi-level optimization problem. In specific, we first use a dual method to transform the constrained bi-level optimization problem to an equivalent unconstrained bi-level optimization problem, and then propose an approximation-based triple-loop algorithm to solve this unconstrained bi-level optimization problem. We quantify the approximation error at each loop and prove that the algorithm attains global optimality. Experiments show that UTILITY outperforms state-of-the-art baselines.

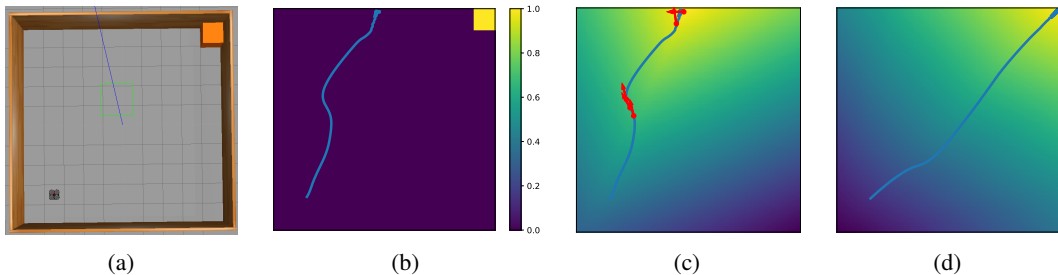

| (a) | (b) | (c) | (d) |

Figure 1: (a) An RL task where a drone starts from the lower-left corner and navigates to the orange goal at the upper-right corner. (b) A failing trajectory (blue) generated by the RL agent and a heat map that visualizes the ground truth reward. (c) The two-level explanation of why the blue trajectory fails to reach the goal. The high-level explanation is the learned reward (visualized as the heat map) to which the RL agent is actually optimal. The low-level explanation is the misleading state-action pairs (red circles and linked red arrows). (d) The trajectory after improvement reaches the goal.

**Illustrative example**. Figure 1 uses an example to illustrate our proposed framework. Suppose we use RL to navigate a drone to the orange goal in Figure 1a. The state is the 2-D coordinate and the action is the moving direction. The ground truth reward is one at the goal states and zero otherwise. Figure 1b uses a heat map to visualize the ground truth reward and the blue trajectory is generated by the learned policy. This learned policy is not optimal because it fails to reach the goal.

Figure 1c visualizes our two-level explanation of why the learned policy is not optimal. At the high level, we use the heat map to visualize the learned reward function to which the RL agent's

trajectory (policy) is actually optimal. The high-level explanation is that the RL agent's policy is actually optimal to the learned reward function (visualized as the heat map in Figure 1c), and this learned reward function is very different from the ground truth reward function (visualized as the heat map in Figure 1b). Note that we normalize all the reward functions in Figures 1b-1d to $[0, 1]$ for better comparison. For the low-level explanation, we identify the top five "misleading" state-action pairs (i.e., the red circles and arrows) in the blue trajectory where the red circles are the states and the linked red arrows are the corresponding actions chosen by the non-optimal RL agent. These state-action pairs are "misleading" since the correct actions should point to the goal.

Figure 1d shows the improvement where the heat map visualizes the learned shaping reward function and the blue trajectory is generated by the learned policy after improvement. We can see that the learned policy after improvement successfully reaches the goal.

## 2 RELATED WORKS

Due to the space limit, we only include the related works on improving RL performance here, and we include more related works in Appendix E.

**Reward shaping**. Reward shaping can improve the RL performance by shaping the ground truth reward function. Current works on reward shaping has two main categories. The first category (Ng et al., 1999; Hu et al., 2020; Devlin & Kudenko, 2012; Gupta et al., 2022) requires an external source, such as a human expert, to provide domain knowledge as an ingredient to shape the ground truth reward function. However, when the tasks become complicated, it could be difficult and even infeasible for humans to provide domain knowledge. The second category does not need domain knowledge, including reward shaping based on exploration bonus (Bellemare et al., 2016; Ostrovski et al., 2017), learning an intrinsic reward (Zheng et al., 2018; Memarian et al., 2021), and combining exploration bonus and intrinsic reward (Devidze et al., 2022). The first category usually has better performance, while the second category does not require human-domain knowledge. Our method enjoys the benefits of both categories because we use domain knowledge but the domain knowledge is not given by human expert but learned by XRL.

**Other methods that can improve RL performance**. Lazy-MDP (Jacq et al., 2022) shows performance improvement with the help of a provided default policy. It uses the "lazy-gap" to determine whether to choose greedy action or follow a default policy on each state $s$. Self-imitation learning (Oh et al., 2018) aims to encourage deep exploration by reproducing previous good decisions. Papers (Wang & Taylor, 2017; Taylor, 2018; Taylor et al., 2023) aim to improve the RL performance by utilizing external assistance, such as the assistance of a pre-trained RL agent (Wang & Taylor, 2017) or a human (Taylor, 2018; Taylor et al., 2023), which may not be accessible in some scenarios.

## 3 TWO-LEVEL EXPLANATION OF WHY THE RL AGENT IS NON-OPTIMAL

This section provides a two-level explanation to explain why the RL agent is not optimal. The RL agent's decision making is based on a Markov decision process (MDP) $(\mathcal{S}, \mathcal{A}, \gamma, P_0, P, r)$ which consists of a state set $\mathcal{S}$, an action set $\mathcal{A}$, a discount factor $\gamma \in (0, 1)$, an initial state distribution $P_0(\cdot)$, a state transition function $P(\cdot|\cdot, \cdot)$, and the ground truth reward function $r(\cdot, \cdot)$. The RL agent's learned policy is denoted by $\pi_A$ and the cumulative reward is defined as $J_r(\pi) \triangleq E^\pi[\sum_{t=0}^\infty \gamma^t r(s_t, a_t)]$ where the initial state is drawn from $P_0$. When we say that the RL agent is not optimal, it means that $\pi_A \notin \arg\max_\pi J_r(\pi)$.

**The black-box assumption**. To ensure practicability, following (Bewley & Lawry, 2021; Guo et al., 2021; Cheng et al., 2023; Guidotti et al., 2019), we only treat the RL agent as a black box with no access to its internal structure. In specific, we do not assume the access to the learned value/Q-function nor the learned policy $\pi_A$ of the RL agent. We can only observe a set of $m$ trajectories $\mathcal{D} \triangleq \{\zeta^j\}_{j=1}^m$ demonstrated by the RL agent (using the non-optimal policy $\pi_A$) where each trajectory $\zeta^j = s_0, a_0, \cdots$ is a state-action sequence.

**The high-level explanation**. At a high level, since the RL agent is not optimal to the ground truth reward function $r$, we can learn a reward function $\hat{r}$ to which the RL agent's policy $\pi_A$ is actually optimal, and use this learned reward function $\hat{r}$ to generate explanations. A recent work (Xie et al.,

2022) uses the state-action pairs $(s, a)$ with the highest $\hat{r}(s, a)$ as an explanation, however, these state-action pairs cannot explain why the RL agent is not optimal. Therefore, we extend (Xie et al., 2022) by comparing the learned reward function $\hat{r}$ to the ground truth reward function $r$ to explain why $\pi_A$ is not optimal to the ground truth $r$. Figures 1b-1c provide an example of our high-level explanation: the policy $\pi_A$ is actually optimal to the learned reward function $\hat{r}$ (visualized in Figure 1c), and this learned reward function $\hat{r}$ is very different from the ground truth reward function $r$ (visualized in Figure 1b).

Inverse reinforcement learning (IRL) (Abbeel & Ng, 2004; Ziebart et al., 2008; Arora & Doshi, 2021) can learn the reward function $\hat{r}$ and an associated policy $\hat{\pi}_A$ from the demonstration set $\mathcal{D}$ such that the behaviors of policy $\pi_A$ demonstrated in $\mathcal{D}$ are optimal to the reward function $\hat{r}$ learned by IRL, and the learned policy $\hat{\pi}_A$ can imitate the policy $\pi_A$. We use maximum likelihood IRL (Zeng et al., 2022; 2023) to learn the reward function $\hat{r}$ and policy $\hat{\pi}_A$.

While the learned reward function $\hat{r}$ can be used for the high-level explanation, it is only interpretable to humans in low dimension, e.g., we can use heat maps to plot reward functions (as in Figure 1). When the state and action become high dimensional, the learned reward function $\hat{r}$ is hard for humans to understand and thus it is difficult to straightforwardly compare $\hat{r}$ to $r$ (as we did in Figures 1b-1c). Therefore, we need the low-level explanation which is still interpretable in high dimension.

**The low-level explanation**. At a low level, the RL agent is not optimal meaning that it visits some critical points that lead to the non-optimality. Recent works (Guo et al., 2021; Cheng et al., 2023; Amir & Amir, 2018; Jacq et al., 2022) identify the states that are most influential to the cumulative reward as critical points. In order to explain why the RL agent is not optimal, we extend their idea by redefining the critical points as the state-action pairs that lead $\pi_A$ to be non-optimal. We refer to these critical points as "misleading" state-action pairs and we aim to identify the top $K$ "misleading" state-action pairs in the demonstration set $\mathcal{D}$ as the low-level explanation. Note that we use infinite time-horizon MDP and it is not possible to identify the top $K$ "misleading" state-pairs if a trajectory has infinitely many different state-action pairs. However, in practice, the trajectory length is usually finite and we can unify the notions of finite time horizon and infinite time horizon by introducing "absorbing state" (Sutton & Barto, 2018). In specific, we can treat the terminal state of a finite-time-horizon trajectory as a state keeping transitioning only to itself and generating zero reward.

The *key challenge* to identify the top $K$ "misleading" state-action pairs in the demonstration set $\mathcal{D}$ is to propose a proper criterion or metric to define what a "misleading" state-action pair is. A straightforward way is to identify the state-action pairs that an optimal policy will not visit as "misleading", and thus use $\pi_A(a|s) - \pi^*(a|s)$ as the metric where $\pi^*$ is an optimal policy. However, this metric is infeasible because the optimal policy $\pi^*$ is not accessible.

In contrast, we derive a feasible metric in Definition 1 that uses a $Q$-function to find misleading state-action pairs in the demonstration set $\mathcal{D}$. The $Q$-function under the policy $\pi$ and reward function $r$ is $Q_r^\pi(s, a) \triangleq E^\pi[\sum_{t=0}^\infty \gamma^t r(s_t, a_t)|s_0 = s, a_0 = a]$.

**Definition 1.** *A state-action pair $(s, a) \in \mathcal{D}$ is a misleading state-action pair if $l(s, a) > 0$ where $l(s, a) \triangleq \max_{a'} Q_r^{\pi_A}(s, a') - Q_r^{\pi_A}(s, a)$ is referred to as "misleading level". The larger the misleading level $l(s, a)$ is, the more misleading the state-action pair $(s, a)$ is.*

We include the derivation of how we come up with this metric $l$ and the proof of why $(s, a) \in \mathcal{D}$ is misleading if $l(s, a) > 0$ in Appendix B.1. In brief, we prove in Appendix B.1 that the policy $\pi_A$ will be an optimal policy if $l(s, a) = 0$ for all $(s, a)$ such that $a \in \pi_A(s)$ where $\pi_A(s)$ is the set of actions that the policy $\pi_A$ has nonzero probability to choose at the state $s$. Therefore, any state-action pair $(s, a) \in \mathcal{D}$ such that $l(s, a) > 0$ can be regarded as a "misleading" state-action pair that leads the policy $\pi_A$ to be non-optimal. The larger the misleading level $l(s, a)$ is, the more "misleading" the state-action pair $(s, a)$ is, because the $Q$ value of the chosen action $a$ has a larger gap from the maximum $Q$ value at the state $s$. We denote the set of the identified top $K$ "misleading" state-action pairs by $\mathcal{C}$, which serves as the low-level explanation.

While we cannot access the policy $\pi_A$, we have already learned the policy $\hat{\pi}_A$ using IRL and the policy $\hat{\pi}_A$ imitates the policy $\pi_A$. Therefore, we can use $Q_r^{\hat{\pi}_A}$ to substitute for $Q_r^{\pi_A}$. Given that we can access $\hat{\pi}_A$ and $r$, we can simply learn $Q_r^{\hat{\pi}_A}$ by sampling the environment to collect enough data and doing regression. Due to the space limit, we include the method of learning $Q_r^{\hat{\pi}_A}$ in Appendix B.2. We are aware that in RL, it is usually sample inefficient and computationally expensive if we

want to sample enough data to learn precise $Q$-functions corresponding to all the learning policy in the learning procedure. However, our case is different because we only need to learn one precise $Q$-function, which corresponds to the specific policy $\hat{\pi}_A$.

## 4 UTILIZING THE TWO-LEVEL EXPLANATION TO IMPROVE RL

This section provides a theoretical framework that utilizes the two-level explanation in Section 3 to improve the RL performance. In specific, Subsection 4.1 formulates the problem as a constrained bi-level optimization problem. Subsection 4.2 proposes a novel theoretical framework and thereby an algorithm to solve the constrained bi-level optimization problem.

### 4.1 PROBLEM FORMULATION

We aim to utilize the two-level explanation to improve the RL performance. For the high-level explanation $\hat{r}$, we use it to formulate a domain knowledge and learn how to use this domain knowledge to shape the ground truth reward $r$ such that the learned shaping reward can lead the policy to maximize the cumulative ground truth reward $J_r(\pi)$. In specific, we use the comparison $r - \hat{r}$ between the ground truth reward $r$ and the high-level explanation $\hat{r}$ as the domain knowledge. Note that in practice, we need to first scale $\hat{r}$ to the same scale with $r$ for a better comparison. Since this comparison quantifies the RL agent's misunderstanding of the ground truth reward function $r$, it has the potential to help patch the error. Towards this end, we propose to learn a shaping function $\theta(\cdot, \cdot)$. For a given state-action pair $(s, a)$, the corresponding shaping reward is $r_\theta(s, a) = r(s, a) + \theta(s, a)(r(s, a) - \hat{r}(s, a))$.

For the low-level explanation $\mathcal{C}$, we discourage the RL agent from visiting the "misleading" state-action pairs in $\mathcal{C}$. Towards this end, we design a cost function $c(\cdot, \cdot)$ such that $c(s, a) \in (0, c_{\max}]$ when $(s, a) \in \mathcal{C}$, and $c(s, a) = 0$ otherwise, where $c_{\max}$ is a positive constant. We discourage from visiting $\mathcal{C}$ by constraining the cumulative cost $J_c(\pi) \triangleq E^\pi[\sum_{t=0}^\infty \gamma^t c(s_t, a_t)]$ under a budget $b$.

To utilize the two-level explanation, we formulate a constrained bi-level optimization problem:

$$\max_\theta \ J_r(\pi_{r_\theta}), \text{ where } \pi_{r_\theta} = \arg\max_\pi\{J_{r_\theta}(\pi) + H(\pi), \text{ s.t. } J_c(\pi) \le b\}, \tag{1}$$

where $J_{r_\theta}(\pi) \triangleq E^\pi[\sum_{t=0}^\infty \gamma^t r_\theta(r(s_t, a_t), r(s_t, a_t) - \hat{r}(s_t, a_t))]$ is the cumulative shaping reward, and the causal entropy $H(\pi) \triangleq E^\pi[\sum_{t=0}^\infty -\gamma^t \log \pi(a_t|s_t)]$ is to encourage exploration and is widely used in soft Q-learning (Haarnoja et al., 2017) and soft actor-critic (Haarnoja et al., 2018).

In the problem (1), the upper level aims to learn a shaping reward function $r_\theta$ such that the corresponding policy $\pi_{r_\theta}$ can achieve maximum cumulative ground truth reward $J_r(\pi_{r_\theta})$. Given the current learned shaping reward function $r_\theta$, the lower-level problem in (1) is to compute the corresponding policy $\pi_{r_\theta}$ by solving a constrained RL problem. The constrained RL problem encourages $\pi_{r_\theta}$ to maximize the entropy-regularized cumulative shaping reward $(J_{r_\theta}(\pi) + H(\pi))$ and discourages $\pi_{r_\theta}$ from visiting $\mathcal{C}$ by controlling the cumulative cost $J_c(\pi)$ under the budget $b$. Note that if we choose $b = 0$, it means that the policy $\pi_{r_\theta}$ should totally avoid the set $\mathcal{C}$.

**Remark on why the problem (1) can improve** $\pi_A$. Recall that $\pi_A$ is an non-optimal policy obtained by running an RL algorithm until convergence, i.e., $\pi_A$ stops improving. The lower level in (1) improves $\pi_A$ by facilitating a policy improvement step. In specific, according to monotonic policy improvement theorem, $\pi_A$ will improve if it chooses greedy actions according to its Q-function $Q_r^{\pi_A}$ at all the states. Given that $\pi_A$ stops improving, it means that $\pi_A$ must choose some nongreedy actions at some states, i.e., the misleading state-action pairs. Constraining these misleading state-action pairs means that we constrain the nongreedy actions $\pi_A$ originally chooses at the states. This constraint can help $\pi_A$ choose greedy actions because it eliminates some nongreedy actions and thus $\pi_A$ only needs to find greedy actions from smaller action sets. Since this constraint can help $\pi_A$ find greedy actions, it can help improve $\pi_A$. The upper level calibrates the reward function to improve performance. In specific, we can treat the RL agent as a biased black box such that if the input is the ground truth reward function $r$, the output is a policy $\pi_A$ that is optimal to another reward function $\hat{r}$. We want to change the input of this biased black box such that the output is a policy that is optimal to the ground truth reward function $r$. More formally, we aim to learn a shaping reward $r_\theta$ as the input such that the output is optimal to $r$. A straightforward way is to directly learn a neural reward

$r_\theta : \mathcal{S} \times \mathcal{A} \to R$. However, in practice, this is difficult to learn because the neural reward needs to search over a very large function space. We find empirically that if we use the shaping reward in the form of $r_\theta(s, a) = r(s, a) + \theta(s, a)(r(s, a) - \hat{r}(s, a))$, the learning results will be much better. As the problem (1) facilitates policy improvement, we can further improve the final learned policy of problem (1) by generating its two-level explanation and use the explanation to formulate problem (1) again. We can iteratively formulate and solve problem (1) to keep improving the learned policy. In this paper, we only discuss the scenario where we formulate and solve the problem (1) once.

Before solving the problem (1), we need to first make sure that the problem (1) is well-defined. In specific, since the lower-level problem in (1) is non-convex, it may have more than one optimal solution, i.e., $\pi_{r_\theta}$ is not unique. Therefore, given a reward parameter $\theta$, the corresponding upper-level objective function value $J_r(\pi_{r_\theta})$ may not be unique as $\pi_{r_\theta}$ is not unique. This will make the problem (1) ill-defined. The following theorem guarantees that the problem (1) is well-defined.

**Theorem 1.** *Given reward $r_\theta$, the optimal solution $\pi_{r_\theta}$ of the lower-level problem in (1) is unique.*

## 4.2 Theoretical framework

While the current state-of-the-arts (Xu & Zhu, 2023a; Khanduri et al., 2023) on constrained bi-level optimization can only deal with strongly convex lower-level problems, both the objective function and the constraint of the lower-level problem in (1) are non-convex. Therefore, a novel theoretical framework is desired to solve the problem (1). This subsection proposes a novel theoretical framework to solve the problem (1).

The proposed theoretical framework has three parts. (i) The first part transforms the original constrained bi-level optimization problem (1) to an equivalent unconstrained bi-level optimization problem. The benefit of this transformation is that the equivalent unconstrained bi-level optimization problem has an unconstrained and convex lower-level problem, which is more tractable and easier to solve. (ii) The second part proposes a novel algorithm to solve the problem (1) by solving the equivalent unconstrained bi-level optimization problem. (iii) The third part theoretically guarantees that the proposed algorithm attains global optimality.

### 4.2.1 Problem transformation

The lower-level problem of the problem (1) is non-convex. To deal with the non-convexity issue, we introduce the dual function of the lower-level problem in (1): $G(\lambda; \theta) \triangleq \max_\pi J_{r_\theta}(\pi) + H(\pi) - \lambda(J_c(\pi) - b)$ where $\lambda$ is the dual variable. The dual function $G(\lambda; \theta)$ is convex in $\lambda$ since it is the point-wise maximum over a set of affine functions of $\lambda$ (Boyd & Vandenberghe, 2004).

**Theorem 2.** *The optimal solution of the lower-level problem in (1) is uniquely the constrained soft policy $\pi_{\lambda^*(\theta);\theta}$ where $\lambda^*(\theta)$ is the unique optimal solution of the dual problem $\min_\lambda G(\lambda; \theta)$.*

We include the analytical expression of the constrained soft policy $\pi_{\lambda^*(\theta);\theta}$ (Liu & Zhu, 2022) in Appendix C. Theorem 2 indicates that $\pi_{\lambda^*(\theta);\theta}$ is the unique optimal solution of the lower-level problem in (1) (i.e., $\pi_{\lambda^*(\theta);\theta} = \pi_{r_\theta}$), and $\lambda^*(\theta) = \arg\min_\lambda G(\lambda; \theta)$. Therefore, we can replace $\pi_{r_\theta}$ with $\pi_{\lambda^*(\theta);\theta}$ and replace the lower-level problem in (1) with its dual problem, and thereby transform the constrained bi-level optimization problem (1) to the following unconstrained bi-level optimization problem:

$$\max_\theta \ J_r(\pi_{\lambda^*(\theta);\theta}), \text{ where } \lambda^*(\theta) = \arg\min_\lambda G(\lambda; \theta). \tag{2}$$

Compared to the original problem (1), the lower-level problem of the problem (2) is unconstrained and convex. However, there are still two challenges to solve the new problem (2).

**Challenge (i)**: Evaluating the dual function $G(\lambda; \theta)$ needs to obtain the constrained soft policy $\pi_{\lambda;\theta} = \arg\max_\pi J_{r_\theta}(\pi) + H(\pi) - \lambda(J_c(\pi) - b)$. However, current RL algorithms can only approach $\pi_{\lambda;\theta}$ at a certain rate and only obtain the exact $\pi_{\lambda;\theta}$ when iteration number goes to infinity. In practice, we can only run an algorithm for finite iterations and thus we cannot obtain the exact $\pi_{\lambda;\theta}$. This will cause errors when we evaluate the dual function $G$.

**Challenge (ii)**: Even if we can obtain the exact $\pi_{\lambda;\theta}$, we cannot guarantee to get the exact optimal solution $\lambda^*(\theta)$ of the lower-level problem in finite time. This makes it difficult to evaluate and solve the upper-level problem in (2) since the upper-level problem in (2) requires $\lambda^*(\theta)$.

---

**Algorithm 1** Utilizing explainable reinforcement learning to improve reinforcement learning

---

**Input**: Demonstration set $\mathcal{D}$, initial shaping reward parameter $\theta_0$, dual parameter $\lambda_0$, and policy $\pi_0$
**Output**: Shaping reward $r_{\theta_N}$ and the policy after improvement $\hat{\pi}_{\hat{\lambda}(\theta_N);\theta_N}$

1: Generate the two-level explanation $(\hat{r}, \mathcal{C})$
2: **for** $n = 0, \cdots, N-1$ **do**
3:    **for** $\bar{n} = 0, \cdots, \bar{N}-1$ **do**
4:       **for** $\tilde{n} = 0, \cdots, \tilde{N}_{\bar{n}}-1$ **do**
5:          Compute the constrained soft Q function $Q^{\pi_{\tilde{n}}}_{\lambda_{\bar{n}};\theta_n}$
6:          Update the policy $\pi_{\tilde{n}+1}(a|s) \propto \exp(Q^{\pi_{\tilde{n}}}_{\lambda_{\bar{n}};\theta_n}(s,a))$ for any $(s,a) \in \mathcal{S} \times \mathcal{A}$
7:       **end for**
8:       Set $\hat{\pi}_{\lambda_{\bar{n}};\theta_n} = \pi_{\tilde{N}_{\bar{n}}}$ and use $\hat{\pi}_{\lambda_{\bar{n}};\theta_n}$ to compute the approximated gradient $g_{\lambda_n;\theta_n}$
9:       Update $\lambda_{\bar{n}+1} = \lambda_{\bar{n}} - \alpha_{\bar{n}} g_{\lambda_{\bar{n}};\theta_n}$
10:    **end for**
11:    Set $\hat{\lambda}(\theta_n) = \frac{1}{\bar{N}} \sum_{\bar{n}=0}^{\bar{N}-1} \lambda_{\bar{n}}$ and compute $\hat{\pi}_{\hat{\lambda}(\theta_n);\theta_n}$ via $(\bar{N}-1)$-step soft policy iteration
12:    Use $\hat{\pi}_{\hat{\lambda}(\theta_n);\theta_n}$ to compute the approximated gradient $g_{\theta_n}$ and update $\theta_{n+1} = \theta_n + \beta_n g_{\theta_n}$
13: **end for**

---

### 4.2.2 THE PROPOSED ALGORITHM

This part proposes a novel algorithm that solves problem (1) by solving problem (2). The proposed algorithm is triple-loop where the inner loop approximates the constrained soft policy $\pi_{\lambda;\theta}$ and tackles Challenge (i), the middle loop approximates the optimal solution $\lambda^*(\theta)$ of the lower-level problem in (2) and tackles Challenge (ii), and the outer loop solves the upper-level problem in (2). We use $n$, $\bar{n}$, and $\tilde{n}$ to respectively denote the iteration indices of outer, middle, and inner loop.

Algorithm 1 first generates the two-level explanation (line 1) and then uses three loops to utilize the generated two-level explanation. In specific, the inner loop (lines 4-7) approximates the constrained soft policy $\pi_{\lambda;\theta}$. With the approximated policy $\hat{\pi}_{\lambda;\theta}$ (line 8), the middle loop solves the lower-level problem in (2) via $(\bar{N}-1)$-step gradient descent (line 9) to approximate the optimal solution $\lambda^*(\theta)$. With the approximated parameter $\hat{\lambda}(\theta)$ (line 11), the outer loop solves the upper-level problem in (2) via $(N-1)$-step gradient ascent (line 12). In the following, we elaborate each loop respectively.

**The inner loop**. Given the parameter $(\lambda, \theta)$, the inner loop aims to approximate the constrained soft policy $\pi_{\lambda;\theta}$ via $\tilde{N}_{\bar{n}}$-step soft policy iteration (Haarnoja et al., 2017), and $\tilde{N}_{\bar{n}} = \bar{n} + 1$. Soft policy iteration has two steps: policy evaluation and policy improvement. Policy evaluation computes the constrained soft Q-function $Q^{\pi_{\tilde{n}}}_{\lambda;\theta}$ corresponding to the current policy $\pi_{\tilde{n}}$, dual parameter $\lambda$, and reward parameter $\theta$. We include the expression of the constrained soft Q-function in Appendix C. Policy improvement aims to update the policy according to $\pi_{\tilde{n}+1}(a|s) \propto \exp(Q^{\pi_{\tilde{n}}}_{\lambda;\theta}(s,a))$ for any $(s,a) \in \mathcal{S} \times \mathcal{A}$. The output of the inner loop is the approximated policy $\hat{\pi}_{\lambda;\theta} = \pi_{\tilde{N}_{\bar{n}}}$. In practical implementations, we can update the policy $\pi_{\tilde{n}}$ via the policy update in soft Q-learning (Haarnoja et al., 2017) or actor update in soft actor-critic (Haarnoja et al., 2018). While soft Q-learning and soft actor-critic are designed for unconstrained RL, we show in Appendix C that we can revise them to approximate the constrained soft policy.

**The middle loop**. We aim to solve the lower-level problem in (2) via $(\bar{N}\text{-}1)$-step gradient descent.

**Lemma 1.** *The gradient of the dual function $G$ is $\nabla_\lambda G(\lambda; \theta) = b - J_c(\pi_{\lambda;\theta})$.*

The gradient $\nabla_\lambda G(\lambda; \theta)$ requires the exact constrained soft policy $\pi_{\lambda;\theta}$ which is inaccessible. Therefore, we use the approximated policy $\hat{\pi}_{\lambda;\theta}$ obtained from the inner loop to approximate the gradient $\nabla_\lambda G(\lambda; \theta)$ via the gradient approximation $g_{\lambda;\theta} = b - J_c(\hat{\pi}_{\lambda;\theta})$, and solve the lower-level problem via $(\bar{N}\text{-}1)$-step gradient descent $\lambda_{\bar{n}+1} = \lambda_{\bar{n}} - \alpha_{\bar{n}} g_{\lambda_{\bar{n}};\theta}$. The output is $\hat{\lambda}(\theta) = \frac{1}{\bar{N}} \sum_{\bar{n}=0}^{\bar{N}-1} \lambda_{\bar{n}}$.

**The outer loop**. We solve the upper-level problem in (2) via $(N-1)$-step gradient ascent. Towards this end, we generalize the $Q$/value function (Sutton & Barto, 2018). In specific, we define the $Q$-function of cost $c$ under policy $\pi$ as $Q^\pi_c(s,a) \triangleq E^\pi[\sum_{t=0}^\infty \gamma^t c(s_t, a_t)|s_0 = s, a_0 = a]$ and value function of cost as $V^\pi_c(s) \triangleq E^\pi[\sum_{t=0}^\infty \gamma^t c(s_t, a_t)|s_0 = s]$. We define the $Q$-function of reward gradient $\nabla_\theta r_\theta$ as $Q^\pi_{\nabla_\theta r_\theta}(s,a) \triangleq E^\pi[\sum_{t=0}^\infty \gamma^t \nabla_\theta r_\theta(s_t, a_t)|s_0 = s, a_0 = a]$ and value function of

---

reward gradient as $V^{\pi}_{\nabla_\theta r_\theta}(s) \triangleq E^{\pi}[\sum_{t=0}^{\infty} \gamma^t \nabla_\theta r_\theta(s_t, a_t)|s_0 = s]$. We define state-action visitation frequency as $\psi^{\pi}(s, a) \triangleq E^{\pi}[\sum_{t=0}^{\pi} \gamma^t \mathbb{1}\{s_t = s, a_t = a\}]$ where $\mathbb{1}\{\cdot\}$ is the indicator function.

**Lemma 2.** *The upper-level gradient is* $dJ_r(\pi_{\lambda^*(\theta);\theta})/d\theta = E_{(s,a)\sim\psi^{\pi_{\lambda^*(\theta);\theta}}}\left[\left(Q^{\pi_{\lambda^*(\theta);\theta}}_{\nabla_\theta r_\theta}(s, a) - V^{\pi_{\lambda^*(\theta);\theta}}_{\nabla_\theta r_\theta}(s) - C_{\pi_{\lambda^*(\theta);\theta}}(Q^{\pi_{\lambda^*(\theta);\theta}}_c(s, a) - V^{\pi_{\lambda^*(\theta);\theta}}_c(s))\right)Q^{\pi_{\lambda^*(\theta);\theta}}_r(s, a)\right]$ *where* $C_\pi$ *is a constant vector if we fix policy* $\pi$*, and we include the expression of* $C_\pi$ *in Appendix D.3.*

Since the gradient $\frac{dJ_r(\pi_{\lambda^*(\theta);\theta})}{d\theta}$ requires the exact optimal solution $\lambda^*(\theta)$ and the exact constrained soft policy $\pi_{\lambda^*(\theta);\theta}$, we can only use the policy $\hat{\pi}_{\hat{\lambda}(\theta);\theta}$ to approximate $\frac{dJ_r(\pi_{\lambda^*(\theta);\theta})}{d\theta}$ via $g_\theta = E_{(s,a)\sim\psi^{\hat{\pi}_{\hat{\lambda}(\theta);\theta}}}\left[\left(Q^{\hat{\pi}_{\hat{\lambda}(\theta);\theta}}_{\nabla_\theta r_\theta}(s, a) - V^{\hat{\pi}_{\hat{\lambda}(\theta);\theta}}_{\nabla_\theta r_\theta}(s) - C_{\hat{\pi}_{\hat{\lambda}(\theta);\theta}}(Q^{\hat{\pi}_{\hat{\lambda}(\theta);\theta}}_c(s, a) - V^{\hat{\pi}_{\hat{\lambda}(\theta);\theta}}_c(s))\right)Q^{\hat{\pi}_{\hat{\lambda}(\theta);\theta}}_r(s, a)\right]$.
We then solve the upper-level problem in (2) via $(N - 1)$-step gradient ascent $\theta_{n+1} = \theta_n + \beta_n g_{\theta_n}$.

### 4.2.3 Theoretical Analysis

This part quantifies the optimality of the policy after improvement $\hat{\pi}_{\hat{\lambda}(\theta_N);\theta_N}$. The main difficulty is that the inner loop and middle loop can only approximate the policy $\pi_{\lambda;\theta}$ and the optimal solution $\lambda^*(\theta)$, and the approximation error may accumulate and ruin the convergence of the outer loop. In the following context, we sequentially quantify the convergence from the inner loop to the outer loop.

**Lemma 3** (convergence of the inner loop). *Given the parameter* $(\lambda, \theta)$*, the output* $\hat{\pi}_{\lambda;\theta}$ *of the inner loop satisfies* $|\log \hat{\pi}_{\lambda;\theta}(a|s) - \log \pi_{\lambda;\theta}(a|s)| \leq O(\gamma^{\bar{N}_{\bar{n}}})$ *for any* $(s, a) \in \mathcal{S} \times \mathcal{A}$.

Lemma 3 shows that inner loop converges linearly to the exact constrained soft policy $\pi_{\lambda;\theta}$.

**Assumption 1.** *(i) It holds that* $|r_\theta(\cdot, \cdot)| \leq C_1$ *for any* $\theta$ *where* $C_1$ *is a positive constant. (ii) It holds that* $||\nabla_\theta r_\theta(\cdot, \cdot)|| \leq C_2$ *and* $||\nabla^2_{\theta\theta} r_\theta(\cdot, \cdot)|| \leq C_3$*, where* $C_2$ *and* $C_3$ *are some positive constants.*

Assumption 1 assumes that $r_\theta$ is bounded, Lipschitz continuous, and smooth to $\theta$, which is a common assumption in RL (Zheng et al., 2024b; Lan et al., 2024a; Zheng et al., 2024a; Zhang et al., 2025; Zheng et al., 2024c; Lan et al., 2024b). We next quantify the convergence of the middle loop:

**Lemma 4** (convergence of the middle loop). *Suppose Assumption 1 (ii) holds and let* $\alpha_{\bar{n}} = 1/(\bar{n} + 1)^{\bar{\eta}}$ *where* $\bar{\eta} \in (1/2, 1)$*, the outputs* $(\hat{\lambda}(\theta), \hat{\pi}_{\hat{\lambda}(\theta);\theta})$ *of the middle loop satisfy that (i)* $|\hat{\lambda}(\theta) - \lambda^*(\theta)| \leq O(1/\bar{N}^{1-\bar{\eta}})$*; (ii)* $|\log \hat{\pi}_{\hat{\lambda}(\theta);\theta}(a|s) - \log \pi_{\lambda^*(\theta);\theta}(a|s)| \leq O(1/\bar{N}^{1-\bar{\eta}} + \gamma^{\bar{N}})$ *for any* $(s, a) \in \mathcal{S} \times \mathcal{A}$.

Lemma 4 shows that if the iteration $\bar{N}$ of middle loop is sufficiently large, the approximation error of $\lambda^*(\theta)$ and $\pi_{\lambda^*(\theta);\theta}$ can be arbitrarily small. We next quantify the convergence of the outer loop:

**Theorem 3** (convergence of the outer loop). *Suppose Assumption 1 and the conditions in Lemma 4 hold and let* $\beta_n = 1/(n + 1)^\eta$ *where* $\eta \in (1/2, 1)$*, then it holds* $\frac{1}{N} \sum_{n=0}^{N-1} ||\nabla J_r(\pi_{\lambda^*(\theta_n);\theta_n})||^2 \leq O(1/N^{1-\eta} + 1/\bar{N}^{2-2\bar{\eta}} + \gamma^{2\bar{N}})$.

Theorem 3 shows that Algorithm 1 converges to stationarity when the iteration numbers $N$ and $\bar{N}$ go to infinity. When the state-action space is finite, we have the following stronger result:

**Theorem 4** (optimality of the outer loop). *Suppose the conditions in Lemma 4 hold and the state-action space is finite. Let the step size* $\beta_n \leq \min\{(1 - \gamma)^3/8, 1/\bar{L}\}$*, then it holds that* $\lim_{N\to\infty} \lim_{\bar{N}\to\infty} J_r(\hat{\pi}_{\hat{\lambda}(\theta_N);\theta_N}) - J^*_r = 0$ *where* $J^*_r$ *is the maximum value of* $J_r(\pi)$*, and* $\bar{L}$ *is the smoothness constant of* $J_r(\pi_{\lambda^*(\theta);\theta})$ *whose expression is in Lemma 9 in Appendix.*

Theorem 4 shows that when the state-action space is finite, Algorithm 1 can find an optimal policy asymptotically when the iteration numbers $N$ and $\bar{N}$ go to infinity.

## 5 Experiment

This section provides experiment results for the proposed framework. In specific, we aim to answer the question: How does the proposed framework (UTILITY) compare to other RL improvement methods in terms of improving the RL performance. Towards this end, we introduce three RL

Table 1: Experiment results.

| | SAC | UITLITY | RICE | SIL | LIR |
|---|---|---|---|---|---|
| Delayed HalfCheetah | $383.45 \pm 45.50$ | $715.96 \pm 42.78$ | $456.14 \pm 36.32$ | $510.34 \pm 39.28$ | $548.28 \pm 47.94$ |
| Delayed Hopper | $192.90 \pm 27.18$ | $317.99 \pm 19.62$ | $232.55 \pm 16.96$ | $263.46 \pm 20.72$ | $247.27 \pm 31.93$ |
| Delayed Walker2d | $134.91 \pm 20.80$ | $242.63 \pm 14.11$ | $177.45 \pm 20.14$ | $172.28 \pm 24.57$ | $204.72 \pm 25.99$ |
| Delayed Ant | $68.11 \pm 12.52$ | $105.80 \pm 14.38$ | $77.01 \pm 10.89$ | $81.05 \pm 13.43$ | $78.23 \pm 13.11$ |

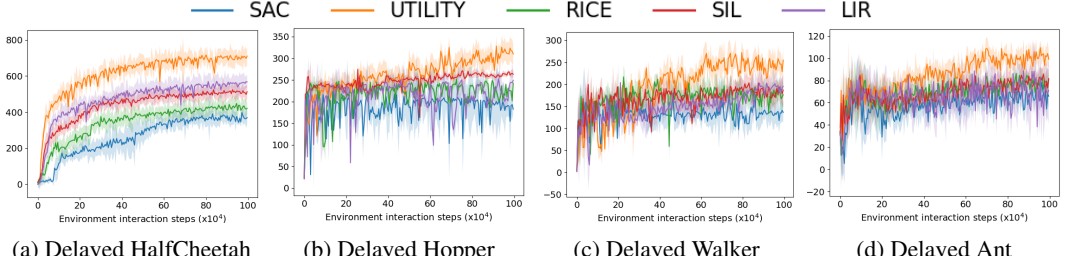

| (a) Delayed HalfCheetah | (b) Delayed Hopper | (c) Delayed Walker | (d) Delayed Ant |
|---|---|---|---|

Figure 2: Improvement curve.

improvement methods for comparisons. (i) **Fine-tune policy on initial states and critical states (RICE)** (Cheng et al., 2024): This method fine-tunes the policy starting at the original initial states and the states that are most influential to the cumulative reward. Note that (Guo et al., 2021; Cheng et al., 2023) also use the most influential states to improve performance and (Cheng et al., 2024) shows performance superiority over (Guo et al., 2021; Cheng et al., 2023), thus we pick (Cheng et al., 2024) to compare. (ii) **Self imitation learning (SIL)** (Oh et al., 2018): This method reproduces previous good decisions in order to encourage deep exploration. (iii) **Learning intrinsic reward (LIR)**: This method aims to learn an intrinsic reward $\tilde{r}$ to formulate the shaping reward $r + \tilde{r}$ (Zheng et al., 2018). We choose these three methods to compare because they respectively belong to three different categories: XRL method (RICE), reward shaping method (LIR), and other methods that can improve RL (SIL). We use soft actor-critic (SAC) (Haarnoja et al., 2018) as the baseline RL algorithm that all the above RL improvement methods use and improve from. We aim to show the improvement of the above RL improvement methods compared to SAC. We first run SAC until convergence to get the non-optimal policy $\pi_A$ and use $\pi_A$ to generate the demonstration set $\mathcal{D}$ for UTILITY to generate the two-level explanation and thus improve.

We test the algorithms on delayed MuJoCo environments (Zheng et al., 2018; Memarian et al., 2021; Oh et al., 2018) where the rewards are accumulated for 20 time steps and provided only at the end of these periods. Note that this makes the reward become sparse, and we include the additional experiment results on dense reward in Appendix F.1. We use four different delayed MuJoCo tasks: delayed HalfCheetah, delayed Hopper, delayed Walker2d, and delayed Ant. Following (Finn et al., 2017), each episode has the length of 100 in our experiments.

Figure 2 shows the learning curves of the algorithms where the $x$-axis is the interaction steps with the MDP environment and the $y$-axis is the cumulative reward. We plot both the mean (i.e., the solid line) and standard deviation (i.e, the shadow area) of the algorithms. The mean and standard deviation are computed using five random seeds. From the figures, we can observe that UTILITY improves the baseline RL algorithm SAC by a large margin. While the other three methods (i.e., RICE, SIL, and LIR) can also improve SAC to some extent, UTILITY achieves the highest cumulative reward. This is due to the fact that UTILITY (i) learns a shaping reward that makes it easier to learn a good policy and (ii) discourages the policy from making the mistakes (i.e., the "misleading" state-action pairs) made by SAC. Note that the learned shaping reward is dense while the ground truth reward in the delayed MuJoCo environments is sparse, so that UTILITY can guide the learned policy to achieve higher reward. In contrast, RICE and SIL still suffer from the sparse reward. While LIR can also learn a dense shaping reward, UTILITY has the domain knowledge $r - \hat{r}$ formulated by the high-level explanation to help better shape the reward. Moreover, UTILITY has the "misleading" state-action pairs to avoid. Note that the learned shaping reward not only helps in the sparse reward scenario, we include additional results in Appendix F.1 to show that UTILITY can still largely improve SAC when the ground truth reward is dense.

Table 1 shows the final performance of the algorithms. We can observe that UTILITY has the highest cumulative reward among all the algorithms.

**The ablation study**. Since UTILITY uses both the high-level explanation (i.e., the learned reward) to shape the ground truth reward and the low-level explanation (i.e., misleading state-action pairs) to formulate a constraint to improve SAC, we include an ablation study to separately study the effect of the shaping reward and the constraint. In specific, we consider two methods: "shaping only" and "constraint only". The "shaping only" method only uses the high-level explanation to learn a shaping reward but does not uses the low-level explanation to formulate a constraint. The "constraint only" method only uses the low-level explanation to formulate the constraint but does not shape the original reward. We include the results for the delayed environments in Table 2 and the results for the dense environments in Appendix F.2. The results in Table 2 and Appendix F.2 show that both the "shaping only" and "constraint only" methods can improve SAC. Moreover, the "shaping only" method has a larger impact to improve the performance. This is because the shaping reward improves the policy globally as it changes the reward value for all $(s, a)$, while the constraint may only improve the policy locally around the misleading state-action pairs.

Table 2: Ablation study.

|  | SAC | UITLITY | shaping only | constraint only |
| --- | --- | --- | --- | --- |
| Delayed HalfCheetah | $383.45 \pm 45.50$ | $715.96 \pm 42.78$ | $695.63 \pm 33.66$ | $422.15 \pm 22.86$ |
| Delayed Hopper | $192.90 \pm 27.18$ | $317.99 \pm 19.62$ | $289.10 \pm 18.41$ | $210.12 \pm 15.77$ |
| Delayed Walker2d | $134.91 \pm 20.80$ | $242.63 \pm 14.11$ | $211.37 \pm 18.64$ | $175.66 \pm 15.27$ |
| Delayed Ant | $68.11 \pm 12.52$ | $105.80 \pm 14.38$ | $88.18 \pm 8.66$ | $75.16 \pm 6.58$ |

**Evaluation of the generated two-level explanation**. Following (Guo et al., 2021; Cheng et al., 2023), we use fidelity as the metric to respectively evaluate the high-level and low-level explanations. The fidelity means the correctness of the two-level explanation. Since the two-level explanation is to explain why the RL agent is not optimal, one way to validate the fidelity of the explanation is to see whether the cumulative reward increases after we improve from the explanations.

From the last two columns in Table 2, we can see that both the high-level and low-level explanations are correct explanations because both the shaping only method and the constraint only method improve the performance. Moreover, the shaping only method (the fourth column in Table 2) has a higher cumulative reward than LIR (the last column in Table 1), and the constraint only method (the last column in Table 2) has a higher cumulative reward than RICE (the fourth column in Table 1). This shows the high fidelity of our two-level explanation.

To compare the fidelity of our explanation with other methods, we fix the improvement method and change the explanation to compare. For the low-level explanation, we compare with RICE. In specific, we still use the constraint only method but now the constraint is to discourage from visiting the critical states identified by RICE. We refer to this method as "RICE+constaint". Note that it is expected that "RICE+constaint" has low fidelity in our case because RICE does not aim to explain why the RL agent is not optimal. For the high-level explanation, since there is no existing XRL method to compare, we use our shaping-only method without the domain knowledge $r - \hat{r}$ to compare. We refer to this method as "shaping without $r - \hat{r}$". We include the results for sparse reward and dense reward in Appendix F.3. The results show that both the high-level and low-level explanations of UTILITY have high fidelity.

## 6 CONCLUSION

This paper proposes a theoretical and systematic framework that utilizes XRL to improve RL. We first provide an explanation for why the RL agent is not optimal, and then formulate the problem of utilizing the explanation to improve RL as a constrained bi-level optimization problem. We propose a novel theoretical framework to solve this problem, and use experiments to validate that the proposed framework can improve the RL performance. Despite the benefit, one limitation of the proposed algorithm is that it requires to interact with the environment. Therefore, one future work is to extend our method to the offline RL setting.

## 7 ACKNOWLEDGEMENTS

This work is partially supported by the National Science Foundation through grants ECCS 1846706 and ECCS 2140175. We thank the reviewers for their insightful and constructive suggestions.

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

## A  APPENDIX

This appendix has three sections. Section B provides additional content of the two-level explanation. Section C provides notions and notations that serve as the building blocks of the appendix. Section D provides the proof of all the lemmas and theorems in the paper. Section F provides experiment details.

## B  TWO-LEVEL EXPLANATION

This section has two subsections. Subsection B.1 provides the derivation steps of how we come up with the mathematical metric $l$ to find "misleading" state-action pairs in Section 3 and proves that a state-action pair $(s, a) \in \mathcal{D}$ is misleading if $l(s, a) > 0$. Subsection B.2 provides an algorithm to learn the $Q$-function $Q_r^{\hat{\pi}_A}$.

### B.1 Justification of the mathematical metric to find the "misleading" state-action pairs

Since the misleading state-action pairs lead the policy $\pi_A$ to be non-optimal, we can say that the policy $\pi_A$ will be an optimal policy if it does not visit misleading state-action pairs. Therefore, in order to identify "misleading" state-action pairs using $Q$-functions, we need to first build a connection between $Q$-functions and policy:

**Definition 2.** *(i) When we say that a Q-function $\bar{Q}_r^\pi$ **indicates** a policy $\pi$, it means that $\pi(s) = \arg\max_a \bar{Q}_r^\pi(s,a)$ for any $s \in \mathcal{S}$ where $\pi(s)$ is the set of actions that the policy $\pi$ will choose at the state $s$.*
*(ii) We use $\bar{\mathcal{Q}}_r^\pi$ to denote the set of all the Q-functions that indicate the policy $\pi$, and thus we can say that $\bar{\mathcal{Q}}_r^\pi$ **indicates** the policy $\pi$. Moreover, when we say that $\bar{\mathcal{Q}}_r^\pi$ **indicates** an action $a$ at a state $s$, it means that $a \in \arg\max_{a'} \bar{Q}_r^\pi(s,a')$ where $\bar{Q}_r^\pi$ is an arbitrary Q-function in $\bar{\mathcal{Q}}_r^\pi$.*
*(iii) When we say that $Q_r^\pi$ **disagrees with** $\bar{\mathcal{Q}}_r^\pi$ **on** $(s,a)$, it means that $\bar{\mathcal{Q}}_r^\pi$ indicates the action $a$ at the state $s$ but $a \notin \arg\max_{a'} Q_r^\pi(s,a')$.*

Definition 2 establishes a connection between $Q$-functions and policy. With this connection, the following theorem provides a way to define a mathematical metric to find "misleading" state-action pairs using $Q$-functions:

**Theorem 5.** *The policy $\pi_A$ will be an optimal policy if $\pi_A$ never visits any state-action pair $(s,a)$, on which $Q_r^{\pi_A}$ disagrees with $\bar{\mathcal{Q}}_r^{\pi_A}$.*

*Proof.* Suppose there is no state-action pair $(s,a) \in \mathcal{S} \times \mathcal{A}$, on which $Q_r^{\pi_A}$ disagrees with $\bar{\mathcal{Q}}_r^{\pi_A}$. Then for any $(s,a) \in \mathcal{S} \times \mathcal{A}$, if $a \in \arg\max_{a'} \bar{Q}_r^{\pi_A}(s,a')$ where $\bar{Q}_r^{\pi_A} \in \bar{\mathcal{Q}}_r^{\pi_A}$, it holds that $a \in \arg\max_{a'} Q_r^{\pi_A}(s,a')$. Since $\pi_A(s) = \arg\max_a \bar{Q}_r^{\pi_A}(s,a)$ where $\bar{Q}_r^{\pi_A} \in \bar{\mathcal{Q}}_r^{\pi_A}$, it holds that $\pi_A(s) \subseteq \arg\max_a Q_r^{\pi_A}(s,a)$ for any $s \in \mathcal{S}$. Recall that $V_r^{\pi_A}$ and $Q_r^{\pi_A}$ are respectively the value function and $Q$-function of the policy $\pi_A$ under the reward function $r$, then it holds that $Q_r^{\pi_A}(s,a) = r(s,a) + \gamma E_{s' \sim P(\cdot|s,a)}[V_r^{\pi_A}(s')]$ and $V_r^{\pi_A}(s) = E_{a' \sim \pi_A(\cdot|s)}[Q_r^{\pi_A}(s,a')]$. Since $\pi_A(s) \subseteq \arg\max_a Q_r^{\pi_A}(s,a)$ for any $s \in \mathcal{S}$, then $V_r^{\pi_A}(s) = E_{a' \sim \pi_A(\cdot|s)}[Q_r^{\pi_A}(s,a')] = \max_{a'} Q_r^{\pi_A}(s,a')$ for any $s \in \mathcal{S}$. Therefore, the $Q$-function $Q_r^{\pi_A}$ satisfies the Bellman optimality equation $Q_r^{\pi_A}(s,a) = r(s,a) + \gamma E_{s' \sim P(s'|s,a)}[\max_{a'} Q_r^{\pi_A}(s',a')]$ for any $(s,a) \in \mathcal{S} \times \mathcal{A}$, and thus $Q_r^{\pi_A}$ is the optimal $Q$-function because the Bellman optimality equation is uniquely satisfied by the optimal $Q$-function. Since $\pi_A(s) \subseteq \arg\max_a Q_r^{\pi_A}(s,a)$ for any $s \in \mathcal{S}$, the policy $\pi_A$ should be an optimal policy. $\square$

Since the misleading state-action pairs lead the policy $\pi_A$ to be non-optimal, we can say that the policy $\pi_A$ will be an optimal policy if it has zero probability to visit misleading state-action pairs. Therefore, Theorem 5 shows that the "misleading" state-action pairs can be mathematically defined as the state-action pairs, on which $Q_r^{\pi_A}$ disagrees with $\bar{\mathcal{Q}}_r^{\pi_A}$. Therefore, we can develop a mathematical metric to find the top $K$ "misleading" state-action pairs in the demonstration set $\mathcal{D}$ using $Q$-functions. Since the demonstration set $\mathcal{D}$ is generated by the policy $\pi_A$ and $\bar{\mathcal{Q}}_r^{\pi_A}$ indicates the policy $\pi_A$, $\bar{\mathcal{Q}}_r^{\pi_A}$ will indicate the action $a$ at the state $s$ for any $(s,a) \in \mathcal{D}$. In order to find the "misleading" state-actions, we need to find $(s,a) \in \mathcal{D}$ such that $a \notin \arg\max_{a'} Q_r^{\pi_A}(s,a')$ or in other words, $Q_r^{\pi_A}(s,a) < \max_{a'} Q_r^{\pi_A}(s,a')$. Therefore, we can use the function $l(s,a) \triangleq \max_{a'} Q_r^{\pi_A}(s,a') - Q_r^{\pi_A}(s,a)$ as the metric to identify "misleading" state-action pairs in the demonstration set $\mathcal{D}$. The larger the loss $l(s,a)$ is, the more "misleading" the state-action pair $(s,a)$ is, because the $Q$ value of the chosen action $a$ has a larger gap from the maximum $Q$ value at the state $s$ under the $Q$-function $Q_r^{\pi_A}$.

### B.2 Method

In this subsection, we use a standard regression method to learn the Q-function $Q_r^{\hat{\pi}_A}$. In specific, we roll out the policy $\hat{\pi}_A$ to generate a set $\bar{\mathcal{D}}$ of many $(s,a)$ samples. For each $(s,a) \in \mathcal{D}$, we use $\hat{\pi}_A$ to generate many trajectories starting from $(s,a)$ and use these trajectories to estimate $Q_r^{\hat{\pi}_A}(s,a)$. Since we can generate many trajectories, we can estimate the $Q$ value $Q_r^{\hat{\pi}_A}(s,a)$ for each $(s,a) \in \bar{\mathcal{D}}$ quite accurately. Then we use a neural network $Q_\omega$ parameterized by $\omega$ to solve the following

regression problem:

$$\min_{\omega} \sum_{(s,a)\in\mathcal{D}} ||Q_{\omega}(s,a) - Q_r^{\hat{\pi}_A}(s,a)||^2.$$

## C  NOTIONS AND NOTATIONS

The shaping reward function $r_{\theta}(r(s,a), r(s,a) - \hat{r}(s,a))$ is a function of $r(s,a)$ and $r(s,a) - \hat{r}(s,a)$, and $r(s,a)$ and $r(s,a) - \hat{r}(s,a)$ are both functions of $(s,a)$. Therefore, the shaping reward function $r_{\theta}$ is also a function of $(s,a)$. For simple notations, we use $r_{\theta}(s,a)$ to denote the shaping reward of $(s,a)$. Given the policy $\pi$ and the parameters $(\lambda, \theta)$s, the corresponding constrained soft Q-function and constrained soft value function are:

$$Q_{\lambda;\theta}^{\pi}(s,a) \triangleq r_{\theta}(s,a) - \lambda c(s,a) + \gamma \int_{s'\in\mathcal{S}} P(s'|s,a)V_{\lambda;\theta}^{\pi}(s')ds',$$

$$V_{\lambda;\theta}^{\pi}(s) \triangleq E^{\pi}\left[\sum_{t=0}^{\infty} \gamma^t (r_{\theta}(s_t,a_t) - \lambda c(s_t,a_t) - \log\pi(a_t|s_t))\bigg|s_0 = s\right].$$

Moreover, it can be shown (Zeng et al., 2022; Haarnoja et al., 2017; Liu & Zhu, 2024a;b) that $\exp(V_{\lambda;\theta}^{\pi}(s)) = \int_{a\in\mathcal{A}} \exp(Q_{\lambda;\theta}^{\pi}(s,a))da$.

The constrained soft policy (Liu & Zhu, 2022; 2023) is

$$\pi_{\lambda;\theta}(a|s) = \frac{\exp(Q_{\lambda;\theta}(s,a))}{\exp(V_{\lambda;\theta}(s))}, \tag{3}$$

$$Q_{\lambda;\theta}(s,a) = r_{\theta}(s,a) - \lambda c(s,a) + \gamma \int_{s'\in\mathcal{S}} P(s'|s,a)V_{\lambda;\theta}(s')ds', \tag{4}$$

$$V_{\lambda;\theta}(s) = \log \int_{a\in\mathcal{A}} \exp(Q_{\lambda;\theta}(s,a))da. \tag{5}$$

We can obtain the constrained soft policy via soft Q-learning (Haarnoja et al., 2017) or soft actor-critic (Haarnoja et al., 2018) by treating $r_{\theta} - \lambda c$ as the new reward function. We define the cumulative cost under the policy $\pi$ starting from $(s,a)$ as $Q_c^{\pi}(s,a) \triangleq E^{\pi}[\sum_{t=0}^{\infty} \gamma^t c(s_t,a_t)|s_0 = s, a_0 = a]$ and the cumulative cost starting from $s$ as $V_c^{\pi}(s) \triangleq E^{\pi}[\sum_{t=0}^{\infty} \gamma^t c(s_t,a_t)|s_0 = s]$. We define the cumulative reward under the policy $\pi$ starting from $(s,a)$ as $Q_r^{\pi}(s,a) \triangleq E^{\pi}[\sum_{t=0}^{\infty} \gamma^t r(s_t,a_t)|s_0 = s, a_0 = a]$ and the cumulative reward starting from $s$ as $V_r^{\pi}(s) \triangleq E^{\pi}[\sum_{t=0}^{\infty} \gamma^t r(s_t,a_t)|s_0 = s]$. We define the cumulative reward gradient under the policy $\pi$ starting from $(s,a)$ as $Q_{\nabla_{\theta}r_{\theta}}^{\pi}(s,a) \triangleq E^{\pi}[\sum_{t=0}^{\infty} \gamma^t \nabla_{\theta}r_{\theta}(s_t,a_t)|s_0 = s, a_0 = a]$ and the cumulative reward starting from $s$ as $V_{\nabla_{\theta}r_{\theta}}^{\pi}(s) \triangleq E^{\pi}[\sum_{t=0}^{\infty} \gamma^t \nabla_{\theta}r_{\theta}(s_t,a_t)|s_0 = s]$. We define the state visitation frequency under a policy $\pi$ as $\psi^{\pi}(s) \triangleq E^{\pi}[\sum_{t=0}^{\pi} \gamma^t \mathbb{1}\{s_t = s\}]$ and state-action visitation frequency as $\psi^{\pi}(s,a) \triangleq E^{\pi}[\sum_{t=0}^{\pi} \gamma^t \mathbb{1}\{s_t = s, a_t = a\}]$ where $\mathbb{1}\{\cdot\}$ is the indicator function.

## D  PROOF

This section provides the proof of all the lemmas and theorems in the paper.

**Lemma 5.** *The gradients of the constrained soft policy are respectively* $\nabla_{\lambda}\log\pi_{\lambda;\theta}(a|s) = V_c^{\pi_{\lambda;\theta}}(s) - Q_c^{\pi_{\lambda;\theta}}(s,a)$ *and* $\nabla_{\theta}\log\pi_{\lambda;\theta}(a|s) = Q_{\nabla_{\theta}r_{\theta}}^{\pi_{\lambda;\theta}}(s,a) - V_{\nabla_{\theta}r_{\theta}}^{\pi_{\lambda;\theta}}(s)$.

*Proof.* Recall from (3), we know that $\nabla_{\lambda}\log\pi_{\lambda;\theta}(a|s) = \nabla_{\lambda}Q_{\lambda;\theta}(s,a) - \nabla_{\lambda}V_{\lambda;\theta}(s)$. Recall from (4), we know that

$$\nabla_{\lambda}Q_{\lambda;\theta}(s,a) = -c(s,a) + \gamma \int_{s'\in\mathcal{S}} P(s'|s,a)\nabla_{\lambda}V_{\lambda;\theta}(s')ds',$$

$$\overset{(a)}{=} -c(s,a) + \gamma \int_{s'\in\mathcal{S}} P(s'|s,a)\frac{1}{\exp(V_{\lambda;\theta}(s'))} \int_{a'\in\mathcal{A}} \nabla_{\lambda}\exp(Q_{\lambda;\theta}(s',a'))da'ds',$$

$$= -c(s,a) + \gamma \int_{s' \in \mathcal{S}} \int_{a' \in \mathcal{A}} P(s'|s,a) \frac{\exp(Q_{\lambda;\theta}(s',a'))}{\exp(V_{\lambda;\theta}(s'))} \nabla_\lambda Q_{\lambda;\theta}(s',a') da' ds', ,$$

$$\overset{(b)}{=} -c(s,a) + \gamma \int_{s' \in \mathcal{S}} \int_{a' \in \mathcal{A}} P(s'|s,a) \pi_{\lambda;\theta}(a|s) \nabla_\lambda Q_{\lambda;\theta}(s',a') da' ds',$$

$$= -c(s,a) + \gamma \int_{s' \in \mathcal{S}} \int_{a' \in \mathcal{A}} P(s'|s,a) \pi_{\lambda;\theta}(a|s) \left[ -c(s',a') + \gamma \int_{s'' \in \mathcal{S}} P(s''|s',a') \nabla_\lambda V_{\lambda;\theta}(s'') ds'' \right],$$

where $(a)$ follows (5), $(b)$ follows (3). Keep the expansion, we can see that

$$\nabla_\lambda Q_{\lambda;\theta}(s,a) = -E^{\pi_{\lambda;\theta}} \left[ \sum_{t=0}^{\infty} \gamma^t c(s_t, a_t) | s_0 = s, a_0 = a \right] = -Q_c^{\pi_{\lambda;\theta}}(s,a). \tag{6}$$

Similarly, we can get that:

$$\nabla_\lambda V_{\lambda;\theta}(s) = -E^{\pi_{\lambda;\theta}} \left[ \sum_{t=0}^{\infty} \gamma^t c(s_t, a_t) | s_0 = s \right] = -V_c^{\pi_{\lambda;\theta}}(s), \tag{7}$$

$$\nabla_\theta Q_{\lambda;\theta}(s,a) = E^{\pi_{\lambda;\theta}} \left[ \sum_{t=0}^{\infty} \gamma^t \nabla_\theta r_\theta(s_t, a_t) | s_0 = s, a_0 = a \right] = Q_{\nabla_\theta r_\theta}^{\pi_{\lambda;\theta}}(s,a), \tag{8}$$

$$\nabla_\theta V_{\lambda;\theta}(s) = E^{\pi_{\lambda;\theta}} \left[ \sum_{t=0}^{\infty} \gamma^t \nabla_\theta r_\theta(s_t, a_t) | s_0 = s \right] = V_{\nabla_\theta r_\theta}^{\pi_{\lambda;\theta}}(s), \tag{9}$$

Therefore, we can compute the gradients

$$\nabla_\lambda \log \pi_{\lambda;\theta}(a|s) = \nabla_\lambda Q_{\lambda;\theta}(s,a) - \nabla_\lambda V_{\lambda;\theta}(s) = V_c^{\pi_{\lambda;\theta}}(s) - Q_c^{\pi_{\lambda;\theta}}(s,a),$$
$$\nabla_\theta \log \pi_{\lambda;\theta}(a|s) = \nabla_\theta Q_{\lambda;\theta}(s,a) - \nabla_\theta V_{\lambda;\theta}(s) = Q_{\nabla_\theta r_\theta}^{\pi_{\lambda;\theta}}(s,a) - V_{\nabla_\theta r_\theta}^{\pi_{\lambda;\theta}}(s).$$

$\square$

**Lemma 6.** *The constrained soft operator $\mathcal{T}_{\lambda;\theta}^{soft}$:*

$$(\mathcal{T}_{\lambda;\theta}^{soft} Q)(s,a) \triangleq r_\theta(s,a) - \lambda c(s,a) + \gamma \int_{s' \in \mathcal{S}} P(s'|s,a) \log \left[ \int_{a' \in \mathcal{A}} \exp(Q(s',a')) da' \right] ds',$$

$$(\mathcal{T}_{\lambda;\theta}^{soft} V)(s) \triangleq \log \left[ \int_{a \in \mathcal{A}} \exp \left( r_\theta(s,a) - \lambda c(s,a) + \gamma \int_{s' \in \mathcal{S}} P(s'|s,a) V(s') ds' \right) da \right],$$

*is a contraction map with constant $\gamma$.*

*Proof.* It has been proved that $\mathcal{T}_{\lambda;\theta}^{soft} Q$ is a contraction map with constant $\gamma$ (Appendix A.2 in (Haarnoja et al., 2017) if we replace $r$ with $r_\theta - \lambda c$). Here we show that $\mathcal{T}_{\lambda;\theta}^{soft} V$ is a contraction map with constant $\gamma$. Define a norm of $V$ as $||V_1 - V_2|| = \sup_{s \in \mathcal{S}} |V_1(s) - V_2(s)|$ and suppose $||V_1 - V_2|| = \epsilon$. Then we have that

$$\mathcal{T}_{\lambda;\theta}^{soft} V_1(s) = \log \left[ \int_{a \in \mathcal{A}} \exp \left( r_\theta(s,a) - \lambda c(s,a) + \gamma \int_{s' \in \mathcal{S}} P(s'|s,a) V_1(s') ds' \right) da \right],$$

$$\leq \log \left[ \int_{a \in \mathcal{A}} \exp \left( r_\theta(s,a) - \lambda c(s,a) + \gamma \int_{s' \in \mathcal{S}} P(s'|s,a) [V_2(s') + \epsilon] ds' \right) da \right],$$

$$= \log \left[ \int_{a \in \mathcal{A}} \exp \left( r_\theta(s,a) - \lambda c(s,a) + \gamma \int_{s' \in \mathcal{S}} P(s'|s,a) V_2(s') ds' + \gamma \epsilon \right) da \right],$$

$$= \log \left[ \int_{a \in \mathcal{A}} \exp(\gamma \epsilon) \exp \left( r_\theta(s,a) - \lambda c(s,a) + \gamma \int_{s' \in \mathcal{S}} P(s'|s,a) V_2(s') ds' \right) da \right],$$

$$= \mathcal{T}_{\lambda;\theta}^{soft} V_2(s) + \gamma \epsilon.$$

Similarly, we can get $\mathcal{T}_{\lambda;\theta}^{soft} V_1(s) \geq \mathcal{T}_{\lambda;\theta}^{soft} V_2(s) - \gamma \epsilon$. Therefore, $||\mathcal{T}_{\lambda;\theta}^{soft} V_1 - \mathcal{T}_{\lambda;\theta}^{soft} V_2|| \leq \gamma \epsilon = \gamma ||V_1 - V_2||$.

$\square$

**Lemma 7.** *It holds that* $Q_{\lambda;\theta}^{\pi_{\tilde{n}+1}}(s,a) \geq \mathcal{T}_{\lambda;\theta}^{soft}(Q_{\lambda;\theta}^{\pi_{\tilde{n}}})(s,a)$ *and* $V_{\lambda;\theta}^{\pi_{\tilde{n}+1}}(s) \geq \mathcal{T}_{\lambda;\theta}^{soft}(V_{\lambda;\theta}^{\pi_{\tilde{n}}})(s)$ *for any* $(s,a)$.

*Proof.*

$$Q_{\lambda;\theta}^{\pi_{\tilde{n}+1}}(s,a) \overset{(a)}{=} r_\theta(s,a) - \lambda c(s,a) + \gamma \int_{s'\in\mathcal{S}} P(s'|s,a) E_{a'\sim\pi_{\tilde{n}+1}}[Q_{\lambda;\theta}^{\pi_{\tilde{n}+1}}(s',a') - \log\pi_{\tilde{n}+1}(a'|s')]ds',$$

$$\overset{(b)}{\geq} r_\theta(s,a) - \lambda c(s,a) + \gamma \int_{s'\in\mathcal{S}} P(s'|s,a) E_{a'\sim\pi_{\tilde{n}+1}(\cdot|s')}[Q_{\lambda;\theta}^{\pi_{\tilde{n}}}(s',a') - \log\pi_{\tilde{n}+1}(a'|s')]ds',$$

$$= r_\theta(s,a) - \lambda c(s,a) + \gamma \int_{s'\in\mathcal{S}} P(s'|s,a) \log\left[\int_{a'\in\mathcal{A}} \exp(Q_{\lambda;\theta}^{\pi_{\tilde{n}}}(s',a'))da'\right]ds',$$

$$= \mathcal{T}_{\lambda;\theta}^{soft}(Q_{\lambda;\theta}^{\pi_{\tilde{n}}})(s,a),$$

where $(a)$ follow equations (2)-(3) in (Haarnoja et al., 2018) and $(b)$ follows policy improvement theorem (Theorem 4 in (Haarnoja et al., 2017)). Similarly, we can get that

$$V_{\lambda;\theta}^{\pi_{\tilde{n}+1}}(s) = E_{a\sim\pi_{\tilde{n}+1}(\cdot|s)}[Q_{\lambda;\theta}^{\pi_{\tilde{n}+1}}(s,a) - \log\pi_{\tilde{n}+1}(a|s)],$$

$$\geq E_{a\sim\pi_{\tilde{n}+1}(\cdot|s)}[Q_{\lambda;\theta}^{\pi_{\tilde{n}}}(s,a) - \log\pi_{\tilde{n}+1}(a|s)],$$

$$= \log\left[\int_{a\in\mathcal{A}} \exp(Q_{\lambda;\theta}^{\pi_{\tilde{n}}}(s,a))da\right],$$

$$= \log\left[\int_{a\in\mathcal{A}} \exp\left(r_\theta(s,a) - \lambda c(s,a) + \gamma \int_{s'\in\mathcal{S}} P(s'|s,a) V_{\lambda;\theta}^{\pi_{\tilde{n}}}(s')ds'\right)da\right],$$

$$= \mathcal{T}_{\lambda;\theta}^{soft}(V_{\lambda;\theta}^{\pi_{\tilde{n}}})(s).$$

$\square$

## D.1 PROOF OF LEMMA 1

It has been proved in Theorem 1 in (Haarnoja et al., 2017) that the constrained soft policy $\pi_{\lambda;\theta} = \arg\max_\pi J_{r_\theta}(\pi) - \lambda J_c(\pi) + H(\pi)$ where we treat $r_\theta - \lambda c$ as the new reward function. Recall that the dual function is $G(\lambda;\theta) = \max_\pi J_{r_\theta}(\pi) - \lambda(J_c(\pi) - b) + H(\pi)$, therefore, we know that $G(\lambda;\theta) = J_{r_\theta}(\pi_{\lambda;\theta}) - \lambda(J_c(\pi_{\lambda;\theta}) - b) + H(\pi_{\lambda;\theta})$. Since $\pi_{\lambda;\theta}$ is the optimal solution and the $G$ is differentiable to the policy, we know that $\frac{\partial G(\lambda;\theta)}{\partial\pi_{\lambda;\theta}(a|s)} = 0$ for every $(s,a) \in \mathcal{S} \times \mathcal{A}$. Note that

$$\frac{\partial G(\lambda;\theta)}{\partial\pi_{\lambda;\theta}(a|s)} = \frac{\partial}{\partial\pi_{\lambda;\theta}(a|s)} \psi^{\pi_{\lambda;\theta}}(s,a)\left[r_\theta(s,a) - \lambda c(s,a) - \log\pi_{\lambda;\theta}(a|s)\right] = 0 \qquad (10)$$

$$\nabla_\lambda G(\lambda;\theta) = \nabla_\lambda\left[J_{r_\theta}(\pi_{\lambda;\theta}) - \lambda J_c(\pi_{\lambda;\theta}) + H(\pi_{\lambda;\theta})\right] - (J_c(\pi_{\lambda;\theta}) - b),$$

$$= \int_{s\in\mathcal{S}}\int_{a\in\mathcal{A}} \nabla_\lambda\left\{\psi^{\pi_{\lambda;\theta}}(s,a)\left[r_\theta(s,a) - \lambda c(s,a) - \log\pi_{\lambda;\theta}(a|s)\right]\right\}dads - (J_c(\pi_{\lambda;\theta}) - b),$$

$$= \int_{s\in\mathcal{S}}\int_{a\in\mathcal{A}} \nabla_\lambda\pi_{\lambda;\theta}(a|s) \cdot \frac{\partial}{\partial\pi_{\lambda;\theta}(a|s)}\left\{\psi^{\pi_{\lambda;\theta}}(s,a)\left[r_\theta(s,a) - \lambda c(s,a) - \log\pi_{\lambda;\theta}(a|s)\right]\right\}dads$$

$$- (J_c(\pi_{\lambda;\theta}) - b),$$

$$\overset{(a)}{=} b - J_c(\pi_{\lambda;\theta}),$$

where $(a)$ follows (10).

## D.2 PROOF OF THEOREM 1 AND THEOREM 2

We first prove Theorem 2 and then prove Theorem 1. The fundamental logic is that we first prove that the dual problem has a unique optimal solution $\lambda^*(\theta)$ and then we prove that the primal problem

(i.e. the lower-level problem in (2)) and the dual problem have the same set of optimal solutions. Since the optimal solution of the dual problem is unique, then the optimal solution of the primal problem is also unique.

To show the optimal solution of the dual problem is unique, we prove that the dual function is strictly convex by showing that the Hessian of the dual function to $\lambda$ is positive definite.

From Lemma 1, we know that $\nabla_\lambda G(\lambda; \theta) = b - J_c(\pi_{\lambda;\theta})$, therefore, we have that

$$\nabla^2_{\lambda\lambda} G(\lambda; \theta) = -\nabla_\lambda J_c(\pi_{\lambda;\theta}),$$

$$= -\nabla_\lambda \int_{s_0 \in \mathcal{S}} P_0(s_0) \int_{a_0 \in \mathcal{A}} \pi_{\lambda;\theta}(a_0|s_0) \left[ c(s_0, a_0) + \gamma \int_{s_1 \in \mathcal{S}} P(s_1|s_0, a_0) Q_c^{\pi_{\lambda;\theta}}(s_1) ds_1 \right] da_0 ds_0,$$

$$= -\int_{s_0 \in \mathcal{S}} P_0(s_0) \int_{a_0 \in \mathcal{A}} \left\{ \nabla_\lambda \pi_{\lambda;\theta}(a_0|s_0) \cdot \left[ c(s_0, a_0) + \gamma \int_{s_1 \in \mathcal{S}} P(s_1|s_0, a_0) Q_c^{\pi_{\lambda;\theta}}(s_1) ds_1 \right] \right.$$

$$+ \left. \pi_{\lambda;\theta}(a_0|s_0) \cdot \left[ \gamma \int_{s_1 \in \mathcal{S}} P(s_1|s_0, a_0) \nabla_\lambda Q_c^{\pi_{\lambda;\theta}}(s_1) ds_1 \right] \right\} da_0 ds_0,$$

$$= -\int_{s_0 \in \mathcal{S}} P_0(s_0) \int_{a_0 \in \mathcal{A}} \left\{ \nabla_\lambda \pi_{\lambda;\theta}(a_0|s_0) \cdot Q_c^{\pi_{\lambda;\theta}}(s_0, a_0) \right.$$

$$+ \left. \pi_{\lambda;\theta}(a_0|s_0) \cdot \gamma \int_{s_1 \in \mathcal{S}} P(s_1|s_0, a_0) \nabla_\lambda \left[ \int_{a_1 \in \mathcal{A}} \pi_{\lambda;\theta}(a_1|s_1) \cdot Q_c^{\pi_{\lambda;\theta}}(s_1, a_1) da_1 ds_1 \right] \right\} da_0 ds_0.$$

Keep the expansion, we can get that

$$\nabla^2_{\lambda\lambda} G(\lambda; \theta) = -\int_{s \in \mathcal{S}} \int_{a \in \mathcal{A}} \psi^{\pi_{\lambda;\theta}}(s) \nabla_\lambda \pi_{\lambda;\theta}(a|s) \cdot Q_c^{\pi_{\lambda;\theta}}(s, a) da ds,$$

$$= -\int_{s \in \mathcal{S}} \int_{a \in \mathcal{A}} \psi^{\pi_{\lambda;\theta}}(s) \pi_{\lambda;\theta}(a|s) \nabla_\lambda \log \pi_{\lambda;\theta}(a|s) \cdot Q_c^{\pi_{\lambda;\theta}}(s, a) da ds,$$

$$\overset{(a)}{=} \int_{s \in \mathcal{S}} \psi^{\pi_{\lambda;\theta}}(s) \int_{a \in \mathcal{A}} \pi_{\lambda;\theta}(a|s) \left[ Q_c^{\pi_{\lambda;\theta}}(s, a) - V_c^{\pi_{\lambda;\theta}}(s) \right] Q_c^{\pi_{\lambda;\theta}}(s, a) da ds, \qquad (11)$$

where $(a)$ follows Lemma 5. Note that $V_c^{\pi_{\lambda;\theta}}(s) = E_{a \sim \pi_{\lambda;\theta}(\cdot|s)}[Q_c^{\pi_{\lambda;\theta}}(s, a)]$. If we use the random variable $X_{sa}$ to denote $Q_c^{\pi_{\lambda;\theta}}(s, a)$, then its expectation is $E_{a \sim \pi_{\lambda;\theta}(\cdot|s)}(X_{sa}) = V_c^{\pi_{\lambda;\theta}}(s)$. We know that the variance $Var(X_{sa}) = E[X_{sa}[X_{sa} - E(X_{sa})]]$. Therefore, we can see that the equation (11) is actually a variance:

$$\nabla^2_{\lambda\lambda} G(\lambda; \theta) = \int_{s \in \mathcal{S}} \psi^{\pi_{\lambda;\theta}}(s) Var(X_{sa}) ds.$$

From the expression (3), we know that $\pi_{\lambda;\theta}$ is always stochastic. Therefore, the variance $Var(X_{sa}) > 0$. Then we know that $\nabla^2_{\lambda\lambda} G(\lambda; \theta) > 0$ and thus $G$ is strictly convex. Therefore, the optimal solution $\lambda^*(\theta)$ is unique.

Since $G(\lambda; \theta)$ attains its minimum at $\lambda^*(\theta)$, the gradient of $G$ at $\lambda^*(\theta)$ should be zero, i.e., $J_c(\pi_{\lambda^*(\theta);\theta}) - b = 0$. Let $p^*$ and $d^*$ be the optimal value of the primal problem and the dual problem. Since $G(\lambda; \theta) = \max_\pi J_{r_\theta}(\pi) + H(\pi) - \lambda(J_c(\pi) - b)$, we know that $G(\lambda; \theta) \geq J_{r_\theta}(\pi) + H(\pi)$ for any $(\lambda, \theta)$, which means that $d^* \geq p^*$. Therefore, we have that:

$$p^* \leq d^* = G(\lambda^*(\theta); \theta) \overset{(b)}{=} J_{r_\theta}(\pi_{\lambda^*(\theta);\theta}) + H(\pi_{\lambda^*(\theta);\theta}) \leq p^*,$$

where $(b)$ follows the fact that $J_c(\pi_{\lambda^*(\theta);\theta}) - b = 0$. Therefore, $\pi_{\lambda^*(\theta);\theta}$ is an optimal solution of the primal problem. Suppose the primal problem has another optimal solution $\pi'$, then it holds that $\pi' \in \arg\max_\pi G(\lambda^*(\theta); \theta)$. However, it has been proved in Lemma 1 in (Zhou et al., 2017) that given an arbitrary $\lambda$, the optimal policy of $\max_\pi J_{r_\theta}(\pi) + H(\pi) - \lambda(J_c(\pi) - b)$ is unique. Therefore, $\pi_{\lambda^*(\theta);\theta}$ is the unique optimal solution of the primal problem (i.e., the lower-level problem in (2)).

### D.3 Proof of Lemma 2

Since $\lambda^*(\theta) = \arg\min G(\lambda; \theta)$, we know that $\nabla_\lambda G(\lambda^*(\theta); \theta) = 0$. Therefore, we have that

$$\frac{d\nabla_\lambda G(\lambda^*(\theta); \theta)}{d\theta} = 0,$$

$$\Rightarrow \nabla_{\theta\lambda}^2 G(\lambda^*(\theta);\theta) + \nabla_{\lambda\lambda}^2 G(\lambda^*(\theta);\theta)\nabla\lambda^*(\theta) = 0,$$

$$\Rightarrow \nabla\lambda^*(\theta) = -[\nabla_{\lambda\lambda}^2 G(\lambda^*(\theta);\theta)]^{-1}\nabla_{\theta\lambda}^2 G(\lambda^*(\theta);\theta). \tag{12}$$

Now we take a look at the term $\nabla_{\theta\lambda}^2 G(\lambda;\theta)$. From Lemma 1, we know that $\nabla_{\lambda} G(\lambda;\theta) = b - J_c(\pi_{\lambda;\theta})$, therefore, we have that

$$\nabla_{\lambda\theta}^2 G(\lambda;\theta) = -\nabla_{\theta} J_c(\pi_{\lambda;\theta}),$$

$$= -\nabla_{\theta} \int_{s_0 \in \mathcal{S}} P_0(s_0) \int_{a_0 \in \mathcal{A}} \pi_{\omega;\theta}(a_0|s_0) Q_c^{\pi_{\lambda;\theta}}(s_0,a_0) da_0 ds_0,$$

$$= -\int_{s_0 \in \mathcal{S}} P_0(s_0) \int_{a_0 \in \mathcal{A}} \pi_{\omega;\theta}(a_0|s_0)\Big[\nabla_{\theta}\log\pi_{\omega;\theta}(a_0|s_0)\cdot Q_c^{\pi_{\lambda;\theta}}(s_0,a_0) + \nabla_{\theta}Q_c^{\pi_{\lambda;\theta}}(s_0,a_0)\Big] da_0 ds_0,$$

$$= -\int_{s_0 \in \mathcal{S}} P_0(s_0) \int_{a_0 \in \mathcal{A}} \pi_{\omega;\theta}(a_0|s_0)\Big[\nabla_{\theta}\log\pi_{\omega;\theta}(a_0|s_0)\cdot Q_c^{\pi_{\lambda;\theta}}(s_0,a_0)$$

$$- \nabla_{\theta}[c(s_0,a_0) + \gamma\int_{s_1 \in \mathcal{S}} P(s_1|s_0,a_0)Q_c^{\pi_{\lambda;\theta}}(s_1)ds_1]\Big] da_0 ds_0,$$

$$= -\int_{s_0 \in \mathcal{S}} P_0(s_0) \int_{a_0 \in \mathcal{A}} \pi_{\omega;\theta}(a_0|s_0)\Big[\nabla_{\theta}\log\pi_{\omega;\theta}(a_0|s_0)\cdot Q_c^{\pi_{\lambda;\theta}}(s_0,a_0)$$

$$- \gamma\int_{s_1 \in \mathcal{S}} P(s_1|s_0,a_0)\nabla_{\theta}\int_{a_1 \in \mathcal{A}} \pi_{\lambda;\theta}(a_1|s_1)Q_c^{\pi_{\lambda;\theta}}(s_1,a_1)da_1 ds_1\Big] da_0 ds_0.$$

Keep the expansion, we can get that

$$\nabla_{\lambda\theta}^2 G(\lambda;\theta) = -\int_{s \in \mathcal{S}} \int_{a \in \mathcal{A}} \psi^{\pi_{\lambda;\theta}}(s,a)\nabla_{\theta}\log\pi_{\lambda;\theta}(a|s)Q_c^{\pi_{\lambda;\theta}}(s,a)dads,$$

$$\overset{(a)}{=} -\int_{s \in \mathcal{S}} \int_{a \in \mathcal{A}} \psi^{\pi_{\lambda;\theta}}(s,a)\Big[Q_{\nabla_{\theta}r_{\theta}}^{\pi_{\lambda;\theta}}(s,a) - V_{\nabla_{\theta}r_{\theta}}^{\pi_{\lambda;\theta}}(s)\Big]Q_c^{\pi_{\lambda;\theta}}(s,a)dads, \tag{13}$$

where $(a)$ follows Lemma 5.

Now, we take the full gradient of $\log\pi_{\lambda^*(\theta);\theta}(a|s)$ to $\theta$:

$$\frac{d\log\pi_{\lambda^*(\theta);\theta}(a|s)}{d\theta} = \nabla_{\theta}\log\pi_{\lambda^*(\theta);\theta}(a|s) + \nabla_{\lambda}\log\pi_{\lambda^*(\theta);\theta}(a|s)\cdot\nabla\lambda^*(\theta),$$

$$\overset{(b)}{=} Q_{\nabla_{\theta}r_{\theta}}^{\pi_{\lambda^*(\theta);\theta}}(s,a) - V_{\nabla_{\theta}r_{\theta}}^{\pi_{\lambda^*(\theta);\theta}}(s) + (Q_c^{\pi_{\lambda^*(\theta);\theta}}(s,a) - V_c^{\pi_{\lambda^*(\theta);\theta}}(s))[\nabla_{\lambda\lambda}^2 G(\lambda^*(\theta);\theta)]^{-1}\nabla_{\theta\lambda}^2 G(\lambda^*(\theta);\theta),$$

$$\overset{(c)}{=} Q_{\nabla_{\theta}r_{\theta}}^{\pi_{\lambda^*(\theta);\theta}}(s,a) - V_{\nabla_{\theta}r_{\theta}}^{\pi_{\lambda^*(\theta);\theta}}(s) - (Q_c^{\pi_{\lambda^*(\theta);\theta}}(s,a) - V_c^{\pi_{\lambda^*(\theta);\theta}}(s))\cdot$$

$$\frac{\int_{s \in \mathcal{S}}\int_{a \in \mathcal{A}}\psi^{\pi_{\lambda^*(\theta);\theta}}(s,a)\Big[Q_{\nabla_{\theta}r_{\theta}}^{\pi_{\lambda^*(\theta);\theta}}(s,a) - V_{\nabla_{\theta}r_{\theta}}^{\pi_{\lambda^*(\theta);\theta}}(s)\Big]Q_c^{\pi_{\lambda^*(\theta);\theta}}(s,a)dads}{\int_{s \in \mathcal{S}}\int_{a \in \mathcal{A}}\psi^{\pi_{\lambda^*(\theta);\theta}}(s,a)\Big[Q_c^{\pi_{\lambda^*(\theta);\theta}}(s,a) - V_c^{\pi_{\lambda^*(\theta);\theta}}(s)\Big]Q_c^{\pi_{\lambda^*(\theta);\theta}}(s,a)dads},$$

$$= Q_{\nabla_{\theta}r_{\theta}}^{\pi_{\lambda^*(\theta);\theta}}(s,a) - V_{\nabla_{\theta}r_{\theta}}^{\pi_{\lambda^*(\theta);\theta}}(s) - C_{\pi_{\lambda^*(\theta);\theta}}(Q_c^{\pi_{\lambda^*(\theta);\theta}}(s,a) - V_c^{\pi_{\lambda^*(\theta);\theta}}(s)), \tag{14}$$

where $C_{\pi_{\lambda;\theta}} \triangleq \dfrac{\int_{s \in \mathcal{S}}\int_{a \in \mathcal{A}}\psi^{\pi_{\lambda;\theta}}(s,a)\Big[Q_{\nabla_{\theta}r_{\theta}}^{\pi_{\lambda;\theta}}(s,a)-V_{\nabla_{\theta}r_{\theta}}^{\pi_{\lambda;\theta}}(s)\Big]Q_c^{\pi_{\lambda;\theta}}(s,a)dads}{\int_{s \in \mathcal{S}}\int_{a \in \mathcal{A}}\psi^{\pi_{\lambda;\theta}}(s,a)\Big[Q_c^{\pi_{\lambda;\theta}}(s,a)-V_c^{\pi_{\lambda;\theta}}(s)\Big]Q_c^{\pi_{\lambda;\theta}}(s,a)dads}$, $(b)$ follows Lemma 5, and

$(c)$ follows (11) and (13). Note that we can equivalently reformulate $C_{\pi_{\lambda;\theta}}$ as:

$$C_{\pi_{\lambda;\theta}} = \frac{E_{(s,a)\sim\psi^{\pi_{\lambda;\theta}}(\cdot,\cdot)}[(Q_{\nabla_{\theta}r_{\theta}}^{\pi_{\lambda;\theta}}(s,a) - V_{\nabla_{\theta}r_{\theta}}^{\pi_{\lambda;\theta}}(s))Q_c^{\pi_{\lambda;\theta}}(s,a)]}{E_{(s,a)\sim\psi^{\pi_{\lambda;\theta}}(\cdot,\cdot)}[(Q_c^{\pi_{\lambda;\theta}}(s,a) - V_c^{\pi_{\lambda;\theta}}(s))Q_c^{\pi_{\lambda;\theta}}(s,a)]}. \tag{15}$$

Therefore, we can compute the hyper-gradient as:

$$\frac{dJ_r(\pi_{\lambda^*(\theta);\theta})}{d\theta} \overset{(d)}{=} E_{(s,a)\sim\psi^{\pi_{\lambda^*(\theta);\theta}}}\Big[\frac{d\log\pi_{\lambda^*(\theta);\theta}}{d\theta}Q_r^{\pi_{\lambda^*(\theta);\theta}}(s,a)\Big],$$

$$\overset{(e)}{=} E_{(s,a)\sim\psi^{\pi_{\lambda^*(\theta);\theta}}}\left[\left(Q_{\nabla_\theta r_\theta}^{\pi_{\lambda^*(\theta);\theta}}(s,a) - V_{\nabla_\theta r_\theta}^{\pi_{\lambda^*(\theta);\theta}}(s) - C_{\pi_{\lambda^*(\theta);\theta}}(Q_c^{\pi_{\lambda^*(\theta);\theta}}(s,a) - V_c^{\pi_{\lambda^*(\theta);\theta}}(s))\right)\cdot\right.$$
$$\left. Q_r^{\pi_{\lambda^*(\theta);\theta}}(s,a)\right],$$

where $(d)$ follows the standard result of policy gradient (Sutton & Barto, 2018), and $(e)$ follows (14).

## D.4 PROOF OF LEMMA 3

Since $\pi_{\tilde{n}+1}(a|s) \propto \exp(Q_{\lambda;\theta}^{\pi_{\tilde{n}}}(s,a))$, from Appendix C, we can see that $\pi_{\tilde{n}+1}(a|s) = \frac{\exp(Q_{\lambda;\theta}^{\pi_{\tilde{n}}}(s,a))}{\exp(V_{\lambda;\theta}^{\pi_{\tilde{n}}}(s))}$.

$$|\log\pi_{\tilde{n}+1}(a|s) - \log\pi_{\lambda;\theta}(a|s)| = |Q_{\lambda;\theta}^{\pi_{\tilde{n}}}(s,a) - V_{\lambda;\theta}^{\pi_{\tilde{n}}}(s) - Q_{\lambda;\theta}(s,a) + V_{\lambda;\theta}(s)|,$$
$$\overset{(a)}{=} Q_{\lambda;\theta}(s,a) - Q_{\lambda;\theta}^{\pi_{\tilde{n}}}(s,a) + V_{\lambda;\theta}(s) - V_{\lambda;\theta}^{\pi_{\tilde{n}}}(s),$$
$$\overset{(b)}{\leq} Q_{\lambda;\theta}(s,a) - \mathcal{T}_{\lambda;\theta}^{\text{soft}}(Q_{\lambda;\theta}^{\pi_{\tilde{n}-1}})(s,a) + V_{\lambda;\theta}(s) - \mathcal{T}_{\lambda;\theta}^{\text{soft}}(V_{\lambda;\theta}^{\pi_{\tilde{n}-1}})(s),$$
$$\overset{(c)}{=} \mathcal{T}_{\lambda;\theta}^{\text{soft}}(Q_{\lambda;\theta})(s,a) - \mathcal{T}_{\lambda;\theta}^{\text{soft}}(Q_{\lambda;\theta}^{\pi_{\tilde{n}-1}})(s,a) + \mathcal{T}_{\lambda;\theta}^{\text{soft}}(V_{\lambda;\theta})(s) - \mathcal{T}_{\lambda;\theta}^{\text{soft}}(V_{\lambda;\theta}^{\pi_{\tilde{n}-1}})(s),$$
$$\overset{(d)}{\leq} \gamma\left[Q_{\lambda;\theta}(s,a) - Q_{\lambda;\theta}^{\pi_{\tilde{n}-1}}(s,a) + V_{\lambda;\theta}(s) - V_{\lambda;\theta}^{\pi_{\tilde{n}-1}}(s)\right],$$
$$\leq \gamma^{\tilde{n}+1}\left[Q_{\lambda;\theta}(s,a) - Q_{\lambda;\theta}^{\pi_0}(s,a) + V_{\lambda;\theta}(s) - V_{\lambda;\theta}^{\pi_0}(s)\right],$$

where $(a)$ follows policy improvement theorem (Theorem 4 in (Haarnoja et al., 2017)) (note that $Q_{\lambda;\theta}$ and $V_{\lambda;\theta}$ are the optimal Q/value functions under $(\lambda, \theta)$), $(b)$ follows Lemma 7, $(c)$ follows the fact that the optimal Q/value functions are the fixed points of the contraction operator $\mathcal{T}_{\lambda;\theta}^{\text{soft}}$, and $(d)$ follows Lemma 6.

**Lemma 8.** *For any $\theta$ and $\bar{n} \geq 0$, it holds that $\nabla_{\lambda\lambda}^2 G(\lambda_{\bar{n}};\theta) \succeq \tau_G I$ where $\tau_G$ is a positive constant.*

*Proof.* It has been proved in Subsection D.2 that $\nabla_{\lambda\lambda}^2 G(\lambda;\theta) \succ 0$. To prove that $\nabla_{\lambda\lambda}^2 G(\lambda_{\bar{n}};\theta) \succeq \tau_G I$, we first prove that $\nabla_{\lambda\lambda}^2 G(\lambda;\theta)$ is continuous in $\lambda$ and then prove that the trajectory of $\lambda_{\bar{n}}$ is bounded within a compact set for any $\bar{n} \geq 0$.

From (11), we know that

$$\nabla_{\lambda\lambda}^2 G(\lambda;\theta) = E^{\pi_{\lambda;\theta}}\left[\sum_{t=0}^{\infty}\gamma^t\left(Q_c^{\pi_{\lambda;\theta}}(s,a) - V_c^{\pi_{\lambda;\theta}}(s,a)\right)Q_c^{\pi_{\lambda;\theta}}(s,a)\right].$$

Since $\pi_{\lambda;\theta}, Q_c^{\pi_{\lambda;\theta}}(s,a)$, and $V_c^{\pi_{\lambda;\theta}}(s,a)$ are differentiable to $\lambda$, we know that $\nabla_{\lambda\lambda}^2 G(\lambda;\theta)$ is continuous to $\lambda$. Now, we show that the trajectory of $\lambda_{\bar{n}}$ is bounded within a compact set.

$$\begin{aligned}
||\lambda_{\bar{n}+1} - \lambda^*(\theta)||^2 &= ||\lambda_{\bar{n}} - \alpha_{\bar{n}}g_{\lambda_{\bar{n}};\theta} - \lambda^*(\theta)||^2, \\
&= ||\lambda_{\bar{n}} - \lambda^*(\theta)||^2 + \alpha_{\bar{n}}^2||g_{\lambda_{\bar{n}};\theta}||^2 - \alpha_{\bar{n}}\langle g_{\lambda_{\bar{n}};\theta}, \lambda_{\bar{n}} - \lambda^*(\theta)\rangle, \\
&= ||\lambda_{\bar{n}} - \lambda^*(\theta)||^2 + \alpha_{\bar{n}}^2||g_{\lambda_{\bar{n}};\theta}||^2 - \alpha_{\bar{n}}\langle \nabla_\lambda G(\lambda_{\bar{n}};\theta), \lambda_{\bar{n}} - \lambda^*(\theta)\rangle \\
&\quad - \alpha_{\bar{n}}\langle g_{\lambda_{\bar{n}};\theta} - \nabla_\lambda G(\lambda_{\bar{n}};\theta), \lambda_{\bar{n}} - \lambda^*(\theta)\rangle, \\
&\leq ||\lambda_{\bar{n}} - \lambda^*(\theta)||^2 + \alpha_{\bar{n}}^2||g_{\lambda_{\bar{n}};\theta}||^2 - \alpha_{\bar{n}}[G(\lambda_{\bar{n}};\theta) - G(\lambda^*(\theta);\theta)] \\
&\quad - \alpha_{\bar{n}}\langle g_{\lambda_{\bar{n}};\theta} - \nabla_\lambda G(\lambda_{\bar{n}};\theta), \lambda_{\bar{n}} - \lambda^*(\theta)\rangle, \\
&\leq ||\lambda_{\bar{n}} - \lambda^*(\theta)||^2 + \alpha_{\bar{n}}^2||g_{\lambda_{\bar{n}};\theta}||^2 - \alpha_{\bar{n}}\langle g_{\lambda_{\bar{n}};\theta} - \nabla_\lambda G(\lambda_{\bar{n}};\theta), \lambda_{\bar{n}} - \lambda^*(\theta)\rangle, \\
&\leq ||\lambda_{\bar{n}} - \lambda^*(\theta)||^2 + \alpha_{\bar{n}}^2||g_{\lambda_{\bar{n}};\theta}||^2 + \alpha_{\bar{n}}||g_{\lambda_{\bar{n}};\theta} - \nabla_\lambda G(\lambda_{\bar{n}};\theta)|| \cdot ||\lambda_{\bar{n}} - \lambda^*(\theta)||, \\
&\overset{(a)}{\leq} ||\lambda_{\bar{n}} - \lambda^*(\theta)||^2 + \alpha_{\bar{n}}^2||g_{\lambda_{\bar{n}};\theta}||^2 + \alpha_{\bar{n}}C\gamma^{\bar{n}}||\lambda_{\bar{n}} - \lambda^*(\theta)||, \\
&= ||\lambda_{\bar{n}} - \lambda^*(\theta)||^2 + \alpha_{\bar{n}}^2||g_{\lambda_{\bar{n}};\theta}||^2 + \alpha_{\bar{n}}C\gamma^{\bar{n}}||\lambda_0 - \lambda^*(\theta) - \sum_{i=0}^{\bar{n}-1}\alpha_i g_{\lambda_i;\theta}||,
\end{aligned} \tag{16}$$

$$\leq ||\lambda_{\bar{n}} - \lambda^*(\theta)||^2 + \alpha_{\bar{n}}^2 ||g_{\lambda_{\bar{n}};\theta}||^2 + \alpha_{\bar{n}} C \gamma^{\bar{n}} ||\lambda_0 - \lambda^*(\theta)|| + \alpha_{\bar{n}} C \gamma^{\bar{n}} || \sum_{i=0}^{\bar{n}-1} \alpha_i g_{\lambda_i;\theta}||,$$

$$\overset{(b)}{\leq} ||\lambda_{\bar{n}} - \lambda^*(\theta)||^2 + \alpha_{\bar{n}}^2 (b + \frac{c_{\max}}{1-\gamma})^2 + \alpha_{\bar{n}} C \gamma^{\bar{n}} ||\lambda_0 - \lambda^*(\theta)|| + \alpha_{\bar{n}} C \gamma^{\bar{n}} \sum_{i=0}^{\bar{n}-1} \alpha_i (b + \frac{c_{\max}}{1-\gamma}),$$

(17)

where $(a)$ follows (18), and $(b)$ follows that $||g_{\lambda;\theta}|| = ||b - J_c(\pi_{\lambda;\theta})|| \leq b + \frac{c_{\max}}{1-\gamma}$. Now we show that $\sum_{\bar{n}=1}^{\infty} \alpha_{\bar{n}} \gamma^{\bar{n}} \sum_{i=0}^{\bar{n}-1} \alpha_i$ is bounded. Since $\alpha_i \propto 1/i^{\bar{\eta}}$, we know that $\sum_{i=0}^{\bar{n}-1} \alpha_i = O(\bar{n}^{1-\bar{\eta}})$. Therefore, we know that $\alpha_{\bar{n}} \sum_{i=0}^{\bar{n}-1} \alpha_i = O(\bar{n}^{1-2\bar{\eta}}) \leq \bar{C}$ where $\bar{C}$ is a positive constant. Therefore, $\sum_{\bar{n}=1}^{\infty} \alpha_{\bar{n}} \gamma^{\bar{n}} \sum_{i=0}^{\bar{n}-1} \alpha_i \leq \bar{C} \sum_{\bar{n}=1}^{\infty} \gamma^{\bar{n}}$ is bounded. Now, we sum the both sides of (17) from $\bar{n} = 1$ to $\bar{N} - 1$:

$$\sum_{\bar{n}=0}^{\bar{N}-1} ||\lambda_{\bar{n}+1} - \lambda^*(\theta)||^2,$$

$$\leq \sum_{\bar{n}=0}^{\bar{N}-1} ||\lambda_{\bar{n}} - \lambda^*(\theta)||^2 + \alpha_{\bar{n}}^2 (b + \frac{c_{\max}}{1-\gamma})^2 + \alpha_{\bar{n}} C \gamma^{\bar{n}} ||\lambda_0 - \lambda^*(\theta)|| + \alpha_{\bar{n}} C \gamma^{\bar{n}} \sum_{i=0}^{\bar{n}-1} \alpha_i (b + \frac{c_{\max}}{1-\gamma}),$$

$$\Rightarrow ||\lambda_{\bar{N}} - \lambda^*(\theta)||^2,$$

$$\leq ||\lambda_0 - \lambda^*(\theta)||^2 + \sum_{\bar{n}=0}^{\bar{N}-1} \alpha_{\bar{n}}^2 (b + \frac{c_{\max}}{1-\gamma})^2 + \alpha_{\bar{n}} C \gamma^{\bar{n}} ||\lambda_0 - \lambda^*(\theta)|| + \alpha_{\bar{n}} C \gamma^{\bar{n}} \sum_{i=0}^{\bar{n}-1} \alpha_i (b + \frac{c_{\max}}{1-\gamma}),$$

$$\leq ||\lambda_0 - \lambda^*(\theta)||^2 + \sum_{\bar{n}=0}^{\infty} \alpha_{\bar{n}}^2 (b + \frac{c_{\max}}{1-\gamma})^2 + \alpha_{\bar{n}} C \gamma^{\bar{n}} ||\lambda_0 - \lambda^*(\theta)|| + \alpha_{\bar{n}} C \gamma^{\bar{n}} \sum_{i=0}^{\bar{n}-1} \alpha_i (b + \frac{c_{\max}}{1-\gamma}).$$

Note that $\alpha_{\bar{n}} \propto \frac{1}{(\bar{n}+1)^{\bar{\eta}}}$ and $\bar{\eta} \in (\frac{1}{2}, 1)$, it is obvious that $|\lambda_{\bar{N}} - \lambda^*(\theta)|^2$ is bounded. Therefore, the trajectory of $\lambda_{\bar{n}}$ is bounded for any $\bar{n} \geq 0$. Therefore, we can always find a positive constant $\tau_G$ such that $\nabla_{\lambda\lambda}^2 G(\lambda; \theta) \succeq \tau_G I$. $\qquad\square$

### D.5 PROOF OF LEMMA 4

We first quantify the gradient approximation error $|\nabla_\lambda G(\lambda; \theta) - g_{\lambda;\theta}|$ and then show the convergence of the middle loop.

$$|\nabla_\lambda G(\lambda; \theta) - g_{\lambda;\theta}| = |J_c(\pi_{\lambda;\theta}) - J_c(\hat{\pi}_{\lambda;\theta})|,$$

$$= \left| \int_{s\in\mathcal{S}} \int_{a\in\mathcal{A}} \left[ \psi^{\pi_{\lambda;\theta}}(s,a) - \psi^{\hat{\pi}_{\lambda;\theta}}(s,a) \right] c(s,a) da ds \right|,$$

$$\leq c_{\max} \int_{s\in\mathcal{S}} \int_{a\in\mathcal{A}} \left| \psi^{\pi_{\lambda;\theta}}(s,a) - \psi^{\hat{\pi}_{\lambda;\theta}}(s,a) \right| da ds,$$

$$\overset{(a)}{\leq} c_{\max} C_d \int_{s\in\mathcal{S}} \int_{a\in\mathcal{A}} \left| Q_{\lambda;\theta}(s,a) - Q_{\lambda;\theta}^{\hat{\pi}_{\lambda;\theta}}(s,a) \right| da ds,$$

$$\leq c_{\max} C_d C_{SA} \max_{(s,a)\in\mathcal{S}\times\mathcal{A}} \left\{ |Q_{\lambda;\theta}(s,a) - Q_{\lambda;\theta}^{\hat{\pi}_{\lambda;\theta}}(s,a)| \right\},$$

$$\overset{(b)}{\leq} c_{\max} C_d C_{SA} \gamma^{\tilde{N}_{\bar{n}}} \max_{(s,a)\in\mathcal{S}\times\mathcal{A}} \left\{ |Q_{\lambda;\theta}(s,a) - Q_{\lambda;\theta}^{\pi_0}(s,a)| \right\},$$

$$= C \gamma^{\tilde{N}_{\bar{n}}}$$

(18)

where $(a)$ follows step $(iv)$ of equation (64) in (Zeng et al., 2022) and $C_d$ is a positive constant, $C_{SA}$ can be any positive constant that is larger that the area of $\mathcal{S} \times \mathcal{A}$, $(b)$ follows the proof in Subsection D.4, and $C = c_{\max} C_d C_{SA} \max_{(s,a)\in\mathcal{S}\times\mathcal{A}} \{|Q_{\lambda;\theta}(s,a) - Q_{\lambda;\theta}^{\pi_0}(s,a)|\}$.

Now, we quantify the convergence of the middle loop. From the expression (11) of $\nabla_{\lambda\lambda}^2 G(\lambda; \theta)$, we know that $||\nabla_{\lambda\lambda}^2 G(\lambda; \theta)|| \leq \frac{2c_{\max}^2}{(1-\gamma)^2}$. From (16), we know that:

$$\alpha_{\bar{n}}[G(\lambda_{\bar{n}}; \theta) - G(\lambda^*(\theta); \theta)] \leq ||\lambda_{\bar{n}} - \lambda^*(\theta)||^2 - ||\lambda_{\bar{n}+1} - \lambda^*(\theta)||^2$$

$$+ \alpha_{\bar{n}}^2 ||g_{\lambda_{\bar{n}};\theta}||^2 - \alpha_{\bar{n}} \langle g_{\lambda_{\bar{n}};\theta} - \nabla_\lambda G(\lambda_{\bar{n}};\theta), \lambda_{\bar{n}} - \lambda^*(\theta) \rangle,$$

$$\leq ||\lambda_{\bar{n}} - \lambda^*(\theta)||^2 - ||\lambda_{\bar{n}+1} - \lambda^*(\theta)||^2 + \alpha_{\bar{n}}^2 ||g_{\lambda_{\bar{n}};\theta}||^2 + \alpha_{\bar{n}} ||g_{\lambda_{\bar{n}};\theta} - \nabla_\lambda G(\lambda_{\bar{n}};\theta)|| \cdot ||\lambda_{\bar{n}} - \lambda^*(\theta)||,$$

$$\overset{(c)}{\leq} ||\lambda_{\bar{n}} - \lambda^*(\theta)||^2 - ||\lambda_{\bar{n}+1} - \lambda^*(\theta)||^2 + \alpha_{\bar{n}}^2 ||g_{\lambda_{\bar{n}};\theta}||^2 + \alpha_{\bar{n}} \gamma^{\bar{n}} \tilde{C},$$

$$\overset{(d)}{\leq} ||\lambda_{\bar{n}} - \lambda^*(\theta)||^2 - ||\lambda_{\bar{n}+1} - \lambda^*(\theta)||^2 + \alpha_{\bar{n}}^2 (b + \frac{c_{\max}}{1-\gamma})^2 + \alpha_{\bar{n}} \gamma^{\bar{n}} \tilde{C}, \tag{19}$$

where $(c)$ follows (18) and the fact that $||\lambda_{\bar{n}} - \lambda^*(\theta)||$ is bounded (proved in Lemma 8), $\tilde{C}$ is a positive constant, and $(d)$ follows that $||g_{\lambda;\theta}|| = ||b - J_c(\pi_{\lambda;\theta})|| \leq b + \frac{c_{\max}}{1-\gamma}$. Telescoping (19) from $\bar{n} = 0$ to $\bar{N} - 1$:

$$\sum_{\bar{n}=0}^{\bar{N}-1} \alpha_{\bar{n}} [G(\lambda_{\bar{n}};\theta) - G(\lambda^*(\theta);\theta)],$$

$$\leq ||\lambda_0 - \lambda^*(\theta)||^2 - ||\lambda_{\bar{N}} - \lambda^*(\theta)||^2 + \sum_{\bar{n}=0}^{\bar{N}-1} \alpha_{\bar{n}}^2 (b + \frac{c_{\max}}{1-\gamma})^2 + \sum_{\bar{n}=0}^{\bar{N}-1} \alpha_{\bar{n}} \gamma^{\bar{n}} \tilde{C}.$$

Since $\alpha_{\bar{n}} = \frac{1}{(\bar{n}+1)^{\bar{\eta}}}$ and $\bar{\eta} \in (\frac{1}{2}, 1)$, there is a positive constant $D_{\max}$ such that $\sum_{\bar{n}=0}^{\bar{N}} \alpha_{\bar{n}}^2 (b + \frac{c_{\max}}{1-\gamma})^2 + \sum_{\bar{n}=0}^{\bar{N}} \alpha_{\bar{n}} \gamma^{\bar{n}-1} \tilde{C} \leq D_{\max}$. Therefore, we have that

$$\sum_{\bar{n}=0}^{\bar{N}-1} \frac{1}{\bar{N}^{\bar{\eta}}} [G(\lambda_{\bar{n}};\theta) - G(\lambda^*(\theta);\theta)] \leq \sum_{\bar{n}=0}^{\bar{N}-1} \alpha_{\bar{n}} [G(\lambda_{\bar{n}};\theta) - G(\lambda^*(\theta);\theta)],$$

$$\leq ||\lambda_0 - \lambda^*(\theta)||^2 - ||\lambda_{\bar{N}} - \lambda^*(\theta)||^2 + D_{\max},$$

$$\Rightarrow \frac{1}{\bar{N}} \sum_{\bar{n}=0}^{\bar{N}-1} [G(\lambda_{\bar{n}};\theta) - G(\lambda^*(\theta);\theta)] \leq \frac{1}{N^{1-\bar{\eta}}} \left[ ||\lambda_0 - \lambda^*(\theta)||^2 - ||\lambda_{\bar{N}} - \lambda^*(\theta)||^2 + D_{\max} \right]. \tag{20}$$

Therefore, we have that

$$||\hat{\lambda}(\theta) - \lambda^*(\theta)|| \overset{(e)}{\leq} \frac{2}{\tau_G} [G(\hat{\lambda}(\theta);\theta) - G(\lambda^*(\theta);\theta)] \overset{(f)}{\leq} \frac{2}{\tau_G} [\frac{1}{\bar{N}} \sum_{\bar{n}=0}^{\bar{N}-1} G(\lambda_{\bar{n}};\theta) - G(\lambda^*(\theta);\theta)],$$

$$\overset{(g)}{\leq} O(\frac{1}{\bar{N}^{1-\bar{\eta}}}), \tag{21}$$

where $(e)$ follows the fact that $G(\lambda;\theta)$ is $\tau_G$-strongly convex (Lemma 8), $(f)$ follows Jensen's inequality (note that $\hat{\lambda}(\theta) = \frac{1}{\bar{N}} \sum_{\bar{n}=0}^{\bar{N}-1} \lambda_{\bar{n}}$), and $(g)$ follows (20).

Now, we take a look at the term

$$|\log \pi_{\lambda^*(\theta);\theta}(a|s) - \log \hat{\pi}_{\hat{\lambda}(\theta);\theta}(a|s)|,$$

$$\leq |\log \pi_{\lambda^*(\theta);\theta}(a|s) - \log \pi_{\hat{\lambda}(\theta);\theta}(a|s)| + |\log \pi_{\hat{\lambda}(\theta);\theta}(a|s) - \log \hat{\pi}_{\hat{\lambda}(\theta);\theta}(a|s)|,$$

$$\overset{(h)}{\leq} \frac{2c_{\max}}{1-\gamma} ||\hat{\lambda}(\theta) - \lambda^*(\theta)|| + |\log \pi_{\hat{\lambda}(\theta);\theta}(a|s) - \log \hat{\pi}_{\hat{\lambda}(\theta);\theta}(a|s)|,$$

$$\overset{(i)}{\leq} O(\frac{1}{\bar{N}^{1-\bar{\eta}}} + \gamma^{\bar{N}}), \tag{22}$$

where $(h)$ follows Lemma 5 such that $|\nabla_\lambda \log \pi_{\lambda;\theta}| \leq \frac{2c_{\max}}{1-\gamma}$, and $(i)$ follows (21) and Lemma 3.

**Lemma 9.** *The upper-level loss function $J_r(\pi_{\lambda^*(\theta);\theta})$ is $L$-Lipschitz and $\bar{L}$-smooth where $L$ and $\bar{L}$ are positive constants. Moreover, it holds that $||g_\theta|| \leq L$ and $||\nabla_\theta g_\theta|| \leq \bar{L}$.*

*Proof.* This suffices to show that the norms $||\nabla J_r(\pi_{\lambda^*(\theta);\theta})||$ and $||\nabla^2 J_r(\pi_{\lambda^*(\theta);\theta})||$ are upper bounded by $L$ and $\bar{L}$. From Subsection D.3, we know that

$$\frac{dJ_r(\pi_{\lambda^*(\theta);\theta})}{d\theta},$$

$$= E_{(s,a)\sim\psi^{\pi_{\lambda^*(\theta);\theta}}}\left[\left(Q_{\nabla_\theta r_\theta}^{\pi_{\lambda^*(\theta);\theta}}(s,a) - V_{\nabla_\theta r_\theta}^{\pi_{\lambda^*(\theta);\theta}}(s) - C_{\pi_{\lambda^*(\theta);\theta}}(Q_c^{\pi_{\lambda^*(\theta);\theta}}(s,a) - V_c^{\pi_{\lambda^*(\theta);\theta}}(s))\right)\cdot\right.$$

$$\left. Q_r^{\pi_{\lambda^*(\theta);\theta}}(s,a)\right],$$

where $C_{\pi_\lambda^*(\theta);\theta} = [\nabla_{\lambda\lambda}^2 G(\lambda^*(\theta);\theta)]^{-1}\nabla_{\lambda\theta}^2 G(\lambda^*(\theta);\theta)$. Since $||[\nabla_{\lambda\lambda}^2 G(\lambda^*(\theta);\theta)]^{-1}|| \le \frac{1}{\tau_G}$ (Lemma 8) and $||\nabla_{\lambda\theta}^2 G(\lambda;\theta)|| = ||E_{(s,a)\sim\psi^{\pi_{\lambda;\theta}}(\cdot,\cdot)}[(Q_{\nabla_\theta r_\theta}^{\pi_{\lambda;\theta}}(s,a) - V_{\nabla_\theta r_\theta}^{\pi_{\lambda;\theta}}(s))Q_c^{\pi_{\lambda;\theta}}(s,a)]|| \le \frac{2C_2 c_{\max}}{(1-\gamma)^2}$, we know that $||C_{\pi_\lambda^*(\theta);\theta}|| \le \frac{2C_2 c_{\max}}{(1-\gamma)^2\tau_G}$. Therefore, it holds that

$$||\frac{dJ_r(\pi_{\lambda^*(\theta);\theta})}{d\theta}|| \le \frac{1}{1-\gamma}\cdot\left[\left(\frac{2C_2}{1-\gamma} + \frac{2C_2 c_{\max}}{(1-\gamma)^2\tau_G}\cdot\frac{2c_{\max}}{1-\gamma}\right)\frac{C_1}{1-\gamma}\right] = L. \tag{23}$$

Similarly, we can see that $||g_\theta|| \le L$.

Now, we take a look at the Hessian term $\nabla^2 J_r(\pi_{\lambda^*(\theta);\theta})$. We define $h^\pi(s,a) \triangleq Q_{\nabla_\theta r_\theta}^\pi(s_t,a_t) - V_{\nabla_\theta r_\theta}^\pi(s_t) - C_\pi(Q_c^\pi(s_t,a_t) - V_c^\pi(s_t))]Q_r^\pi(s_t,a_t)$, $H^\pi(s,a) \triangleq E^\pi[\sum_{t=0}^\infty \gamma^t h^\pi(s_t,a_t)|s_0 = s, a_0 = a]$, and $H^\pi(s) \triangleq E^\pi[\sum_{t=0}^\infty \gamma^t h^\pi(s_t,a_t)|s_0 = s]$. We know that $||H^\pi(s,a)|| \le L$ and $||H^\pi(s)|| \le L$. Therefore, we have that

$$\nabla^2 J_r(\pi_{\lambda^*(\theta);\theta}) = \nabla\int_{s_0\in\mathcal{S}} P_0(s_0)\int_{a_0\in\mathcal{A}}\pi_{\lambda^*(\theta);\theta}(a_0|s_0)H^{\pi_{\lambda^*(\theta);\theta}}(s_0,a_0)da_0 ds_0,$$

$$= \int_{s_0\in\mathcal{S}} P_0(s_0)\int_{a_0\in\mathcal{A}}\left[\nabla\pi_{\lambda^*(\theta);\theta}(a_0|s_0)\cdot H^{\pi_{\lambda^*(\theta);\theta}}(s_0,a_0)\right.$$

$$\left. + \pi_{\lambda^*(\theta);\theta}(a_0|s_0)\cdot\nabla H^{\pi_{\lambda^*(\theta);\theta}}(s_0,a_0)da_0 ds_0\right],$$

$$= \int_{s_0\in\mathcal{S}} P_0(s_0)\int_{a_0\in\mathcal{A}}\left[\pi_{\lambda^*(\theta);\theta}(a_0|s_0)\nabla\log\pi_{\lambda^*(\theta);\theta}(a_0|s_0)\cdot H^{\pi_{\lambda^*(\theta);\theta}}(s_0,a_0)\right.$$

$$\left. + \pi_{\lambda^*(\theta);\theta}(a_0|s_0)\cdot\left(\nabla h^{\pi_{\lambda^*(\theta);\theta}}(s_0,a_0) + \gamma\int_{s_1\in\mathcal{S}} P(s_1|s_0,a_0)\nabla H^{\pi_{\lambda^*(\theta);\theta}}(s_1)ds_1\right)da_0 ds_0\right].$$

Keep the expansion, we know that

$$\nabla^2 J_r(\pi_{\lambda^*(\theta);\theta}),$$

$$= \int_{(s,a)\in\mathcal{S}\times\mathcal{A}}\psi^{\pi_{\lambda^*(\theta);\theta}}(s,a)\left[\nabla h^{\pi_{\lambda^*(\theta);\theta}}(s,a) + \nabla\log\pi_{\lambda^*(\theta);\theta}(a|s)\cdot H^{\pi_{\lambda^*(\theta);\theta}}(s,a)\right]dads,$$

$$\tag{24}$$

Since $\nabla\log\pi_{\lambda^*(\theta);\theta}(a|s)$ and $H^{\pi_{\lambda^*(\theta);\theta}}(s,a)$ are both bounded, the only thing left is to bound $||\nabla h^{\pi_{\lambda^*(\theta);\theta}}(s,a)||$. We aim to bound $||\nabla h^{\pi_{\lambda^*(\theta);\theta}}(s,a)||$ by bounding each term in $||h^{\pi_{\lambda^*(\theta);\theta}}(s,a)||$. Note that $\nabla C_{\pi_\lambda^*(\theta);\theta} = \nabla([\nabla_{\lambda\lambda}^2 G(\lambda^*(\theta);\theta)]^{-1}\nabla_{\lambda\theta}^2 G(\lambda^*(\theta);\theta)) = [\nabla_{\lambda\lambda}^2 G(\lambda^*(\theta);\theta)]^{-2}\nabla_{\lambda\theta}^2 G(\lambda^*(\theta);\theta)\frac{d}{d\theta}(\nabla_{\lambda\lambda}^2 G(\lambda^*(\theta);\theta)) + [\nabla_{\lambda\lambda}^2 G(\lambda^*(\theta);\theta)]^{-1}\frac{d}{d\theta}\nabla_{\lambda\theta}^2 G(\lambda^*(\theta);\theta)$. Therefore, it suffices to show that $||\frac{d}{d\theta}Q_{\nabla_\theta r_\theta}^{\pi_{\lambda^*(\theta);\theta}}(s,a)||$ and $||\frac{d}{d\theta}Q_c^{\pi_{\lambda^*(\theta);\theta}}(s,a)||$ are bounded.

$$\frac{d}{d\theta}Q_{\nabla_\theta r_\theta}^{\pi_{\lambda^*(\theta);\theta}}(s,a) = \frac{d}{d\theta}E^{\pi_\lambda^*(\theta);\theta}[\sum_{t=0}^\infty\gamma^t\nabla_\theta r_\theta(s_t|a_t)|s_0 = s, a_0 = a],$$

$$= \frac{d}{d\theta}\int_{s_0\in\mathcal{S}} P_0(s_0)\int_{s_0\in\mathcal{A}}\left[\nabla\pi_{\lambda^*(\theta);\theta}(a_0|s_0)\cdot Q_{\nabla_\theta r_\theta}^{\pi_{\lambda^*(\theta);\theta}}(s_0,a_0)\right.$$

$$\left. + \pi_{\lambda^*(\theta);\theta}(a_0|s_0)\cdot\nabla Q_{\nabla_\theta r_\theta}^{\pi_{\lambda^*(\theta);\theta}}(s_0,a_0)\right]da_0 ds_0,$$

$$= \frac{d}{d\theta}\int_{s_0\in\mathcal{S}} P_0(s_0)\int_{s_0\in\mathcal{A}}\left[\pi_{\lambda^*(\theta);\theta}(a_0|s_0)\nabla\log\pi_{\lambda^*(\theta);\theta}(a_0|s_0)\cdot Q_{\nabla_\theta r_\theta}^{\pi_{\lambda^*(\theta);\theta}}(s_0,a_0)\right.$$

$$\left. + \pi_{\lambda^*(\theta);\theta}(a_0|s_0)\cdot\left(\nabla_{\theta\theta}^2 r_\theta(s_0,a_0) + \gamma\int_{s_1\in\mathcal{S}} P(s_1|s_0,a_0)\nabla V_{\nabla_\theta r_\theta}^{\pi_{\lambda^*(\theta);\theta}}(s_1)ds_1\right)\right]da_0 ds_0.$$

Keep the expansion, we can get that

$$\frac{d}{d\theta} Q_{\nabla_\theta r_\theta}^{\pi_{\lambda^*(\theta);\theta}}(s,a),$$

$$= \int_{(s,a)\in\mathcal{S}\times\mathcal{A}} \psi^{\pi_{\lambda^*(\theta);\theta}} \left[ \nabla_{\theta\theta}^2 r_\theta(s_0,a_0) + \nabla \log \pi_{\lambda^*(\theta);\theta}(a_0|s_0) \cdot Q_{\nabla_\theta r_\theta}^{\pi_{\lambda^*(\theta);\theta}}(s_0,a_0) \right] dads.$$

Therefore, we can see that $||\frac{d}{d\theta} Q_{\nabla_\theta r_\theta}^{\pi_{\lambda^*(\theta);\theta}}(s,a)|| \leq \frac{C_3}{1-\gamma} + \left[ \frac{2c_{\max}}{(1-\gamma)} \cdot \frac{C_2}{\tau_G(1-\gamma)} + \frac{2C_2}{(1-\gamma)} \right] \cdot \frac{C_2}{(1-\gamma)^2}$.

Similarly, we can also see that $\frac{d}{d\theta} Q_c^{\pi_{\lambda^*(\theta);\theta}}(s,a)$ is also bounded. Therefore, we can find a positive constant $\bar{L}$ such that $||\nabla^2 J_r(\pi_{\lambda^*(\theta);\theta})|| \leq \bar{L}$. With the same procedure, we can see that $||\nabla_\theta g_\theta|| \leq \bar{L}$. □

**Lemma 10.** *The hyper-gradient approximation error is upper bounded, i.e.,* $||\frac{d}{d\theta} J_r(\pi_{\lambda^*(\theta);\theta}) - g_\theta|| \leq O(\gamma^{\bar{N}} + \frac{1}{\bar{N}^{1-\bar{\eta}}})$.

*Proof.*

$$||\frac{d}{d\theta} J_r(\pi_{\lambda^*(\theta);\theta}) - g_\theta||,$$

$$\leq \left|\left| \int_{(s,a)\in\mathcal{S}\times\mathcal{A}} \left[ \psi^{\hat{\pi}_{\hat{\lambda}(\theta);\theta}}(s,a) h^{\hat{\pi}_{\hat{\lambda}(\theta);\theta}}(s,a) - \psi^{\pi_{\lambda^*(\theta);\theta}}(s,a) h^{\pi_{\lambda^*(\theta);\theta}}(s,a) \right] dads \right|\right|,$$

$$\overset{(a)}{\leq} (1-\gamma) L \int_{(s,a)\in\mathcal{S}\times\mathcal{A}} ||\psi^{\hat{\pi}_{\hat{\lambda}(\theta);\theta}}(s,a) - \psi^{\pi_{\lambda^*(\theta);\theta}}(s,a)|| dads,$$

$$\leq (1-\gamma) L \int_{(s,a)\in\mathcal{S}\times\mathcal{A}} ||\psi^{\hat{\pi}_{\hat{\lambda}(\theta);\theta}}(s,a) - \psi^{\pi_{\hat{\lambda}(\theta);\theta}}(s,a)|| + ||\psi^{\pi_{\hat{\lambda}(\theta);\theta}}(s,a) - \psi^{\pi_{\lambda^*(\theta);\theta}}(s,a)|| dads,$$

$$\overset{(b)}{\leq} (1-\gamma) C_d C_{SA} \left[ \max\{|Q_{\hat{\lambda}(\theta);\theta}(s,a) - Q_{\hat{\lambda}(\theta);\theta}^{\hat{\pi}_{\hat{\lambda}(\theta);\theta}}(s,a)|\} + \max\{|Q_{\hat{\lambda}(\theta);\theta}(s,a) - Q_{\lambda^*(\theta);\theta}(s,a)|\} \right],$$

$$\overset{(c)}{\leq} (1-\gamma) C_d C_{SA} \left[ \gamma^{\bar{N}} \max\{|Q_{\hat{\lambda}(\theta);\theta}(s,a) - Q_{\hat{\lambda}(\theta);\theta}^{\pi_0}(s,a)|\} \right.$$

$$\left. + \max\{|Q_{\hat{\lambda}(\theta);\theta}(s,a) - Q_{\lambda^*(\theta);\theta}(s,a)|\} \right],$$

$$\overset{(d)}{\leq} (1-\gamma) C_d C_{SA} \left[ \gamma^{\bar{N}} \max\{|Q_{\hat{\lambda}(\theta);\theta}(s,a) - Q_{\hat{\lambda}(\theta);\theta}^{\pi_0}(s,a)|\} + \frac{c_{\max}}{1-\gamma} ||\lambda^*(\theta) - \hat{\lambda}(\theta)|| \right],$$

$$\overset{(e)}{\leq} O(\gamma^{\bar{N}} + \frac{1}{\bar{N}^{1-\bar{\eta}}}),$$

where $(a)$ follows (23), $(b)$ follows step $(iv)$ of equation (64) in (Zeng et al., 2022), $(c)$ follows (18), $(d)$ follows the fact that $||\nabla_\lambda Q_{\lambda;\theta}(s,a)|| = |Q_c^{\pi_{\lambda;\theta}}(s,a)| \leq \frac{c_{\max}}{1-\gamma}$, and $(e)$ follows Lemma 4. □

## D.6 Proof of Theorem 3

We define a function $f(\theta)$ such that $\nabla f(\theta) = g_\theta$ and $\nabla^2 f(\theta) = \nabla_\theta g_\theta$, therefore, we have that

$$f(\theta_{n+1}) \leq f(\theta_n) + \langle \nabla f(\theta_n), \theta_{n+1} - \theta_n \rangle + \frac{\bar{L}}{2} ||\theta_{n+1} - \theta_n||^2,$$

$$= f(\theta_n) - \beta_n ||\nabla f(\theta_n)||^2 + \frac{\bar{L}\beta_n^2}{2} ||\nabla f(\theta_n)||^2,$$

$$\Rightarrow \beta_n ||\nabla f(\theta_n)||^2 \leq f(\theta_n) - f(\theta_{n+1}) + \frac{\bar{L}\beta_n^2}{2} ||\nabla f(\theta_n)||^2. \tag{25}$$

Telescoping (25) from $n = 0$ to $N - 1$, we have that

$$\sum_{n=0}^{N-1} \beta_n ||\nabla f(\theta_n)||^2 \leq f(\theta_0) - f(\theta_N) + \sum_{n=0}^{N-1} \frac{\bar{L}\beta_n^2}{2} ||\nabla f(\theta_n)||^2,$$

$$\Rightarrow \sum_{n=0}^{N-1} \frac{1}{N^\eta} ||\nabla f(\theta_n)||^2 \leq \sum_{n=0}^{N-1} \beta_n ||\nabla f(\theta_n)||^2 \overset{(a)}{\leq} f(\theta_0) - f(\theta_N) + \sum_{n=0}^{\infty} \frac{L^2 \bar{L} \beta_n^2}{2},$$

$$\Rightarrow \frac{1}{N} \sum_{n=0}^{N-1} ||\nabla f(\theta_n)||^2 \leq \frac{1}{N^{1-\eta}} \left( f(\theta_0) - f(\theta_N) + \sum_{n=0}^{\infty} \frac{L^2 \bar{L} \beta_n^2}{2} \right) \overset{(b)}{=} O(\frac{1}{N^{1-\eta}}), \tag{26}$$

where $(a)$ follows Lemma 9, and $(b)$ follows the fact that $\sum_{n=0}^{\infty} \beta_n^2$ is bounded as $\beta_n = \frac{1}{(n+1)^\eta}$ and $\eta \in (\frac{1}{2}, 1)$. Therefore, we have that

$$\frac{1}{N} \sum_{n=0}^{N-1} ||\nabla J_r(\pi_{\lambda^*(\theta);\theta})||^2 \leq \frac{1}{N} \sum_{n=0}^{N-1} \left( ||\nabla f(\theta_n)||^2 + ||\nabla f(\theta_n) - \nabla J_r(\pi_{\lambda^*(\theta);\theta})||^2 \right),$$

$$\overset{(c)}{\leq} O(\frac{1}{N^{1-\eta}} + \gamma^{2\bar{N}} + \frac{1}{\bar{N}^{2-2\bar{\eta}}}),$$

where $(c)$ follows (26) and Lemma 10.

### D.7 PROOF OF THEOREM 4

We first define $F(\theta) \triangleq J_r(\pi_{\lambda^*(\theta);\theta})$. Theorem 10 in (Agarwal et al., 2021) shows that the policy gradient method can achieve global optimality asymptotically under softmax policy parameterization. The constrained soft policy $\pi_{\lambda^*(\theta);\theta} = \lim_{\bar{N} \to \infty} \hat{\pi}_{\hat{\lambda}(\theta);\theta}$ can be regarded as a softmax policy parameterized by $Q_{\lambda^*(\theta);\theta}$ but the decision variable in our case is $\theta$ instead of $Q_{\lambda^*(\theta);\theta}$. However, we can still build on Theorem 10 in (Agarwal et al., 2021) by building connections between $\theta$ and $Q_{\lambda^*(\theta);\theta}$. In specific, in order to use the result of Theorem 10 in (Agarwal et al., 2021), we need to prove (i) $F(\theta_n)$ is monotonically increasing, i.e., $F(\theta_{n+1}) \geq F(\theta_n)$ for any $n \geq 0$; (ii) $\frac{dF(\bar{\theta})}{dQ_{\lambda^*(\theta);\bar{\theta}}} = 0$ if $\frac{dF(\bar{\theta})}{d\theta} = 0$ where $Q_{\lambda^*(\theta);\theta}$ is a vector with the length $|\mathcal{S}| \times |\mathcal{A}|$ whose components are $\{Q_{\lambda^*(\theta);\theta}(s,a)\}_{(s,a) \in \mathcal{S} \times \mathcal{A}}$. Once proving these two, given that $\lim_{N \to \infty} \lim_{\bar{N} \to \infty} \frac{dF(\theta_N)}{d\theta} = 0$ from Theorem 3, we can use Theorem 10 in (Agarwal et al., 2021) to prove Theorem 4.

Now, we first show that $F(\theta_n)$ is monotonically increasing. This is a straightforward result of Theorem 10.15 in (Beck, 2017) if we choose $\beta_n \leq \frac{1}{L}$. Note that Theorem 10 in (Agarwal et al., 2021) requires $\beta_n \leq \frac{(1-\gamma)^3}{8}$, so that we can choose $\beta_n \leq \min\{\frac{1}{L}, \frac{(1-\gamma)^3}{8}\}$. We next show that $\frac{dF(\bar{\theta})}{dQ_{\lambda^*(\bar{\theta});\bar{\theta}}} = 0$ if $\frac{dF(\bar{\theta})}{d\theta} = 0$.

We know that $\frac{dF(\bar{\theta})}{d\theta} = \frac{dF(\bar{\theta})}{dQ_{\lambda^*(\bar{\theta});\bar{\theta}}} \cdot \frac{dQ_{\lambda^*(\bar{\theta});\bar{\theta}}}{d\theta}$, so that it suffices to show that $\frac{dQ_{\lambda^*(\bar{\theta});\bar{\theta}}(s,a)}{d\theta} \neq 0$ for any $(s,a)$ and any $\theta$. Therefore, we have that

$$\frac{dQ_{\lambda^*(\bar{\theta});\bar{\theta}}(s,a)}{d\theta} = \nabla_\lambda Q_{\lambda^*(\bar{\theta});\bar{\theta}}(s,a) \nabla \lambda^*(\bar{\theta}) + \nabla_\theta Q_{\lambda^*(\bar{\theta});\bar{\theta}}(s,a),$$

$$\overset{(a)}{=} Q_{\nabla_\theta r_\theta}^{\pi_{\lambda^*(\bar{\theta});\bar{\theta}}}(s,a) - Q_c^{\pi_{\lambda^*(\bar{\theta});\bar{\theta}}}(s,a) \nabla \lambda^*(\bar{\theta}), \tag{27}$$

where $(a)$ follows (6) and (8). Now, we first prove that the term $Q_{\nabla_\theta r_\theta}^{\pi_{\lambda^*(\bar{\theta});\bar{\theta}}}(s,a) = E^{\pi_{\lambda^*(\theta);\theta}}[\sum_{t=0}^{\infty} \gamma^t \nabla_\theta r_{\bar{\theta}}(s_t, a_t)]$ is nonzero. Define $l(\theta; \lambda, s, a) \triangleq E^{\pi_{\lambda;\theta}}[\sum_{t=0}^{\infty} \gamma^t r_\theta(s_t, a_t) | s_0 = s, a_0 = a]$, and therefore $\nabla_\theta l(\theta; \lambda, s, a) \triangleq E^{\pi_{\lambda;\theta}}[\sum_{t=0}^{\infty} \gamma^t r_\theta(s_t, a_t) | s_0 = s, a_0 = a]$. We use $\psi^\pi(s'|s,a)$ and $\psi^\pi(s', a'|s,a)$ to denote the state and state-action visitation frequency when the initial state-action is $(s,a)$. Now, we take a look at the Hessian term $\nabla_{\theta\theta}^2 l(\theta; \lambda, s, a)$:

$$\nabla_{\theta\theta}^2 l(\theta; \lambda, s, a),$$

$$= \nabla_{\theta\theta}^2 r_\theta(s,a) + \gamma \nabla_\theta \int_{s_1 \in \mathcal{S}} P(s_1|s,a) \int_{a_1 \in \mathcal{A}} \pi_{\lambda;\theta}(a_1|s_1) \nabla_\theta l(\theta; \lambda, s_1, a_1) da_1 ds_1,$$

$$= \nabla_{\theta\theta}^2 r_\theta(s,a) + \gamma \int_{s_1 \in \mathcal{S}} P(s_1|s,a) \int_{a_1 \in \mathcal{A}} \left[ \nabla_\theta \pi_{\lambda;\theta}(a_1|s_1) \cdot \nabla_\theta l(\theta; \lambda, s_1, a_1) \right.$$

$$\left. + \pi_{\lambda;\theta}(a_1|s_1) \cdot \nabla_{\theta\theta}^2 l(\theta; \lambda, s_1, a_1) \right] da_1 ds_1.$$

Keep the expansion and note that $\nabla_\theta l(\theta; \lambda, s, a) = Q^{\pi_{\lambda;\theta}}_{\nabla_\theta r_\theta}(s, a)$, we can get that

$$\nabla^2_{\theta\theta} l(\theta; \lambda, s, a) = \int_{s' \in \mathcal{S}} \int_{a' \in \mathcal{A}} \psi^{\pi_{\lambda;\theta}}(s'|s, a) \nabla_\theta \pi_{\lambda;\theta}(a'|s') \cdot Q^{\pi_{\lambda;\theta}}_{\nabla_\theta r_\theta}(s', a') da' ds',$$

$$= \int_{s' \in \mathcal{S}} \int_{a' \in \mathcal{A}} \psi^{\pi_{\lambda;\theta}}(s'|s, a) \pi_{\lambda;\theta}(a'|s') \nabla_\theta \log \pi_{\lambda;\theta}(a'|s') \cdot Q^{\pi_{\lambda;\theta}}_{\nabla_\theta r_\theta}(s', a') da' ds',$$

$$\overset{(b)}{=} \int_{s' \in \mathcal{S}} \psi^{\pi_{\lambda;\theta}}(s'|s, a) \int_{a' \in \mathcal{A}} \pi_{\lambda;\theta}(a'|s') \left[ Q^{\pi_{\lambda;\theta}}_{\nabla_\theta r_\theta}(s', a') - V^{\pi_{\lambda;\theta}}_{\nabla_\theta r_\theta}(s') \right] Q^{\pi_{\lambda;\theta}}_{\nabla_\theta r_\theta}(s', a') da' ds', \quad (28)$$

where $(b)$ follows Lemma 5. Note that $V^{\pi_{\lambda;\theta}}_{\nabla_\theta r_\theta}(s') = E_{a' \sim \pi_{\lambda;\theta}(\cdot|s')}[Q^{\pi_{\lambda;\theta}}_{\nabla_\theta r_\theta}(s', a')]$. If we use the random variable $Y_{s'a'}$ to denote $Q^{\pi_{\lambda;\theta}}_{\nabla_\theta r_\theta}(s', a')$, then its expectation is $E_{a' \sim \pi_{\lambda;\theta}(\cdot|s')}(Y_{s'a'}) = V^{\pi_{\lambda;\theta}}_{\nabla_\theta r_\theta}(s')$. We know that the variance $Var(Y_{s'a'}) = E[Y_{s'a'}[Y_{s'a'} - E(Y_{s'a'})]]$. Therefore, we can see that the equation (28) is actually a variance:

$$\nabla^2_{\theta\theta} l(\theta; \lambda, s, a) = \int_{s' \in \mathcal{S}} \psi^{\pi_{\lambda;\theta}}(s'|s, a) Var(Y_{s'a'}) ds' \succeq 0.$$

Therefore, the function $l(\theta; \lambda, s, a)$ is convex in $\theta$ for any $(\lambda, s, a)$. If $\nabla_\theta l(\bar{\theta}; \lambda, s, a) = 0$, this means that $l(\bar{\theta}; \lambda, s, a)$ achieves its optimum. However, $l(\bar{\theta}; \lambda, s, a)$ does not have an optimum, i.e., $l(\theta; \lambda, s, a)$ can be infinitely large or infinitely small because $r_\theta(s, a)$ can be any arbitrarily large value. This is a contradiction, therefore, $l(\bar{\theta}; \lambda, s, a) \neq 0$ for any $(\lambda, s, a)$. Then, $l(\bar{\theta}; \lambda^*(\bar{\theta}), s, a) = Q^{\pi_{\lambda^*(\bar{\theta});\bar{\theta}}}_{\nabla_\theta r_\theta}(s, a) \neq 0$.

Recall from (27) that $\frac{dQ_{\lambda^*(\bar{\theta});\bar{\theta}}(s,a)}{d\theta} = Q^{\pi_{\lambda^*(\bar{\theta});\bar{\theta}}}_{\nabla_\theta r_\theta}(s, a) - Q^{\pi_{\lambda^*(\bar{\theta});\bar{\theta}}}_c(s, a) \nabla \lambda^*(\bar{\theta})$. For any $(s, a) \notin \mathcal{C}$, $Q^{\pi_{\lambda^*(\bar{\theta});\bar{\theta}}}_c(s, a) = 0$ because the policy $\pi_{\lambda^*(\bar{\theta});\bar{\theta}}$ satisfies the constraint of the lower-level problem in (1), i.e., avoiding the set $\mathcal{C}$. Therefore, $\frac{dQ_{\lambda^*(\bar{\theta});\bar{\theta}}(s,a)}{d\theta} = Q^{\pi_{\lambda^*(\bar{\theta});\bar{\theta}}}_{\nabla_\theta r_\theta}(s, a) \neq 0$. For any $(s, a) \in \mathcal{C}$, we know that $Q^{\pi_{\lambda^*(\bar{\theta});\bar{\theta}}}_c(s, a) = c(s, a)$ because the policy $\pi_{\lambda^*(\bar{\theta});\bar{\theta}}$ avoids the set $\mathcal{C}$ unless its starting state-action pair is in $\mathcal{C}$. Therefore, we can also design $c(s, a)$ such that $Q^{\pi_{\lambda^*(\bar{\theta});\bar{\theta}}}_{\nabla_\theta r_\theta}(s, a) - c(s, a) \nabla \lambda^*(\bar{\theta}) \neq 0$ for $(s, a) \in \mathcal{C}$. Therefore, we can ensure that $\frac{dQ_{\lambda^*(\bar{\theta});\bar{\theta}}(s,a)}{d\theta} \neq 0$ for any $(s, a) \in \mathcal{S} \times \mathcal{A}$. Therefore, $\frac{dF(\bar{\theta})}{dQ_{\lambda^*(\bar{\theta});\bar{\theta}}} = 0$ if $\frac{dF(\bar{\theta})}{d\theta} = 0$.

## E    RELATED WORKS

**XRL methods that has the potential to be used to improve RL performance**. There are some XRL methods that have the potential to improve the RL performance even if they do not mention that they can improve the RL performance. Value-max (Amir & Amir, 2018; Huang et al., 2018) use the value function $V(s)$ to identify the states with highest value as critical points. We can perturb the actions on these critical states to improve the RL performance.

**Constrained reinforcement learning (CRL)**. The lower-level problem in (1) is a CRL problem. The current works on CRL have two major categories: primal-dual approach and primal approach. The primal-dual approach (Achiam et al., 2017; Tessler et al., 2018; Stooke et al., 2020) converts the CRL problem into an unconstrained optimization problem by using the dual method. Our approach can be categorized as a primal-dual approach. The primal approach (Liu et al., 2020; Chow et al., 2018; Xu et al., 2021) enforce constraints via various designs of the objective function or the update process without an introduction of dual variables. However, these previous methods on CRL may not be suitable to the context of constrained bi-level optimization because they cannot guarantee that the upper-level problem in (1) is smooth after the lower-level problem is solved. The non-smoothness of the upper-level problem can make the constrained bi-level optimization problem difficult to solve. In contrast, our approach ensures the smoothness of the upper-level problem in (1) because we derive an analytical solution of the constrained soft policy and this policy is smooth w.r.t. $\theta$.

## F    EXPERIMENT DETAILS

The code was running on a laptop whose CPU is Intel Core i9 12900k and GPU is NVIDIA RTX 3080. The operating system is Windows 10. We use a neural network to parameterize the learned

reward function. The neural network has two hidden layers where each hidden layer has 64 neurons. The activation functions are respectively ReLU and Tanh.

**The delayed MuJoCo environments**. The delayed Mujoco environments are widely used in RL improvement literature (Zheng et al., 2018; Memarian et al., 2021; Oh et al., 2018) where the reward is accumulated by 20 time steps and only provided at the end. For example, for an episode with length 100 (i.e., $0, \cdots, 99$), only the time steps 19, 39, 59, 79, 99 receive nonzero reward while all the other steps receive zero reward. The time step 19 receives the reward that is accumulated from time 0 to time 19. We can see that the delayed MuJoCo tasks have sparse reward.

We also conduct experiments on the original MuJoCo environments, which are widely used in RL literature (Xu & Zhu, 2023b; 2024) We will explore some other tasks like motion planning (Xu & Zhu, 2022; Qiao et al., 2023; 2024) and finance (Yu et al., 2025; Ke et al., 2024).

## F.1    EXPERIMENT RESULTS ON DENSE REWARD

To show that our method can also improve the performance on dense reward scenarios, we include the experiment results on the original MuJoCo environment (where the reward is dense) below:

Table 3: Experiment results (original MuJoCo environment with dense reward).

|  | SAC | UITLITY | RICE | SIL | LIR |
|---|---|---|---|---|---|
| HalfCheetah | $686.40 \pm 41.24$ | $824.42 \pm 42.18$ | $701.44 \pm 45.83$ | $716.62 \pm 39.17$ | $718.25 \pm 42.10$ |
| Hopper | $238.14 \pm 29.94$ | $348.16 \pm 26.32$ | $242.99 \pm 19.10$ | $264.86 \pm 27.11$ | $277.58 \pm 22.69$ |
| Walker2d | $182.21 \pm 24.14$ | $269.14 \pm 25.08$ | $189.19 \pm 39.11$ | $197.63 \pm 21.39$ | $214.16 \pm 21.24$ |
| Ant | $299.79 \pm 26.53$ | $421.63 \pm 25.16$ | $322.26 \pm 34.15$ | $340.14 \pm 26.53$ | $351.14 \pm 28.92$ |

Table 3 shows the final results on the original MuJoCo environment (dense reward). We can observe that UTILITY achieves the highest reward and largely improves SAC.

## F.2    ABLATION STUDY

Since our method has two components to improve the performance: the shaping reward and the constrained formulated by the "misleading" state-action pairs. Here, we separately study the effect of the learned shaping reward and the constraint. In specific, we test the performance of the shaping only method and the constraint only method, and provide the results below:

Table 4: Ablation study for dense reward.

|  | SAC | UITLITY | shaping only | constraint only |
|---|---|---|---|---|
| HalfCheetah | $686.40 \pm 51.24$ | $824.42 \pm 42.18$ | $764.25 \pm 48.11$ | $701.19 \pm 47.43$ |
| Hopper | $238.14 \pm 29.94$ | $348.16 \pm 26.32$ | $311.78 \pm 34.24$ | $268.15 \pm 42.66$ |
| Walker2d | $182.21 \pm 24.14$ | $269.14 \pm 25.08$ | $242.18 \pm 29.62$ | $196.77 \pm 22.19$ |
| Ant | $299.79 \pm 26.53$ | $421.63 \pm 25.16$ | $396.05 \pm 29.18$ | $317.12 \pm 34.59$ |

Table 4 shows that both the learned shaping reward and the constraint can improve the performance, and the shaping reward has a larger impact. Moreover, even if we only use the shaping reward, the performance is better than LIR. This is because our shaping reward uses the domain knowledge formulated by the high-level explanation. Even if we only use the constraint, the performance is better than RICE. The reason is that the "misleading" state-action pairs we find are the points that lead to the failure, and thus avoiding these state-action pairs can improve performance. In contrast, RICE finds the states that are most influential cumulative reward, however, these states may not be the states that lead the RL agent to be non-optimal.

## F.3    FIDELITY OF THE GENERATED TWO-LEVEL EXPLANATION

The fidelity means the correctness of the two-level explanation (Guo et al., 2021; Cheng et al., 2023). Since the two-level explanation is to explain why the RL agent (i.e., SAC) is not optimal, one way to

validate the fidelity of the explanation is to see whether the performance improves after we improve from the explanations. From the last two columns in Table 4, we can see that both the high-level and low-level explanations are the correct explanations because both the shaping only method and the constraint only method improve the performance. Moreover, the shaping only method (the fourth column in Table 4) has a higher cumulative reward than LIR (the last column in Table 3), and the constraint only method (the last column in Table 4) has a higher cumulative reward than RICE (the fourth column in Table 3). This shows the high fidelity of our two-level explanation.

Table 5: Fidelity comparison for sparse reward.

|  | SAC | shaping only (ours) | shaping without $r - \hat{r}$ | constraint only (ours) | RICE+constraint |
|---|---|---|---|---|---|
| Delayed HalfCheetah | $383.45 \pm 45.50$ | $695.63 \pm 33.66$ | $611.08 \pm 39.44$ | $422.15 \pm 22.86$ | $369.14 \pm 19.40$ |
| Delayed Hopper | $192.90 \pm 27.18$ | $289.10 \pm 18.41$ | $255.18 \pm 16.57$ | $210.12 \pm 15.77$ | $181.45 \pm 12.11$ |
| Delayed Walker2d | $134.91 \pm 20.80$ | $211.37 \pm 18.64$ | $191.15 \pm 11.26$ | $175.66 \pm 15.27$ | $140.26 \pm 11.53$ |
| Delayed Ant | $68.11 \pm 12.52$ | $88.18 \pm 8.66$ | $77.11 \pm 5.52$ | $75.16 \pm 6.58$ | $63.11 \pm 4.28$ |

Table 6: Fidelity comparison for dense reward.

|  | SAC | shaping only (ours) | shaping without $r - \hat{r}$ | constraint only (ours) | RICE+constraint |
|---|---|---|---|---|---|
| HalfCheetah | $686.40 \pm 51.24$ | $764.25 \pm 48.11$ | $710.99 \pm 35.16$ | $701.19 \pm 47.43$ | $672.15 \pm 25.16$ |
| Hopper | $238.14 \pm 29.94$ | $311.78 \pm 34.24$ | $279.12 \pm 27.44$ | $268.15 \pm 42.66$ | $211.03 \pm 21.20$ |
| Walker2d | $182.21 \pm 24.14$ | $242.18 \pm 29.62$ | $213.15 \pm 18.94$ | $196.77 \pm 22.19$ | $177.19 \pm 15.33$ |
| Ant | $299.79 \pm 26.53$ | $396.05 \pm 29.18$ | $359.14 \pm 22.85$ | $317.12 \pm 34.59$ | $288.19 \pm 18.02$ |

Tables 5 and 6 show that both the high-level and low-level explanations of our method have higher fidelity. The method "RICE+constraint" has even worse performance than SAC because the critical states influential to the cumulative reward may be the states that lead to high cumulative reward, and thus constraining them may even make the performance worse. However, even if we do not constrain these states but use the fine-tune method as in (Cheng et al., 2023) instead, our constraint-only method (the last column in Table 4) still outperforms RICE (the fourth column in Table 3). For the high-level explanation, we can see that our shaping only method achieves higher cumulative reward than the method "shaping without $r - \hat{r}$". This shows the high fidelity of our high-level explanation.

### F.4 HOW TO ACCELERATE THE TRIPLE-LOOP ALGORITHM

The total iterations of UTILITY is $N \times \bar{N} \times \tilde{N}$ where $N$ is the iteration number of the outer loop, $\bar{N}$ is the iteration number of the middle loop, and $\tilde{N}$ is the iteration number of the inner loop. While the triple-loop structure looks computationally expensive, in practice, we can significantly accelerate the algorithm using warm start in the inner loop and middle loop. Take the inner loop as an example, given the shaping parameter $\theta$ and the current dual parameter $\lambda_{\bar{n}}$, we need to compute the corresponding constrained soft Bellman policy $\pi_{\lambda_{\bar{n}};\theta}$ in the inner loop. Instead of starting from a random policy initialization, we use the policy $\hat{\pi}_{\lambda_{\bar{n}-1};\theta}$ learned in last inner loop as the initialization where $\hat{\pi}_{\lambda_{\bar{n}-1};\theta}$ is an approximation of $\pi_{\lambda_{\bar{n}-1};\theta}$. The intuition behind this is that since $\lambda_{\bar{n}}$ and $\lambda_{\bar{n}-1}$ are close (only different by one-step gradient descent), it is expected that $\pi_{\lambda_{\bar{n}-1};\theta}$ and $\pi_{\lambda_{\bar{n}};\theta}$ are close. Therefore, using $\hat{\pi}_{\lambda_{\bar{n}-1};\theta}$ as the initialization makes it easier to approach $\pi_{\lambda_{\bar{n}};\theta}$. Therefore, the warm start trick enables us to use fewer iterations for the inner loop, i.e., the iteration number $\tilde{N}$ reduces. We use the similar warm start trick for the middle loop to reduce the iteration number $\bar{N}$.

## G POTENTIAL SOCIETAL IMPACT

This paper has positive impact which is to improve the performance of RL agents. However, the paper also has potential negative impacts. Since the two-level explanation identifies the mistakes made by the RL agents. A malicious entity may use these weaknesses or mistakes to launch attack to the RL agents. To alleviate this issue, one solution is to keep the demonstration of the RL agent private, so that the malicious entity cannot get access to the demonstration and thus cannot find the weakness.

# H    LIMITATIONS

One limitation of the method is that it requires to interact with the environment. Therefore, one future work is to extend this method to the offline RL setting.

# I    EVOLUTION FRO FIGURE 1C TO FIGURE 1D

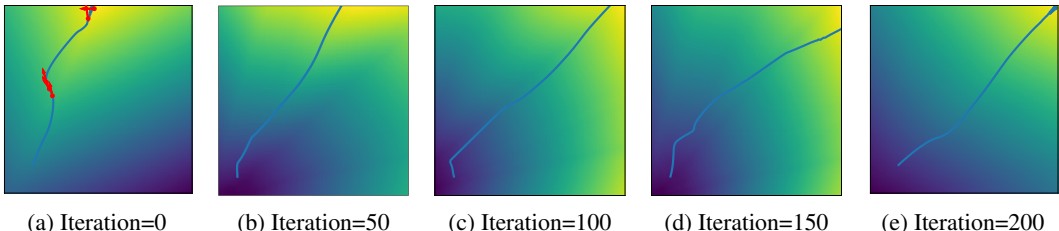

(a) Iteration=0    (b) Iteration=50    (c) Iteration=100    (d) Iteration=150    (e) Iteration=200

Figure 3: Evolution from Figure 1c to Figure 1d.

For this scenario, we run UTILITY for 200 iterations. Figure 3 show the evolution from Figure 1c (iteration=0) to Figure 1d (iteration=200). From the evolution, we can see that the learned reward becomes closer and closer to the ground truth reward and the learned policy becomes more and more optimal.

