# OpenReview forum: "UTILITY: Utilizing Explainable Reinforcement Learning to Improve Reinforcement Learning"
_ICLR.cc/2025/Conference — ICLR 2025 Poster_

### Official Review · Reviewer_iSK2 · 2024-10-24

**Soundness:** 2
**Presentation:** 2
**Contribution:** 3
**Rating:** 8
**Confidence:** 3

**Summary:**

The paper proposes to improve an RL policy by using IRL to learn a reward for which the policy is optimal, and then use this IRL reward to shape the original reward by adding two terms: (1) an additional reward proportional to the difference between IRL reward and true reward, and (2) a constraint that the policy does not visit significantly suboptimal states under the Q function of IRL reward. This results in a constrained RL problem, which the authors propose an iterative solution for. The algorithm is shown to improve over SAC and several other baselines that shape SAC on continuous control tasks.

**Strengths:**

Strengths:
1. The idea of using IRL as a signal for RL is interesting.
2. Figure 1 is very clear and helps understand the method.
3. The experiments and ablation studies seem rather extensive, although I am not familiar enough with the relevant baselines to say if anything important is missing.
4. The theoretical analysis of convergence for the iterative solution of the optimization problem in Section 4.2.3 is nice.

**Weaknesses:**

Post rebuttal: after an interesting discussion with the authors, there is a way to formally establish that Eq. (1) is a sound idea for a kind of "policy iteration" step that is guaranteed to improve the initial policy. This makes the paper an interesting contribution.
The authors promised to make significant chanegs in the writing that will improve the paper's presentation and suitability for the general RL community.
Based on this, I recommend to accept the paper and changed my score.

Weaknesses:
1. In the problem formulation, the authors formulate the constrained optimization problem in Eq. (1) as the problem that the agent needs tosolve. My biggest concern is that while the authors spend considerable efforts on showing that their solution of Eq. (1) is sound, the derivations leading to Eq. (1) seem very ad hoc to me, and I cannot understand why this method would work in general RL problems (or, for what problems should it work?)
2. The writing and presentation could be improved. For example, the paper talks a lot about explainability, but I did not find explainability to be relevant here so much. At some point, the “explainable RL” idea is replaced by standard IRL. I think the authors should tone down the XRL parts and focus instead on using IRL as a signal in RL.

Details:

1. The derivations leading to Eq. (1), the main objective of the method, are not clear to me. I collect here several unclear passages.
    1. Line 185: “At a low level, the RL agent is not optimal meaning that it visits some critical points that lead to the non-optimality.” - I don’t understand this. It could be that the agent is slightly suboptimal *on all* states. This statement seems very ad hoc. Why should it be suboptimal only on some critical points?
    2. Line 199: \pi* and \pi_A are distributions. What does \pi_A - \pi* mean exactly? Why is it a metric?
    3. Line 209: isn’t this a well known result? I.e., a policy is optimal iff it satisfies the Bellman equation.
    4. Line 215: I’m not sure why Top K “misleading” states make sense. Because changing an action in a state affects the future states the agent will visit, I don’t see how ordering by misleading level makes any sense. For example, consider an agent that at the first step can walk either right or left. If it chooses right, it reaches the goal after some time and obtains reward. If it chooses left, after some time it can either choose to reach the goal, or die and get a huge negative reward. Consider a policy that both takes left in the beginning, and then chooses to die and get the negative reward. Because of discounting, the most misleading state here would be the last state with the negative reward. But if the agent chose right in the beginning, it would not have even reached this state at all, and that would be the state that we want to fix.
    5. Line 240: I don’t understand what the difference r - \hat{r}  really means. Is there any formal result that ties between  \hat{r}  that is the max likelihood IRL solution to the true reward in the system? Is there any formal result that makes this ad hoc choice make sense?
    6. Line 247-257: This whole paragraph seems very ad hoc. Why should we discourage the misleading state-actions? Maybe under a different policy, they are actually necessary for optimal behavior? E.g., consider an agent that *must* cross some “bridge” to get to the goal. Now, because the agent is not optimal, state actions on the bridge are misleading. But if we restrict the agent from visiting them, it will *never* get to the goal! The causal entropy term is also not well motivated - why is exploration crucial here?

Summary (and explanation for score): I think this paper really needs more work in explaining *why* the approach makes sense. A theoretical result that ties the max-likelihood IRL that the authors build on to Eq. (1) and some optimal policy in the MDP, for example, would go a long way in establishing the soundness of this work, and what MDPs it should work for (the examples I mentioned above hint that for some MDPs we should not expect this approach to work well).

**Questions:**

see above.

---

> ### Author Response · Authors · 2024-11-20
>
> Thank you very much for your thoughtful consideration of our paper and the helpful comments and suggestions. We would hope that our answers will alleviate your concerns. We address your comments below. We first answer the six details of Weakness 1 and then answer Weakness 2.
>
> **Weakness 1.1**: "Line 185: “At a low level, the RL agent is not optimal meaning that it visits some critical points that lead to the non-optimality.” - I don’t understand this. It could be that the agent is slightly suboptimal on all states. This statement seems very ad hoc. Why should it be suboptimal only on some critical points?"
> **Answer**: Thanks for mentioning the critical points. The critical points are the most important points that \emph{lead} to non-optimality, however, this does not mean that the original policy $\pi\_A$ is only sub-optimal on the critical points. The original policy $\pi\_A$ can be suboptimal on all the states, and we pick the top K most suboptimal ones as critical points. In very rare cases where the policy $\pi\_A$ is equally suboptimal on every state, we randomly pick K critical points. In conclusion, the critical points are the most suboptimal points, and thus are a subset of all the sub-optimal points. As long as the original policy $\pi\_A$ is suboptimal, it must be suboptimal on certain states (including the case of all states) and we pick the most suboptimal ones as critical points. Therefore, we believe that this statement is not very ad hoc.
>
> **Weakness 1.2**: "Line 199: $\pi^{\ast}$ and $\pi\_A$ are distributions. What does $\pi^{\ast}-\pi\_A$ mean exactly? Why is it a metric?"
> **Answer**: The metric in line 199 is $\pi^{\ast}(a|s)-\pi\_A(a|s)$. While $\pi^{\ast}$ and $\pi\_A$ are distributions over $\mathcal{S}\times\mathcal{A}$, $\pi^{\ast}(a|s)$ and $\pi\_A(a|s)$ are specific values (i.e., the probabilities of $\pi^{\ast}$ and $\pi\_A$ choosing the action $a$ at the state $s$). This metric is the probability difference between the optimal policy $\pi^{\ast}$ and the original policy $\pi\_A$ at the state-action pair $(s,a)$. This metric can test whether $(s,a)$ is misleading. In specific, the value of this metric is between -1 and 1. If $\pi^{\ast}(a|s)-\pi\_A(a|s)$ is close to -1, this means that $\pi^{\ast}(a|s)$ has low probability of choosing the action $a$ at the state $s$ while $\pi\_A(a|s)$ has high probability of choosing $a$ at $s$. Therefore, this state-action pair $(s,a)$ is identified as misleading because $\pi\_A$ chooses the state-action pair $(s,a)$ that the optimal policy $\pi^{\ast}$ is not likely to choose. The closer the metric $\pi^{\ast}(a|s)-\pi\_A(a|s)$ is to -1, the more misleading the state-action pair $(s,a)$ is.
>
> **Weakness 1.3**: "Line 209: isn’t this a well known result? I.e., a policy is optimal iff it satisfies the Bellman equation."
> **Answer**: Line 209 is different from Bellman optimality equation. The Bellman optimality equation is $Q\_r^{\pi}(s,a)=r(s,a)+\gamma E\_{s'\sim P(\cdot|s,a)}[\max\_{a'}Q\_r^{\pi}(s',a')]$ for any $(s,a)$. However, what line 209 proves is the case where $Q\_r^{\pi\_A}(s,a)=\max\_{a'}Q\_r^{\pi\_A}(s,a')$ for $(s,a)$ such that $\pi\_A(a|s)>0$. This is not the Bellman optimality equation. In our proof in Appendix B.1, we prove that this condition in line 209 can lead to Bellman optimality equation.

---

> > ### Author Response · Authors · 2024-11-20
> >
> > **Weakness 1.4**: "Line 215: I’m not sure why Top K "misleading” states make sense. Because changing an action in a state affects the future states the agent will visit, I don’t see how ordering by misleading level makes any sense. For example, consider an agent that at the first step can walk either right or left. If it chooses right, it reaches the goal after some time and obtains reward. If it chooses left, after some time it can either choose to reach the goal, or die and get a huge negative reward. Consider a policy that both takes left in the beginning, and then chooses to die and get the negative reward. Because of discounting, the most misleading state here would be the last state with the negative reward. But if the agent chose right in the beginning, it would not have even reached this state at all, and that would be the state that we want to fix."
> > **Answer**: Thanks for mentioning identifying misleading state-action pairs. At the beginning, we would like to review how the misleading state-action pairs are identified. Since we treat the RL agent as a black box (lines 159-164), we can only observe a set $D$ of trajectories generated by the agent's policy $\pi\_A$. We aim to identify the misleading state-action pairs in the set $D$. It is possible that there are some misleading state-action pairs of $\pi\_A$ not included in the set $D$. In this case, it is impossible to identify those misleading state-action pairs because we have zero information about those misleading state-action pairs. In the example you mention, if the set $D$ does not include the last state with negative reward, it is not possible to identify this misleading state. However, this is not the limitation of our method. This is the limitation of the set $D$, i.e., the black box (RL agent) does not provide enough data. If the set $D$ includes the last state with negative reward, our method can identify this. In specific, in this case, the most misleading state-action pair is the state-action pair that lead to the last state with negative reward. We demote this misleading state-action pair by $(\bar{s},\bar{a})$. Since $(\bar{s},\bar{a})$ leads to a state with negative reward, $Q\_r^{\pi\_A}(\bar{s},\bar{a})$ can be very small and even negative. By using the metric "misleading level" $l(s,a)=\max\_{a'}Q\_r^{\pi\_A}(s,a')-Q\_r^{\pi\_A}(s,a)$, we can see that $l(\bar{s},\bar{a})>0$. Moreover, $l(\bar{s},\bar{a})$ can be large because $Q\_r^{\pi\_A}(\bar{s},\bar{a})$ is very small. Therefore, the metric can identify the misleading state-action pair $(\bar{s},\bar{a})$.
> >
> > **Weakness 1.5**: "Line 240: I don’t understand what the difference $r-\hat{r}$ really means. Is there any formal result that ties between $\hat{r}$ that is the max likelihood IRL solution to the true reward in the system? Is there any formal result that makes this ad hoc choice make sense?"
> > **Answer**: Thanks for mentioning the difference $r-\hat{r}$. The reward function $r$ is the ground truth reward function. The RL agent aims to learn a policy to optimize $r$, but only obtains a suboptimal policy $\pi\_A$. Maximum likelihood IRL can learn a reward function $\hat{r}$, to which $\pi\_A$ is optimal. In other words, $\hat{r}$ is the reward function that the RL agent actually optimizes. Therefore, we can treat the RL agent as a biased black box such that if the input is the ground truth reward function $r$, the output is a policy $\pi\_A$ that is optimal to another reward function $\hat{r}$. We want to change the input of this biased black box such that the output is a policy that is optimal to the ground truth reward function $r$. More formally, we aim to learn a shaping reward $r\_{\theta}$ as the input such that the output is optimal to $r$. A straightforward way is to directly learn a neural reward $r\_{\theta}: \mathcal{S}\times\mathcal{A}\rightarrow R$. However, in practice, this is difficult to learn because the neural reward needs to search over a very large function space. We find empirically that if we use the shaping reward in the form that $r\_{\theta}(s,a)=r(s,a)+\theta(s,a)(r(s,a)-\hat{r}(s,a))$, the learning results will be much better (please see the third and fourth columns in Table 3 where we compare shaping with and without $r-\hat{r}$). Therefore, the reason that we uses $r-\hat{r}$ is from an empirical observation that using $r-\hat{r}$ will lead to a better performance. Intuitively, this is because that $r-\hat{r}$ serves as a correction term. If $r(s,a)-\hat{r}(s,a)$ is large, it means that the ground truth shows that $(s,a)$ is important but the RL agent thinks that $(s,a)$ is not important and thus will not be attracted. Therefore, this correction term in $r\_{\theta}(s,a)=r(s,a)+\theta(s,a)(r(s,a)-\hat{r}(s,a))$ will assign even larger reward at $(s,a)$ to augment the signal of attracting the agent.

---

> ### Author Response · Authors · 2024-11-20
>
> **Weakness 1.6**: "Line 247-257: This whole paragraph seems very ad hoc. Why should we discourage the misleading state-actions? Maybe under a different policy, they are actually necessary for optimal behavior? E.g., consider an agent that must cross some “bridge” to get to the goal. Now, because the agent is not optimal, state actions on the bridge are misleading. But if we restrict the agent from visiting them, it will never get to the goal! The causal entropy term is also not well motivated - why is exploration crucial here?"
> **Answer**: Thanks for mentioning the misleading state-action pairs and entropy term. We discuss these two points respectively as follows.
> For the misleading state-action pairs, we agree that there could be false positive, i.e., some misleading state-action pairs should not be avoided. To mitigate this issue, we can impose soft constraint on the misleading state-action pairs, i.e., $b>0$. In this case, we can still visit some misleading state-action pairs to achieve high cumulative reward. We choose $b=0$, $b=5$ and $b=20$ for comparisons. The values in the table below are the cumulative reward improvement.
>
> |         | Delayed HalfCheetah | Delayed Hopper | Delayed Walker | Delayed Ant |
> |---------|---------------------|----------------|----------------|-------------|
> | $b=0$   | $38.70$             | $17.22$        | $40.75$        | $7.05$      |
> | $b=5$   | $38.14$             | $21.63$        | $42.19$        | $7.76$      |
> | $b=20$  | $12.66$             | $5.29$         | $3.29$         | $1.48$      |
>
> The results above show that when $b=5$, the cumulative reward improvement will become larger. This shows that we have false positive misleading state-action pairs and a small positive $b$ can reduce the negative impact of those false positive misleading state-action pairs. When $b=20$, the cumulative improvement becomes much smaller because at this time, many misleading state-action pairs that should be avoided are not avoided. The results above indicate that while our misleading state-action pairs have some false positives, the majority is accurate.
> For the entropy term, RL has a long history about the exploration and exploitation balance. In specific, if we always choose greedy policy under the current policy evaluation, the RL algorithm may only explore a small portion of the state-action space and thus miss the real optimal action. At the beginning, people usually use $\epsilon$-greedy policy to encourage exploration. Recently, soft Q-learning and soft actor-critic propose to add an entropy term to the RL objective to encourage exploration. In our case, the exploration is even more important because we already know that the previous policy $\pi\_A$ is suboptimal. Therefore, exploration is encouraged to explore real optimal actions.
>
> **Weakness 2**: "The writing and presentation could be improved. For example, the paper talks a lot about explainability, but I did not find explainability to be relevant here so much. At some point, the “explainable RL” idea is replaced by standard IRL. I think the authors should tone down the XRL parts and focus instead on using IRL as a signal in RL."
> **Answer**: Thanks for mentioning the XRL part. Our low-level explanation identifies misleading state-action pairs. This idea is widely used in XRL methods [D1,D2,D3,D4] where people want to explain the performance of the RL agent by identifying critical points. Therefore, the low-level explanation belongs to XRL. Our high-level explanation indeed uses IRL to learn a reward $\hat{r}$. However, this also belongs to XRL [D5] where the learned reward function $\hat{r}$ is used to explain the real reward function that the RL agent is actually optimizing.
>
> [D1] Amir, Dan, and Ofra Amir. "Highlights: Summarizing agent behavior to people." International Conference on Autonomous Agents and Multiagent Systems, 2018.
>
> [D2] Guo, Wenbo, et al. "Edge: Explaining deep reinforcement learning policies." Advances in Neural Information Processing Systems, 2021.
>
> [D3] Cheng, Zelei, et al. "Statemask: Explaining deep reinforcement learning through state mask." Advances in Neural Information Processing Systems, 2023.
>
> [D4] Jacq, Alexis, et al. "Lazy-MDPs: Towards Interpretable RL by Learning When to Act." International Conference on Autonomous Agents and Multiagent Systems, 2022.
>
> [D5] Xie, Yuansheng, Soroush Vosoughi, and Saeed Hassanpour. "Towards interpretable deep reinforcement learning models via inverse reinforcement learning." International Conference on Pattern Recognition, 2022.

---

> > ### Author Response · Authors · 2024-11-24
> >
> > Dear Reviewer,
> >
> > We hope that you've had a chance to read our responses and clarification. As the end of the discussion period is approaching, we would greatly appreciate it if you could confirm that our updates have addressed your concerns.
> >
> > Best,
> > Authors.

---

> > > ### Comment · Reviewer_iSK2 · 2024-11-25
> > > **response**
> > >
> > > Thank you for your response. I have read it, but I still find the problem formulation too ad hoc.
> > >
> > > For example, consider the following MDP with 3 states.
> > > In state A, action A1 gets reward 0 and moves to state B. Action A2 gets reward 0 and moves to C.
> > > In state B, action B1 gets reward 0.5 and terminates, action B2 gets reward 1 and terminates.
> > > In state C, action C1 gets reward 0 and terminates, and action C2 gets reward 2 and terminates.
> > >
> > > Optimal policy is clearly A->A2, B->B2, C->C2.
> > >
> > > Consider a policy A->A2, B->B1, C->C1. Under this policy, all state actions are misleading, including (A,A2), even though (A,A2) belongs to the optimal policy.
> > >
> > > Equation (1) posits to add a cost for visiting (A,A2), thereby potentially limiting the algorithm from reaching an optimal policy.
> > >
> > > Now, I understand that by playing around with the parameter b, this cost can be relaxed, but then what is a good value of b that will still allow to benefit from the method? How do we know it in advance? Or is it a hyperparameter that we need to find by trial and error? Not clear, and this is just a 3-state MDP! And we haven't yet talked about the IRL reward part...
> > >
> > > I hope this example clarifies why I think that some theory for motivating Equation (1) is essential.
> > >
> > > PS. Please correct me if I missed something in this example.

---

> ### Author Response · Authors · 2024-11-26
>
> Thank you very much for your detailed reply. We fully understand your comment that avoiding misleading state-action pairs may limit the algorithm from finding an optimal policy and thus theory is needed to motivate problem (1). We address your comment in the following three parts. In part (1), we elaborate that our method (avoiding misleading state-action pairs) is inspired by monotonic policy improvement theorem because our method can improve the original policy $\pi_A$ by helping the policy satisfy the monotonic policy improvement theorem. In part (2), we discuss how our method can improve the policy in your mentioned example. In part (3), we discuss how we get the budget $b$ in the experiments.
>
> **Part (1)**: We would like to first emphasize that our goal is to improve the original policy $\pi_A$ (as shown in the paper title) instead of finding an optimal policy. We acknowledge that finding an optimal policy can improve $\pi_A$, however, improving $\pi_A$ does not necessarily mean that we have to find an optimal policy. We next elaborate why avoiding the misleading state-action pairs can improve $\pi_A$. In brief, we show that our method of avoiding the misleading state-action pairs can help $\pi_A$ satisfy monotonic policy improvement theorem, and thus help improve $\pi_A$.
>
> To start with, we first recall the problem setting of improving $\pi_A$. We first run an RL algorithm (e.g., SAC) until convergence to get $\pi_A$. The converged policy $\pi_A$ is suboptimal, i.e., it is stuck in a sub-optimal stage without further improvement. According to monotonic policy improvement theorem, $\pi_A$ will improve if it chooses greedy actions according to its Q-function $Q_r^{\pi_A}$ at all the states. Given that $\pi_A$ stops improving, it means that $\pi_A$ chooses some nongreedy actions at some states.  In our paper, we identify these as misleading state-action pairs, i.e., the state-action pairs where $\pi_A$ chooses nongreedy actions at the states. Here, we refer to the states of the misleading state-action pairs as critical states. Constraining these misleading state-action pairs means that we constrain the nongreedy actions $\pi_A$ originally chooses at the critical states. This constraint can help $\pi_A$ choose greedy actions because it eliminates some nongreedy actions and thus $\pi_A$ only needs to find greedy actions from smaller action sets at the critical states. Since this constraint can help $\pi_A$ find greedy actions, it can help improve $\pi_A$ according to monotonic policy improvement theorem.
>
> In conclusion, we agree that the nongreedy actions of $\pi_A$ may be optimal actions of an optimal policy, and constraining these actions may limit the algorithm from finding an optimal policy. However, we aim to improve $\pi_A$ instead of finding an optimal policy, and constraining these nongreedy actions can help improve $\pi_A$ according to monotonic policy improvement theorem. This property that constraining the misleading state-action pairs can improve $\pi_A$ motivates us to avoid visiting misleading state-action pairs.
>
> **Part (2)**: In your given example, we agree that constraining the misleading state-action pairs cannot lead to an optimal policy. However, it can lead to a policy better than the original policy. The original policy achieves zero cumulative reward if it starts from A. By avoiding the misleading state-action pairs, the new policy will choose A->A1, B->B2, C->C2 and achieves the cumulative reward of 1. This new policy is better than the original policy. Moreover, this new policy is optimal if the initial state is B or C.
>
> We agree that improving $\pi_A$ to an optimal policy is an interesting problem, however, this is not the problem we study. In practice, improving $\pi_A$ to an optimal policy could be a too ambitious and even unrealistic goal. For example, in the MuJoCo environment, it is even impossible to evaluate whether the obtained policy is optimal or not because we do not know the maximum cumulative reward a policy can achieve.
>
> **Part (3)**: The hyper-parameter $b$ needs to be manually tuned in the experiment. In our experiment, we simply choose $b=0$, i.e., hard constraint. In the ablation study in Table 2 (fifth column), it shows that we can still improve $\pi_A$ even if we simply avoid the misleading state-action pairs. This aligns with the theory we elaborate in part (1), i.e., simply avoiding the misleading state-action pairs may not lead to an optimal policy but can improve the original policy $\pi_A$ by encouraging $\pi_A$ to choose greedy actions. In our answer to Weakness 1.6, we also empirically show how different $b$ will affect the learning performance.
>
> We really appreciate your detailed reply, and we are happy to discuss if you have further comments.

---

> > ### Comment · Reviewer_iSK2 · 2024-11-27
> > **follow up**
> >
> > Thanks for the explanation, this is helpful. I suggest you add the discussion to the paper.
> >
> > Let me make sure that I understand:
> > The misleading state are not changed during the run of the algorithm, and remain constant, right? So in my example above, your method will converge to an improved but suboptimal policy as the misleading states are not changed.
> >
> > Two questions:
> > 1. Why not update the misleading states? In my example, if you would update the misleading states, two iterations of your method would converge to optimal policy.
> > 2. In your experiments, if I understand correctly, the midleading states will only be calculated in the beginning of the run from some demonstration data, and not update afterwards. Where is this demonstration data coming from and what is it exactly?
> >
> > Additional question regarding the X-RL presentation (not related to the discussion above): is there any experirement or evaluation in the paper that actually reports or measures explainability? I still posit that this is not an X-RL paper, as the ideas here are standard RL (monotonic policy improvement, as you explained nicely) + IRL. Am I missing something? (yes, I understand the X-RL motivation in the story, and it should definitely be mentioned, but this is not an X-RL paper if it does not report any X-RL results).

---

> ### Author Response · Authors · 2024-11-27
>
> Thanks for your reply. We appreciate that you find our discussion helpful. Following your suggestion, we add the discussion to the newly uploaded version (highlighted in blue in lines 244-251). Your understanding about the misleading state-action pairs is correct. The misleading state-action pairs are not updated during the run of the algorithm. We next answer your three questions:
>
> **Question 1**: Why not update the misleading states? In my example, if you would update the misleading states, two iterations of your method would converge to optimal policy.
> **Answer**: The main reason is the practical issue of sample complexity. We agree that if we update the misleading state-action pairs when we have a new policy, it will help every learned policy find greedy actions and thus facilitate policy improvement during the whole run. However, finding misleading state-action pairs requires to significantly sample the environment. In specific, in order to find the misleading state-action pairs of a learned policy $\pi$, we need to learn a quite accurate $Q$-function $Q_r^{\pi}$. This requires to sample many episodes from the environment so that we can collect enough data to learn an accurate $Q_r^{\pi}$. In our scenario, we sample 100 episodes to learn $Q_r^{\pi_A}$ in the explanation part. If we want to update the misleading state-action pairs for every learned policy, it means that we need to sample 100 episodes every iteration just to find misleading state-action pairs. In our current algorithm, we only sample 10 episodes every iteration (10\% of the samples needed to find misleading state-action pairs). Moreover, the baselines (e.g., SAC and the other algorithms that can improve SAC) in the experiment also sample 10 episodes every iteration. If we update the misleading state-action pairs, it would be an unfair comparison between our algorithm and SAC because our algorithm would require ten times as many samples as SAC per iteration and SAC typically does not sample many (e.g., 100) episodes per iteration due to the consideration of sample efficiency. Our method aims to improve RL under the constraint that we do not require additional samples during improvement. This is evident in Figure 2 where we plot the performance of our algorithm and the baselines under the same amount of environment interaction steps.
> In fact, sample efficiency is an important problem in RL because sampling can be slow and expensive [D6,D7,D8]. Therefore, many recent RL algorithms (e.g., SAC, DDPG, TD3) are off-policy and use replay buffers to make sample efficient.
> In conclusion, we agree that updating misleading state-action pairs can help improve the performance. However, this can significantly increase sampling overhead and thus is sample inefficient.
>
> **Question 2**: In your experiments, if I understand correctly, the misleading states will only be calculated in the beginning of the run from some demonstration data, and not update afterwards. Where is this demonstration data coming from and what is it exactly?
> **Answer**: In our paper, we aim to improve an RL agent whose original policy is $\pi_A$. In specific, we first run the RL agent until convergence to get $\pi_A$. Following XRL literature [D9,D10,D11], we only treat the RL agent as a black box (lines 155-160). In specific, we can only observe a set $\mathcal{D}$ of trajectories demonstrated by the RL agent (using $\pi_A$) where each trajectory is a sequence of state-action pairs $s_0,a_0,s_1,a_1,\cdots$. In the experiment, we first run SAC until convergence and use the converged policy to sample demonstrated trajectories where each trajectory is a sequence of state-action pairs. These demonstrated trajectories are $\mathcal{D}$, from which we learn misleading state-action pairs.

---

> > ### Author Response · Authors · 2024-11-27
> >
> > **Question 3**: Additional question regarding the X-RL presentation (not related to the discussion above): is there any experiment or evaluation in the paper that actually reports or measures explainability? I still posit that this is not an X-RL paper, as the ideas here are standard RL (monotonic policy improvement, as you explained nicely) + IRL. Am I missing something? (yes, I understand the X-RL motivation in the story, and it should definitely be mentioned, but this is not an X-RL paper if it does not report any X-RL results).
> > **Answer**: In the experiment section, we evaluate our developed explanations in lines 499-519 and Table 3. In specific, we use fidelity as the metric to evaluate our explanations. Fidelity is a widely used metric in XRL literature [D9,D10,D12,D13] to evaluate the correctness of the explanation. Fidelity measures explainability in the sense that it measures whether the explanation is indeed a cause [D12,D13]. In our problem, our explanation explains why the original policy $\pi_A$ does not achieve high cumulative reward. Therefore, one way to evaluate the fidelity of the explanation is to evaluate whether the cumulative reward will increase after we improve from the explanation. Note that this is a standard way to evaluate fidelity in the literature of using XRL to improve RL [D9,D10]. The results in Table 3 show that our explanation has high fidelity.
> >
> > [D6] Haarnoja, Tuomas, et al. "Soft actor-critic: Off-policy maximum entropy deep reinforcement learning with a stochastic actor." International Conference on Machine Learning, 2018.
> >
> > [D7] Wang, Xu, et al. "Deep reinforcement learning: A survey." IEEE Transactions on Neural Networks and Learning Systems, 2022.
> >
> > [D8] Zabihi, Zeinab, Amir Masoud Eftekhari Moghadam, and Mohammad Hossein Rezvani. "Reinforcement learning methods for computation offloading: a systematic review." ACM Computing Surveys, 2023.
> >
> > [D9] Guo, Wenbo, et al. "Edge: Explaining deep reinforcement learning policies." Advances in Neural Information Processing Systems, 2021.
> >
> > [D10] Cheng, Zelei, et al. "Statemask: Explaining deep reinforcement learning through state mask." Advances in Neural Information Processing Systems, 2023.
> >
> > [D11] Bewley, Tom, and Jonathan Lawry. "Tripletree: A versatile interpretable representation of black box agents and their environments." AAAI Conference on Artificial Intelligence, 2021.
> >
> > [D12] Milani, Stephanie, et al. "Explainable reinforcement learning: A survey and comparative review." ACM Computing Surveys, 2024.
> >
> > [D13] Xiong, Yu, et al. "XRL-Bench: A Benchmark for Evaluating and Comparing Explainable Reinforcement Learning Techniques." ACM SIGKDD Conference on Knowledge Discovery and Data Mining, 2024.

---

> > > ### Comment · Reviewer_iSK2 · 2024-11-28
> > > **follow up**
> > >
> > > Thank you for the clarifications. I understand the paper better now.
> > >
> > > I think that the current presentation of the paper, while possibly appropriate for the X-RL community, is not appropriate for the much larger general RL community in ICLR, even though this paper may be of interest as a general RL algorithm to improve a policy.  I propose the following list of changes. If the authors agree to perform them, I will recommend to accept the paper (no need to update the manuscript during rebuttal, take your time and make the changes in the final version).
> > >
> > > 1. Based on the preceding discussion, we understand that the inner optimization in Eq. (1), ignoring the entropy term and assuming $r_{\theta} = r$, is equivalent to a single policy iteration update. Therefore, it is guaranteed to improve the policy (or keep it the same iff it is already optimal). Now, for a linear reward parametrization, if we set $\theta = 0$ we get that $r_\theta = r$, as above. For the outer optimization over $\theta$, since it is optimized over the real reward, and we understand that the case  $r_\theta = r$ corresponds to a particular $\theta$, we are guaranteed that (again, ignoring the entropy term) the output of the optimization in Eq. (1) is at least as good as a policy iteration step, and therefore also guaranteed to improve the policy.
> > > 2. Based on the above, I ask the authors to include a formal proposition that shows that Eq. (1), without the entropy term and for linear reward, results in an improved policy. The authors can also tackle the entropy case, which seems more involved, but I don’t think is necessary as adding an entropy bonus is standard, and can be motivated as such.
> > > 3. Based on the above, I ask the authors to tone down the X-RL terminology of high-level and low-level explanations, and instead present their method as a general method that approximates a policy iteration step. This will be of interest to the general RL community, and will present their method in a way that the general community can understand. No problem with being motivated by X-RL, but the general reader should understand right from the start that what you’re actually doing here is a sort of improved, approximate, policy iteration step.
> > > 4. The authors should explain that in principle, one can use their method iteratively, to perform several policy iteration steps, and eventually converge to an optimal policy (ignoring the entropy term). To motivate why only doing a single step still makes sense, they can refer to the work on the effective horizon [1], which claimed that many problems (Mujoco included) can be solved by doing one step of policy improvement over a random policy. So a single step could still yield a very strong improvement.
> > > 5. In the experiments section, the authors should make their protocol very clear right from the start - how they generate the demonstration data, and why it makes sense. Looking at the comparison with SAC in figure 2, this is very confusing, as SAC and UTILITY are compared along the same X axis (samples), but actually, UTILITY starts off where SAC stopped and plateaued. I would change the figure to clarify that this comparison is not apples to apples (unless I missed something). I don’t find this to be a problem in the evaluation (if SAC has plateaued, improving it is important even if it requires more samples), but this should be clarified better in the presentation of the figure and text.
> > > 6. There are other RL methods that take demonstrations from a policy, and learn an improved policy using interactions with the environment (or with offline data). Take a look at [2], although there are many others since. This part of the literature should be discussed and compared with (at least in the text, but ideally also experimentally).
> > > 7. Bonus: run your method for more than 1 iteration and see if you get improvement (as I explained above, no problem with requiring more samples - if a method plateaued it’s OK to use more samples to improve final performance)
> > >
> > >
> > > [1] Laidlaw, Cassidy, Stuart J. Russell, and Anca Dragan. "Bridging rl theory and practice with the effective horizon." Advances in Neural Information Processing Systems 36 (2023): 58953-59007.
> > >
> > > [2] Brown, Daniel S., Wonjoon Goo, and Scott Niekum. "Better-than-demonstrator imitation learning via automatically-ranked demonstrations." Conference on robot learning. PMLR, 2020.

---

> ### Author Response · Authors · 2024-11-28
>
> Thank you for the detailed and constructive suggestions. We will follow these seven suggestions and revise the paper accordingly. We really appreciate your time and feedback, which significantly help us improve the paper. It is a pleasure to discuss with you. Finally, we would like to thank you for recognizing the contributions of this work.
>
> Best,
> Authors

---

> ### Author Response · Authors · 2024-11-30
>
> Dear Reviewer iSK2,
>
> We would like to thank you again for your constructive suggestions and we are now revising the paper according to your advice. Could you please update the rating so that it is clearer for the AC to know that you recommend to accept the paper?
>
> Best,
> Authors

---

> > ### Author Response · Authors · 2024-11-30
> >
> > We sincerely appreciate your time and effort in reviewing our paper and thank you for recognizing the contributions of this work.

---

### Official Review · Reviewer_KPMH · 2024-10-28

**Soundness:** 4
**Presentation:** 4
**Contribution:** 4
**Rating:** 8
**Confidence:** 2

**Summary:**

The paper proposes to solve two problems: 1) the incomprehensibility of reinforcement learning agents and 2) the suboptimal performance of the agents. It introduces a two-level explanation, consisting of a high-level explanation that illustrates the reward function the agent has learned internally. The low-level explanation shows the state-action pairs where an agent chooses suboptimal actions with respect to an optimal policy. The suboptimal state-action pairs are called misleading. They use the low-level explanations to explain when it is infeasible to utilize the high-level explanations, for example, environments with high dimensional features. The misleading state actions pairs and the high-level explanations together are used to improve policy performance via a new loss function. A new algorithm is proposed to solve the loss function introduced to improve policy performance. The proposed method is experimentally tested in several MuJoCo environments.

Note: my expertise lies in explainability, thus my opinion on the second contribution (model performance) should weigh less.

**Strengths:**

- The paper proposes a new way to utilize misleading state action pairs to improve policy performance. Furthermore, they explain why the policy did not attain optimal performance in the first place.
- The paper has done a great job on covering related work and contrasted to them.
- The method proposed shows better performance in terms of return than all the baseline methods. And has good ablation tests to see which part of the method that contributes most to the results.
- The authors have included code with the paper.

**Weaknesses:**

- It is unclear to me why the policy would learn the wrong reward function internally in the first place. I might have misunderstood something.
- It is unclear why theorem 1 is true. I do not see reference to any proof.
- I am unsure how the performance would translate to environments with high-dimensional features. For example, Atari environments. Is there a reason why only MuJoCo environments are chosen for experiments?
- It is unknown how useful these explanations are to human users, and who is the target audience of these explanations.
- The main novelty is on improving model performance using explanations. The explanations themselves are not new to this work.

**Questions:**

- Definition 1: Why is the Q-function taken with respect to $\pi_A$. Isn't $\pi_A$ the suboptimal policy?
- Line 245: What is the functional form of the reward shaping function?
- Line 249: Why do we not want to visit misleading states? I thought it is where the agent takes a suboptimal action. Can we not fix the problem by just taking the optimal action when we encounter a misleading state?

---

> ### Author Response · Authors · 2024-11-20
>
> Thank you very much for your thoughtful consideration of our paper and the helpful comments and suggestions. We would hope that our answers will alleviate your concerns. We address your comments below:
>
> **Weakness 1**: "It is unclear to me why the policy would learn the wrong reward function internally in the first place. I might have misunderstood something."
> **Answer**: The reason is that the original policy $\pi\_A$ is sub-optimal. Our paper aims to improve the sub-optimal policy $\pi\_A$. The internal reward function $\hat{r}$ of $\pi\_A$ is the reward function, to which $\pi\_A$ is optimal. If the internal reward function is the ground truth reward function (i.e., $\hat{r}=r$), $\pi\_A$ would be an optimal policy because $\pi\_A$ is optimal to the internal reward function.
>
> **Weakness 2}** "It is unclear why theorem 1 is true. I do not see reference to any proof."
> **Answer**: The proof of Theorem 1 is in Appendix D.2. Here we provide the our proof logic of Theorem 1. The fundamental logic is that we first prove that the dual problem (i.e. the lower-level problem in (2)) has a unique optimal solution due to its strict convexity and then we prove that the primal problem (i.e. the lower-level problem in (1)) and the dual problem have the same set of optimal solutions. Since the optimal solution of the dual problem is unique, then the optimal solution of the primal problem is also unique.
>
> **Weakness 3**: "I am unsure how the performance would translate to environments with high-dimensional features. For example, Atari environments. Is there a reason why only MuJoCo environments are chosen for experiments?"
> **Answer**: We choose MuJoCo environments because it is a widely-used RL benchmark. It is interesting to explore how our algorithm performs in Atari, and we will do this in our future works.
>
> **Weakness 4**: "It is unknown how useful these explanations are to human users, and who is the target audience of these explanations."
> **Answer**: The explanations can help humans understand why the original policy $\pi\_A$ is sub-optimal and thus utilize the explanations to improve $\pi\_A$. Therefore, there are two sets of target audience. The first set is the people who are simply curious about why $\pi\_A$ is sub-optimal. Take Figure 1 as an example, our explanations are easy for nonexperts to understand. The second set is the people who want to improve RL. Our explanations help identify some mistakes made by $\pi\_A$, therefore, people can improve $\pi\_A$ by correcting those mistakes.
>
> **Weakness 5**: "The main novelty is on improving model performance using explanations. The explanations themselves are not new to this work."
> **Answer**: We agree that our main novelty is on improving RL performance and our explanations build on some high-level idea of XRL literature. However, our low-level explanation (i.e., finding misleading state-action pairs) is new. The XRL literature [C1,C2,C3] has the idea of identifying critical states that are most important to cumulative reward. However, these critical states do not explain why the original policy $\pi\_A$ is sub-optimal. Built on the "finding critical points" idea, our paper is the first one to explain why the original policy is sub-optimal by identifying "misleading" state-action pairs that lead to sub-optimality.
>
> **Question 1**: "Definition 1: Why is the Q-function taken with respect to $\pi\_A$. Isn't
> $\pi\_A$ the suboptimal policy?"
> **Answer**: $\pi\_A$ is the sub-optimal policy. Definition 1 aims to find misleading state-action pairs of policy $\pi\_A$. We agree that the ideal case is that we can compare the Q-functions between the optimal policy $\pi^{\ast}$ and $\pi\_A$, i.e., $Q\_r^{\pi^{\ast}}(s,a)-Q\_r^{\pi\_A}(s,a)$. However, this metric is infeasible because $\pi^{\ast}$ is unknown. We propose a feasible metric in definition 1 which only uses $\pi\_A$. As mentioned below Definition 1 in lines 209-213, we prove in Appendix B.1 that the policy $\pi\_{A}$ will be an optimal policy if $l(s,a)=0$ for all $(s,a)$ that $\pi\_A$ visits with nonzero probability. Therefore, any $(s,a)$ such that $l(s,a)>0$ can be regarded as a ``misleading" state-action pair that leads the policy $\pi\_A$ to be nonoptimal.
>
> **Question 2**: "Line 245: What is the functional form of the reward shaping function?"
> **Answer**: In the experiment, we consider linear shaping reward, i.e., $r\_{\theta}(s,a)=r(s,a)+\theta(s,a)(r(s,a)-\hat{r}(s,a))$. The shaping parameter $\theta$ is a neural network whose input is $(s,a)$ and output is a scalar.

---

> > ### Author Response · Authors · 2024-11-20
> >
> > **Question 3**: Line 249: Why do we not want to visit misleading states? I thought it is where the agent takes a suboptimal action. Can we not fix the problem by just taking the optimal action when we encounter a misleading state?
> > **Answer**: It is not the case that we do not want to visit misleading states. We do not want to visit misleading state-action pairs instead of misleading states. We can still visit misleading states, but do not choose the misleading actions (i.e., suboptimal actions) at the misleading states.
> >
> > [C1] Amir, Dan, and Ofra Amir. "Highlights: Summarizing agent behavior to people." International Conference on Autonomous Agents and Multiagent Systems, 2018.
> >
> > [C2] Guo, Wenbo, et al. "Edge: Explaining deep reinforcement learning policies." Advances in Neural Information Processing Systems, 2021.
> >
> > [C3] Cheng, Zelei, et al. "Statemask: Explaining deep reinforcement learning through state mask." Advances in Neural Information Processing Systems, 2023.

---

> > > ### Author Response · Authors · 2024-11-24
> > >
> > > Dear Reviewer,
> > >
> > > We hope that you've had a chance to read our responses and clarification. As the end of the discussion period is approaching, we would greatly appreciate it if you could confirm that our updates have addressed your concerns.
> > >
> > > Best,
> > > Authors.

---

> > > > ### Comment · Reviewer_KPMH · 2024-11-25
> > > >
> > > > I would like to thank the authors for comprehensive answers and additional experiments to my and the other reviewers' questions and concerns. I believe the explainability research community can benefit from this work. I will change my score to reflect the changes the authors have made.

---

> > > > > ### Author Response · Authors · 2024-11-25
> > > > >
> > > > > We sincerely appreciate your time and effort in reviewing our paper and thank you for recognizing the contributions of this work

---

### Official Review · Reviewer_sY65 · 2024-11-02

**Soundness:** 3
**Presentation:** 4
**Contribution:** 3
**Rating:** 6
**Confidence:** 3

**Summary:**

This paper proposes an explainable reinforcement learning (XRL) based policy update algorithm. The algorithm utilizes notion of "misleading state-action pairs" in the agent's current behavior that lead the agent to perform poorly compared to the optimal policy. These misleading $(s, a)$ pairs are then to be avoided by the agent to be able to act optimally. The criterion for defining misleading $(s, a)$ is whether the advantage of the policy is negative. For these misleading experiences, a positive cost is assigned. Parallely, from the agent's demonstrations with application of inverse RL on top, a reward function $\hat{r}$ is recovered. This reward function is then used to explain agent's current actions, as the reward function explains why a particular action is taken in the demonstration by assigning higher values. The misleading $(s,a)$ constitute the low-level explanations and the IRL reward function $\hat{r}$ the high-level explanation. A bi-level algorithm is then set up, where the agent has to learn a policy such that its low-level cost is bounded, i.e., not many disadvantageous actions are taken while high-level **ground-truth returns** are maximized. The first level includes both the cost $c(s, a)$ and the reward shaping using ground truth reward $r$ and IRL reward $\hat{r}$.

The paper then goes into the details of optimization where a dual approximation of bilevel algorithm is proposed that substitutes reward function with dual variables. Further, theoretical guarantees are provided about the convergence of a practical approach that uses low-level $c$ and high-level $\hat{r}$ and outputs an optimal policy.


As for the experiments, this paper uses sparser variants of the MuJoCo environments, and tries to answer how much effective improvement their algorithm UTILITY provides against other XRL techniques when combined with standard SAC algorithm. Paper provides ablations for different considerations in the modeling of the algorithm. By removing low-level and high-level explanations, paper shows why both are needed to get superior performance.

**Strengths:**

1. I find a close connection between the proposed algorithm and algorithms in generalized policy iteration class, because (1) avoiding disadvantageous behavior under current policy is analogous to improving the policy (imagine giving a leeway when taking a greedy action and instead use budget over choosing worse actions), (2) learning a Q-function to understand the policy's performance is analogous to policy evaluation. So, as long as these two steps are implemented properly, the proposed algorithm is bound to improve the policy, and eventually help reach the optimum in state values. Again, I might have skipped the specific details about the convergence constants, learning rates, etc., but the algorithm seems about right.
2. The idea of finding K misleading $(s, a)$ to inform the policy update is very neat. This in itself is valuable as many times a post-hoc policy analysis might be necessary for identification of its limitations. The paper does a great job at also integrating the misleading with policy's current beliefs and using ground truth rewards to rectify those beliefs.
3. The paper is well written, the ideas are well presented, figures are illustrative. I like how authors wrote "challenges" bullet points for every proposed method and then provided solutions for them. The theorems and lemmas are invoked appropriately. Some parts were not clear to me, which I have asked in the following sections. Nevertheless, I enjoyed reading the paper for its theory-grounded explainable RL approach to policy optimization.

**Weaknesses:**

There are a few weaknesses of the paper:

1. The notion of l(s, a): The paper assumes access to environment interaction and hence I am considering that there is enough knowledge about the actions per given state. If that is so, the way l(s, a) is defined by the paper in definition 1, is simply negative of advantage function for policy $\pi_A$.
2. The choice of MuJoCo environment and plausible issues: MuJoCo environments allow the agent to complete an episode for full length while other environments might terminate early if certain states are reached. Subsequently, MuJoCo environments provide correlated "second-chances" to the agent to make similar mistakes and the misleading effect is easy to capture. The algorithm's performance depends highly on the coverage of the demonstrations given by the earlier policy. MuJoCo would end up allowing coverage over optimal state-action visitation distribution and hence the algorithm proposed in the paper would work with single low-level and high-level explanation. However, this will not hold in the other complex environments. Also, in the case of sparse environments, where the agent has not received enough feedback from the environment to capture the "misleading" state-actions, the algorithm might misattribute notion of misleading.
3. Issue with complexity: To me, the bilevel algorithm seems an overkill if explanations are to be used for RL policy improvement. First of all, the idea of low-level optimization is simply to keep the policy improving by ensuring only a budgeted amount of disadvantageous actions are taken. Secondly, solving an IRL problem simply to gather a reward function to compare with ground truth reward function and reshaping it for policy update is again analogous to policy evaluation, done in an unorthodox manner. Why should we not simply assign meanings from XRL literature to the policy iteration or rather generalized policy iteration (GPI) procedural steps and keep training the RL agent.
I don't see in any way the explanations during the training are helpful for the humans. The complexity of bilevel approach makes it even harder for the human to understand how various non-linear function approximators might be playing their role. Moreover, I am afraid that allowing SAC the same amount of compute as UTITLITY for per-step improvement, i.e., giving access to online rollouts and updating the policy under UTILITY's computational budget, we might end up with performance as good as that of UTILITY, if we are comparing only the extent of improvement to underlying RL policy updates.

**Questions:**

Suggestion -- I would suggest the authors to use RL-consistent notation and substitute -A(s, a) instead of l(s,a). This will also simplifies elucidation of the technique mentioned in the paper.

Q. Why did you take the dual into consideration? How does that help the algorithm? I mean, please provide description about how the dual forms an upper or a lower bound on the true objective against which this approximation is being taken. Further adding details about how this approximation is still valid with its loosening of the original objective would be helpful too. I find it hard to think that dual formulation for making the lower level optimization unconstrained and convex, and then using the optimal dual solution as reward function is still a valid way.

Suggestion -- To help illustrate UTILITY's explanations during the training iterations, it would be helpful if the authors provided evolution of Figure 1-c to Figure 1-d tracked across policy updates. This would support the utility of the UTILITY.

Q. I tried finding how the cost function is defined for the lower-level optimization. However, I could not find a technical section on definition of $c(s, a)$. Can authors please shed some light on this? Also, how do you decide the top-K misleading state-action pairs? This also was not mentioned anywhere, or I might have missed it.

Q. Can authors please demonstrate UTILITY's execution on a few more gridworld settings with reward variations: sparse, dense, multiple terminal states, etc.?

---

> ### Author Response · Authors · 2024-11-20
>
> Thank you very much for your thoughtful consideration of our paper and the helpful comments and suggestions. We would hope that our answers will alleviate your concerns. We address your comments below:
>
> **Weakness 1**: "The notion of $l(s,a)$: The paper assumes access to environment interaction and hence I am considering that there is enough knowledge about the actions per given state. If that is so, the way $l(s,a)$ is defined by the paper in definition 1, is simply negative of advantage function for policy $\pi\_A$.
> **Answer**: The loss function $l(s,a)$ is not the negative of advantage function. The advantage function of the policy $\pi\_A$ is $A\_r^{\pi\_A}(s,a)=Q\_r^{\pi\_A}(s,a)-V\_r^{\pi\_A}(s)$ where $V\_r^{\pi\_A}(s)=E\_{a\sim\pi(\cdot|s)}[Q\_r^{\pi\_A}(s,a)]$. However, the loss $l(s,a)=\max\_{a'}Q\_r^{\pi\_A}(s,a')-Q\_r^{\pi\_A}(s,a)$. Note that $V\_r^{\pi\_A}(s)=E\_{a\sim\pi(\cdot|s)}[Q\_r^{\pi\_A}(s,a)]\neq \max\_{a'}Q\_r^{\pi\_A}(s,a')$ unless the policy $\pi\_A$ is an optimal policy $\pi^{\ast}$ where $E\_{a\sim\pi^{\ast}(\cdot|s)}[Q\_r^{\pi^{\ast}}(s,a)]=\max\_{a'}Q\_r^{\pi^{\ast}}(s,a')$. However, the original policy $\pi\_A$ is only a suboptimal policy that we want to improve.
>
> **Weakness 2**: "The choice of MuJoCo environment and plausible issues: MuJoCo environments allow the agent to complete an episode for full length while other environments might terminate early if certain states are reached. Subsequently, MuJoCo environments provide correlated "second-chances" to the agent to make similar mistakes and the misleading effect is easy to capture. The algorithm's performance depends highly on the coverage of the demonstrations given by the earlier policy. MuJoCo would end up allowing coverage over optimal state-action visitation distribution and hence the algorithm proposed in the paper would work with single low-level and high-level explanation. However, this will not hold in the other complex environments. Also, in the case of sparse environments, where the agent has not received enough feedback from the environment to capture the "misleading" state-actions, the algorithm might misattribute notion of misleading."
> **Answer**: Thanks for expressing your concerns about **(1)** the misleading effect may be hard to capture for environments where the episode terminates early if certain states are reached; **(2)** the algorithm performance depends highly on the coverage of the demonstrations and may perform bad if demonstrations do not cover optimal state-action visitation distribution; and **(3)** the algorithm may fail in sparse environments where the agent does not receive enough feedback from the environment. We address your concerns respectively as follows:
> **For concern (1)**, we agree that if an agent makes similar mistakes multiple times, it is easier to find misleading state-action pairs. Therefore, for environments where the episode terminates early, it may be more difficult to find accurate misleading state-action pairs. However, this limitation does not bother the high-level explanation, i.e., the learned reward $\hat{r}$. In specific, we can always learn a reward $\hat{r}$ using ML-IRL, to which the original RL agent is optimal, regardless of whether the RL agent makes similar mistakes multiple times in the demonstrations or not. Therefore, as long as the original policy is not optimal, the learned reward $\hat{r}$ can capture this non-optimality and the comparison $r-\hat{r}$ quantifies the RL agent's misunderstanding about the real reward function $r$. Moreover, the ablation results in Table 2 show that the shaping reward (that uses the learned reward $\hat{r}$) is much more important than avoiding the misleading state-action pairs in terms of improving the RL performance. Therefore, even if the low level explanation may become less accurate when the episode terminates, our algorithm's performance will not degrade too much given that the performance highly relies on the shaping reward that uses $\hat{r}$ and we can always learn an accurate $\hat{r}$ using ML-IRL regardless of whether the episode terminates early or not. To better support our claim, we add an additional experiment on Lunar Lander of Gym where the episode terminates when the lander crashes or lands.
>
> | SAC            | UTILITY       | RICE          | SIL           | LIR           |
> |----------------|---------------|---------------|---------------|---------------|
> | $214.27\pm12.11$ | $356.48\pm8.55$ | $229.42\pm6.19$ | $271.56\pm14.57$ | $294.31\pm8.94$ |
>
> The results above show that our method (UTILITY) can still significantly outperform the baselines in environments where the episode terminates if certain states are reached.

---

> > ### Author Response · Authors · 2024-11-20
> >
> > **For concern (2)**, we agree that the coverage of the demonstrations may affect the misleading state-action pairs. However, this will not ruin the improvement led by the high-level explanation $\hat{r}$. Suppose the original policy is very bad, the demonstrations only cover bad state-action pairs and do not cover optimal state-action pairs. Since the demonstrations are optimal w.r.t. $\hat{r}$, the learned reward $\hat{r}$ will assign high reward to encourage bad state-action pairs and low reward to discourage optimal state-action pairs. This is contradictory to the ground truth reward $r$. Therefore, the comparison $r-\hat{r}$ will make the reward difference between the optimal state-action pairs and bad state-action pairs larger, and thus will lead to an augmented effect that encourages optimal state-action pairs and discourages bad state-action pairs.
> > **For concern (3)**, our experiments study sparse environments. Note that we study delayed MuJoCo environment where the reward is sparse because the rewards are accumulated for 20 time steps and provided only at the end of these periods (lines 455-456), and the results show that our method can significantly outperform the baselines. We agree that the sparse reward may make the identified misleading state-action pairs inaccurate. However, as mentioned in our answer to Concern (1), the shaping reward plays a major role in terms of improving performance. Even if the identified misleading state-action pairs are not accurate, the overall performance will not be ruined. To further study the identified misleading state-action pairs in sparse environments, we conduct an additional experiment. In specific, the identified misleading state-action pairs may have false positive (i.e., some identified state-action pairs are not actually misleading) and false negative (i.e., some misleading state-action pairs are not identified). The false negative will not make our algorithm worse than the original SAC because the original SAC does not consider any misleading state-action pairs. However, the false positive may ruin our algorithm because we may avoid some state-action pairs that are important to achieve high cumulative reward and should not be avoided. Our current method imposes hard constraint (i.e., the budget $b=0$) on the misleading state-action pairs, which means that we want to totally avoid the identified misleading state-action pairs. Here, we compare to positive budget $b>0$. When $b>0$, we impose a soft constraint on the misleading state-action pairs, so that we do not totally avoid the misleading state-action pairs. The misleading state-action pairs can still be visited by the policy if they lead to high cumulative reward. We choose $b=5$ and $b=20$ for comparisons. The values in the table below are the cumulative reward improvement.
> >
> > |         | Delayed HalfCheetah | Delayed Hopper | Delayed Walker | Delayed Ant |
> > |---------|---------------------|----------------|----------------|-------------|
> > | $b=0$   | $38.70$             | $17.22$        | $40.75$        | $7.05$      |
> > | $b=5$   | $38.14$             | $21.63$        | $42.19$        | $7.76$      |
> > | $b=20$  | $12.66$             | $5.29$         | $3.29$         | $1.48$      |
> >
> > The results above show that when $b=5$, the cumulative reward improvement will become larger. This shows that we have false positive misleading state-action pairs and a small positive $b$ can reduce the negative impact of those false positive misleading state-action pairs. When $b=20$, the cumulative improvement becomes much smaller because at this time, many misleading state-action pairs that should be avoided are not avoided. The results above indicate that while our misleading state-action pairs have some false positives, the majority is accurate.

---

> > > ### Author Response · Authors · 2024-11-20
> > >
> > > **Weakness 3**: Issue with complexity: To me, the bilevel algorithm seems an overkill if explanations are to be used for RL policy improvement. First of all, the idea of low-level optimization is simply to keep the policy improving by ensuring only a budgeted amount of disadvantageous actions are taken. Secondly, solving an IRL problem simply to gather a reward function to compare with ground truth reward function and reshaping it for policy update is again analogous to policy evaluation, done in an unorthodox manner. Why should we not simply assign meanings from XRL literature to the policy iteration or rather generalized policy iteration (GPI) procedural steps and keep training the RL agent. I don't see in any way the explanations during the training are helpful for the humans. The complexity of bilevel approach makes it even harder for the human to understand how various non-linear function approximators might be playing their role. Moreover, I am afraid that allowing SAC the same amount of compute as UTITLITY for per-step improvement, i.e., giving access to online rollouts and updating the policy under UTILITY's computational budget, we might end up with performance as good as that of UTILITY, if we are comparing only the extent of improvement to underlying RL policy updates.
> > > **Answer**: Thanks for mentioning the comparison of our method and policy iteration. To start with, we first discuss the practical issues of policy iteration. We agree that in theory, policy iteration can lead to optimality. However, in practice, policy iteration may converge to a sub-optimal stage without further improvement, thus leading to sub-optimal RL performance. This is evident in RL literature [B1,B2,B3,B4] where people observe that RL agent is stuck in a sub-optimal stage without further improvement. This may be due to various reasons, including not enough data to get accurate policy evaluation and non-convexity of the neural value network and neural policy network, etc. In any way, the outcome is that the RL agent converges to a sub-optimal stage. In our case, we run the SAC algorithm until convergence to get the original policy $\pi\_A$. Note that the policy $\pi\_A$ already converges and cannot further improve. This is evident in Figure 2 where the curve of SAC converges. In this case, running SAC for more (policy) iterations will not improve $\pi\_A$. Moreover, the horizontal axis in Figure 2 shows the environment interactions of the algorithms. We report the performance of different algorithms under the same environment interactions. Figure 2 shows that under the same environment interactions, UTILITY converges to a much higher cumulative reward than SAC. In conclusion, (1) it is not the case that running more (policy) iterations will improve SAC because SAC already converges to a sub-optimal stage; (2) it is not the case that UTILITY leverages more online rollouts to achieve better performance because UTILITY and SAC use the same amount of online rollouts.
> > > Next, we explain why UTILITY can improve $\pi\_A$. Since $\pi\_A$ is suboptimal, it means that $\pi\_A$ must make some mistakes. Since $\pi\_A$ is stuck in a sub-optimal stage, it means that $\pi\_A$ keeps making mistakes and policy iteration cannot correct these mistakes. The brief idea of UTILITY is that we first find some mistakes made by $\pi\_A$, and then force the algorithm to correct these mistakes. We develop explanations to find mistakes. For the misleading state-action pairs, UTILITY formulates a constraint to force the algorithm to avoid visiting these state-action pairs. For the learned reward $\hat{r}$, the previous SAC is stuck in a sub-optimal stage w.r.t. $r$, and this sub-optimal stage is optimal w.r.t. $\hat{r}$. We can treat SAC as a biased black box in the sense that if the input of this black box is the reward function $r$, the output is a policy optimal to another reward function $\hat{r}$. Therefore, we can learn a shaping reward $r\_{\theta}$ such that the biased black box will output a policy optimal to $r$ if $r\_{\theta}$ is the input. Note that our explanations are not to help humans understand how problem (1) is solved. Instead, our explanations aim to help humans understand why $\pi\_A$ is not optimal by identifying some mistakes made by $\pi\_A$.
> > >
> > > **Suggestion 1**: "Suggestion -- I would suggest the authors to use RL-consistent notation and substitute -A(s, a) instead of l(s,a). This will also simplifies elucidation of the technique mentioned in the paper."
> > > **Answer**: Please refer to our answer to Weakness 1.

---

> > > > ### Author Response · Authors · 2024-11-20
> > > >
> > > > **Question 1**: "Why did you take the dual into consideration? How does that help the algorithm? I mean, please provide description about how the dual forms an upper or a lower bound on the true objective against which this approximation is being taken. Further adding details about how this approximation is still valid with its loosening of the original objective would be helpful too. I find it hard to think that dual formulation for making the lower level optimization unconstrained and convex, and then using the optimal dual solution as reward function is still a valid way."
> > > > **Answer**: Thanks for mentioning that we replace the lower-level constrained RL problem in (1) with its dual problem. This replacement is valid because we prove strong duality (lines 985-986), i.e., the optimal value of the dual problem is same as the optimal value of the primal problem (i.e., the lower-level constrained RL problem), so that this dual replacement is tight instead of forming an upper bound or lower bound of the primal problem. Moreover, Theorem 2 proves that the constrained soft Bellman policy parameterized by $\lambda^{\ast}(\theta)$ is the unique optimal policy of the primal problem, and $\lambda^{\ast}(\theta)$ is the unique optimal solution of the dual problem. Therefore, we can find the optimal policy of the primal problem by solving the dual problem to get $\lambda^{\ast}(\theta)$. In conclusion, replacing the primal problem with its dual problem is not an approximation but an equivalent replacement becasue we can find the optimal solution of the primal problem by solving the dual problem.
> > > >
> > > > **Suggestion 2**: "Suggestion -- To help illustrate UTILITY's explanations during the training iterations, it would be helpful if the authors provided evolution of Figure 1-c to Figure 1-d tracked across policy updates. This would support the utility of the UTILITY."
> > > > **Answer**: Following your suggestion, we provide the evolution of Figure 1c to Figure 1d in Appendix I in the newly uploaded version.
> > > >
> > > > **Question 2**: "I tried finding how the cost function is defined for the lower-level optimization. However, I could not find a technical section on definition of c(s,a). Can authors please shed some light on this? Also, how do you decide the top-K misleading state-action pairs? This also was not mentioned anywhere, or I might have missed it."
> > > > **Answer**: In lines 247-249, we discuss how to design $c(s,a)$. In specific, $c(s,a)\in(0,c\_{\max}]$ if $(s,a)\in C$ where $C$ is the set of identified misleading state-action pairs and $c(s,a)=0$ otherwise. Since we choose hard constraint $b=0$ (line 263), it does not matter what positive value $c(s,a)$ is for $(s,a)\in C$. As long as $c(s,a)>0$ for $(s,a)\in C$, the zero budget will force the algorithm to avoid $C$. In practice, we choose $c(s,a)=1$ for $(s,a)\in C$.
> > > > In lines 213-215, we discuss how to find the top K misleading state-action pairs. We use the misleading level $l(s,a)$ to find misleading state-action pairs. The top K misleading state-action pairs are the state-action pairs with top K highest misleading level $l(s,a)$.

---

> > > > > ### Author Response · Authors · 2024-11-20
> > > > >
> > > > > **Question 3**: "Can authors please demonstrate UTILITY's execution on a few more gridworld settings with reward variations: sparse, dense, multiple terminal states, etc.?"
> > > > > **Answer**: Note that our paper already studies both sparse reward in Table 1 (delayed MuJoCo environments) and dense reward in Table 6 (original MuJoCo environments). Our answer to weakness 2 also includes Lunar Lander experiments where the episode terminates early when the lander crashes or lands.
> > > > > Here we add a new gridworld experiment on Taxi of Gym. In this environment, the taxi needs to pick up a passenger and send the passenger to one of four goals in a gridworld. The action in this environment is discrete so that we use soft Q-learning as the baseline RL algorithm. In this environment, the episode ends when the taxi drops off the passenger at one of four goals. We consider both dense and sparse reward in this setting. The sparse reward assigns +100 to the taxi if the passenger is dropped off at goals and -5 otherwise. Note that the reward -5 aims to encourage the taxi to go to the goal as soon as possible, and does not make the reward dense. Since the reward is -1 almost everywhere except the goal states, the reward is still sparse because this is similar to the case where the reward is 0 everywhere except the goal states. The dense reward assigns +100 to the taxi if the passenger is dropped off at goals and $-d\_{taxi}$ otherwise to the taxi where $d\_{taxi}$ is the manhattan distance between the taxi and the nearest goal.
> > > > >
> > > > > |                  | Soft Q-learning    | UTILITY        | RICE          | SIL           | LIR           |
> > > > > |------------------|--------------------|----------------|---------------|---------------|---------------|
> > > > > | Sparse reward    | $37.28\pm 8.25$   | $78.22\pm5.04$ | $41.35\pm6.19$ | $58.15\pm7.42$ | $69.14\pm2.08$ |
> > > > > | Dense reward     | $69.36\pm3.29$    | $81.06\pm4.28$ | $69.52\pm5.15$ | $73.77\pm3.28$ | $76.57\pm4.32$ |
> > > > >
> > > > > From the results above, we can see that UTILITY outperforms the baselines in both sparse and dense reward settings.
> > > > >
> > > > > [B1] Haarnoja, Tuomas, et al. "Soft actor-critic: Off-policy maximum entropy deep reinforcement learning with a stochastic actor." International conference on machine learning, 2018.
> > > > >
> > > > > [B2] Henderson, Peter, et al. "Deep reinforcement learning that matters." Proceedings of the AAAI conference on artificial intelligence. Vol. 32. No. 1. 2018.
> > > > >
> > > > > [B3] Dulac-Arnold, Gabriel, Daniel Mankowitz, and Todd Hester. "Challenges of real-world reinforcement learning." arXiv preprint arXiv:1904.12901 2019.
> > > > >
> > > > > [B4] Cheng, Zelei, et al. "RICE: Breaking Through the Training Bottlenecks of Reinforcement Learning with Explanation." International Conference on Machine Learning, 2024.

---

> > > > > > ### Comment · Reviewer_sY65 · 2024-11-21
> > > > > >
> > > > > > Thanks for the detailed responses, I am increasing my score to 6.

---

> > > > > > > ### Author Response · Authors · 2024-11-24
> > > > > > >
> > > > > > > We sincerely appreciate your time and effort in reviewing our paper and thank you for recognizing the contributions of this work.

---

### Official Review · Reviewer_euxG · 2024-11-04

**Soundness:** 2
**Presentation:** 3
**Contribution:** 2
**Rating:** 6
**Confidence:** 3

**Summary:**

The authors propose a new approach of using explanations for reinforcement learning developed in the XRL community to improve RL algorithms. The design a two part method – the first part generates the explanations (two levels – learning an implied reward function and identifying ‘misleading’ states) and the second part utilizes this explanation to improve an RL agent’s performance by formulating a constrained bi-level optimization problem. Specifically, these two levels involve learning a reward shaping function and then a policy to optimize for this shaped reward. In addition to providing theoretical guarantees, the authors empirically demonstrate their method on four MuJoCo environments.

**Strengths:**

1. The authors address an important problem in RL – which is using explainability to improve performance. Their framework “UTILITY” is a novel approach to using XRL to optimize the RL agent’s performance.
2. The paper is well written. The authors provide a good background literature survey, explain the problem and their motivation well and also clearly describe their solution.
3. The authors provide a detailed theoretical analysis and proofs for their claims. While I am unable to verify these completely, I have not found any error. However, I think it will still be helpful if the authors provide a proof sketch or outline in the main paper itself.

**Weaknesses:**

1. Can the authors add a discussion on stochastic environments and the impact of the same on their approach? I think the current approach has only been demonstrated in deterministic environments. Specifically, can the authors discuss how their theoretical guarantees and empirical results might change in stochastic environments, and whether any modifications to their approach would be needed?
2. The current method seems to need a lot of environment interaction. Can the authors detail the additional interaction needed for their approach?
3. Can the authors add a discussion on the scalability of their proposed approach, in terms of state/action space size, or computational complexity as the problem size increases?
4. The impact of the error in explanation on approach and final policy has not been covered, i.e., errors in reward learning and Q value error and behavior policy learning error. Can the authors address these with a discussion in the paper? Specifically, can the authors provide a sensitivity analysis showing how errors in each component (reward learning, Q-value estimation, behavior policy learning) impact the final performance with some discussion on potential mitigation strategies for these errors?

**Questions:**

1. The key idea in this paper of learning an implied reward function of a learned RL agent and also identifying the state-action pairs that cause the agent to be non-optimal to be used in learning a new policy seems novel and interesting. However, wouldn’t this require re-learning? Wouldn’t generating the explanations require either access to the environment for generating additional data or knowledge of the environment parameters such as transition function, maximum attainable reward, goal state etc.? Can the authors add a discussion on this topic detailing what extra data and computation are needed by their method?
2. Can the authors also compare their approach with other RL algorithms from literature where the other algorithms have access to the same total quantity of data used by the authors’ approach? The current comparison in Fig 2, does have total environment interactions, can the authors confirm if these interactions also include the additional data mentioned in Point 1 above?
3. It is known from the literature on inverse RL that learning a reward function from data is a problem that can have multiple solutions. Can the authors describe how they address the non-uniqueness problem and how do they choose a single reward function?
4. Can the authors clarify if different plausible high-level explanations (learned reward functions) can lead to different low-level explanations (misleading states)? If so, how do the authors propose to select the best high-level explanation? What is the authors’ opinion on using feedback from the RL agent’s performance post using their approach as input to improve their explanation generation. i.e., can their approach benefit by making the process of explanation generation and policy improvement iterative?
5. Can the authors confirm if misleading states are states where the agent does not act optimally as per its own Q-function? Is this definition valid even if the Q-function learned by the RL agent is not accurate, which is often the case in several practical RL problems?
6. The authors state in Line 222 that: “However, our case is different because we only need to learn one precise Q-function, which corresponds to the specific policy $\hat{\pi}_A$”. But won’t the authors need this Q-function to be accurate for also state-action pairs that are not visited or rarely visited under this policy? This is because in defining the $l(s,a)$ metric, the authors use a max over all $Q$ values of a given state $s$. How do the authors propose to learn these values accurately or adjust the definition of their proposed metric to ensure that this does not impact their approach?
7. Can the authors clarify if the shaping reward function requires access to the ground-truth reward function? If so, how do the authors propose to obtain this?
8. Can the authors specify the loss function used for learning the proposed shaping reward function used in equation (1), i.e., can the authors provide a discussion on the gradients proposed in Lemma 2 and its relation to this shaping reward function?
9. When the authors claim uniqueness of solution in Theorems 1 and 2, it would be helpful if they can clarify whether only $J$ value is unique or is $\pi$ also unique?
10. Minor: There is a typo in “RICE+constaint” which occurs twice in the paper (Lines 513 and 514).

**Details Of Ethics Concerns:**

The ethical concerns are only limited to the concerns for any RL work: that caution must be exercised regarding potential biases and harmful outcomes when using these in real-world use cases.

---

> ### Author Response · Authors · 2024-11-20
>
> Thank you very much for your thoughtful consideration of our paper and the helpful comments and suggestions. We would hope that our answers will alleviate your concerns. We address your comments below:
>
> **Weakness 1**: "Can the authors add a discussion on stochastic environments and the impact of the same on their approach? I think the current approach has only been demonstrated in deterministic environments. Specifically, can the authors discuss how their theoretical guarantees and empirical results might change in stochastic environments, and whether any modifications to their approach would be needed?"
> **Answer**: Thanks for mentioning stochastic environments. While our experiments are only conducted in deterministic MuJoCo environments, our method can work for stochastic environments. The reason is that our method is based on a standard MDP where the state transition function $P(\cdot|\cdot,\cdot)$ is stochastic and we do not assume that the transition function is deterministic. In fact, our current theoretical guarantees hold for stochastic environments because our proof considers stochastic environments. In specific, our proof uses $\int_{s'}P(s'|s,a)ds'$ to consider the next state $s'$ given the current state-action $(s,a)$. This is evident in the proof of every theoretical statement of this paper, e.g., line 951-960 in the proof of Theorem 1, lines 1013-1017 in the proof of Lemma 2, etc. To show that our method works for stochastic environments empirically, we conduct an additional experiment on Lunar Lander of Gym where the state transition is stochastic because the next state is affected by the wind and the wind effect is random. We provide the results below:
>
> | SAC           | UTILITY       | RICE         | SIL          | LIR          |
> |---------------|---------------|--------------|--------------|--------------|
> | $214.27\pm12.11$ | $356.48\pm8.55$ | $229.42\pm6.19$ | $271.56\pm14.57$ | $294.31\pm8.94$ |
>
> From the results above, we can see that our method (UTILITY) can outperform the baselines in stochastic environments.
>
> **Weakness 2**: "The current method seems to need a lot of environment interaction. Can the authors detail the additional interaction needed for their approach?"
> **Answer**: We quantify the environment interactions using the number of episodes. Each episode includes T-step environment interactions where T is the episode length. The environment interactions of our method come from two part: explanation (i.e., IRL and finding misleading state-action pairs) and improvement (solving problem (1)). We discuss these two parts respectively as follows:
> For the explanation part, the IRL samples 500 episodes in total to learn a reward function where we run ML-IRL for 500 iterations and each iteration samples one episode. To find the misleading state-action pairs, we sample 100 episodes to learn the Q function. Therefore, we sample 600 episodes for the explanation.
> For the improvement part (solving problem (1)), Algorithm 1 has three loops. In the inner loop, at each inner iteration, we sample 1 episode from the environment to compute the constrained soft Q function (line 5 in Algorithm 1). Therefore, the inner loop samples totally $\tilde{N}$ episodes from the environment where $\tilde{N}$ is the iteration number of the inner loop. Note that we also use a replay buffer to store previous data. In the middle loop, at each middle iteration, we sample 1 episode to compute $g_{\lambda;\theta}$ (line 8 in Algorithm). Therefore, each middle iteration requires $1+\tilde{N}$ sampled episodes, and the middle loop requires totally $\bar{N}(1+\tilde{N})$ episodes where $\bar{N}$ is the iteration number of the middle loop. At each outer iteration, we compute $\hat{\pi}\_{\hat{\lambda}(\theta);\theta}$ via $(\bar{N}-1)$-step soft policy iteration (line 11 in Algorithm 1) and each soft policy iteration requires to sample 1 episode. Moreover, we sample 5 episodes to compute $g_{\theta}$ (line 12 in Algorithm 1). Therefore, at each outer iteration, the total number of sampled episodes is $\bar{N}(1+\tilde{N})+(\bar{N}-1)+5$. Due to the warm start trick (discussed in Appendix F.6), we can choose very small $\bar{N}$ and $\tilde{N}$, i.e., $\tilde{N}=1$ and $\bar{N}=2$ (mentioned in lines 1585-1586). Therefore, the total episodes sampled at each outer iteration is 10. We run Algorithm 1 for 1000 outer iterations, therefore, the total number of sampled episodes is 10000. For a fair comparison, we run SAC for 1000 iterations and sample 10 episodes at each iterations, so that SAC also samples 10000 episodes in total.
>
> In conclusion, solving problem (1) does not introduce additional interactions compared to SAC. The additional interactions of our method are the 600 episodes required for the explanation. Therefore, compared to standard SAC, our method requires 600/10,000=6% extra environment interactions.

---

> > ### Author Response · Authors · 2024-11-20
> >
> > **Weakness 3**: "Can the authors add a discussion on the scalability of their proposed approach, in terms of state/action space size, or computational complexity as the problem size increases?"
> > **Answer**: Our algorithm has two parts: policy learning (to solve the lower-level problem) and reward learning (to solve the upper-level problem). We discuss their scalability in terms of state/action space size respectively.
> > The policy learning uses SAC, and thus its scalability depends on the scalability of SAC. A standard result in [A1] shows that the computation of SAC scales at $O(|\mathcal{S}\times\mathcal{A}|)$.
> > For the reward learning, the dimension of state-action space can affect the computation in the sense that it will change the input dimension of the reward model, and our algorithm needs to compute the gradient of the reward model w.r.t. its parameters. We use neural network as the reward model, and a standard result in [A2] shows that the computation of the gradient of neural networks scales at $O(d_s+d_a)$ where $d_s$ and $d_a$ are respectively the dimensions of state and action. Therefore, the computation of reward learning part scales at $O(d_s+d_a)$.
> > In conclusion, the computation of the overall algorithm scales at $O((d_s+d_a)|\mathcal{S}\times\mathcal{A}|)$.

---

> ### Author Response · Authors · 2024-11-20
>
> **Weakness 4**: "The impact of the error in explanation on approach and final policy has not been covered, i.e., errors in reward learning and Q value error and behavior policy learning error. Can the authors address these with a discussion in the paper? Specifically, can the authors provide a sensitivity analysis showing how errors in each component (reward learning, Q-value estimation, behavior policy learning) impact the final performance with some discussion on potential mitigation strategies for these errors?"
> **Answer**: Thanks for mentioning how the error in the explanation impacts the final performance. We respectively discuss the impacts of the error of $\hat{r}$ and the impact of the error of behavior policy $\hat{\pi}_A$ and Q value as follows.
>
> **For the error of the reward learning**, suppose that we learn a reward $r'$ such that $||\hat{r}(s,a)-r'(s,a)||\_{\infty}\leq\epsilon$. Theorem 4 shows that if the state-action space is finite and the reward model $r\_\theta$ is linear and we obtain $\hat{r}$, our algorithm can achieve optimality, i.e., $\lim\_{N,\bar{N}\rightarrow\infty}J\_r(\hat{\pi}\_{\hat{\lambda}(\theta\_N);\theta\_N})-J^{\ast}=0$. If we only obtain $r'$ instead of $\hat{r}$, we can get that $\lim\_{N,\bar{N}\rightarrow\infty}J^{\ast}-J\_r(\hat{\pi}\_{\hat{\lambda}(\theta\_N);\theta\_N})\leq O(\epsilon)$. We provide the proof sketch as follows:
> For simplicity, we use $\pi_{\hat{r}}$ to denote the learned policy under $\hat{r}$ (i.e., the optimal soft policy corresponding to $r\_{\theta}(r,r-\hat{r})$) and $\pi\_{r'}$ to denote the learned policy under $r'$ (i.e., the optimal soft policy corresponding to $r\_{\theta}(r,r-r')$). Therefore, $J\_r(\pi\_{\hat{r}})-J\_r(\pi\_{r'})\leq \frac{r\_{\max}}{1-\gamma}||\psi^{\pi\_{\hat{r}}}(s,a)-\psi^{\pi\_{r'}}(s,a)||\_{TV}$ where $r\_{\max}=\max\_{(s,a)\in\mathcal{S}\times\mathcal{A}}|r(s,a)|$, $|\cdot|\_{TV}$ is the total variation, and $\psi^{\pi}$ is the stationary state-action distribution of policy $\pi$. We first ignore the constraint caused by the misleading state-action pairs. Note that $\pi\_{\hat{r}}\propto \exp(Q\_{r\_{\theta}(r,r-\hat{r}),\pi\_{\hat{r}}}^{soft})$ and $\pi\_{r'}\propto \exp(Q\_{r\_{\theta}(r,r-r'),\pi\_{r'}}^{soft})$ (see the expression of the soft policy (equation 3)), from Lemma 3 in [A3] (also equation 64 in [A4]), we can get that $J\_r(\pi\_{\hat{r}})-J\_r(\pi\_{r'})\leq \frac{r\_{\max}}{1-\gamma}||\psi^{\pi\_{\hat{r}}}(s,a)-\psi^{\pi\_{r'}}(s,a)||\_{TV}\leq \frac{Cr\_{\max}}{1-\gamma}||Q\_{r\_{\theta}(r,r-\hat{r}),\pi\_{\hat{r}}}^{soft}(s,a)-Q\_{r\_{\theta}(r,r-r'),\pi\_{r'}}^{soft}(s,a)||\_{\infty}$ where $C$ is a positive constant. Since $\pi\_{r'}$ is the optimal soft policy corresponding to $r\_{\theta}(r,r-r')$, we have that $Q\_{r\_{\theta}(r,r-\hat{r}),\pi\_{\hat{r}}}^{soft}(s,a)-Q\_{r\_{\theta}(r,r-r'),\pi\_{r'}}^{soft}(s,a)\leq Q\_{r\_{\theta}(r,r-\hat{r}),\pi\_{\hat{r}}}^{soft}(s,a)-Q\_{r\_{\theta}(r,r-r'),\pi\_{\hat{r}}}^{soft}(s,a)$. Similarly, we have that $Q\_{r\_{\theta}(r,r-\hat{r}),\pi\_{\hat{r}}}^{soft}(s,a)-Q\_{r\_{\theta}(r,r-r'),\pi\_{r'}}^{soft}(s,a)\geq Q\_{r\_{\theta}(r,r-\hat{r}),\pi\_{r'}}^{soft}(s,a)-Q\_{r\_{\theta}(r,r-r'),\pi\_{r'}}^{soft}(s,a)$. Therefore we have that $J\_r(\pi\_{\hat{r}})-J\_r(\pi_{r'})\leq \frac{Cr\_{\max}}{1-\gamma}\max\{||Q\_{r\_{\theta}(r,r-\hat{r}),\pi\_{\hat{r}}}^{soft}(s,a)-Q\_{r\_{\theta}(r,r-r'),\pi\_{\hat{r}}}^{soft}(s,a)||\_{\infty},||Q\_{r\_{\theta}(r,r-\hat{r}),\pi\_{r'}}^{soft}(s,a)-Q\_{r\_{\theta}(r,r-r'),\pi\_{r'}}^{soft}(s,a)||\_{\infty}\}$. We first bound $||Q\_{r\_{\theta}(r,r-\hat{r}),\pi\_{\hat{r}}}^{soft}(s,a)-Q\_{r\_{\theta}(r,r-r'),\pi\_{\hat{r}}}^{soft}(s,a)||\_{\infty}$. Since $r\_\theta$ is linear, we have that $|r\_{\theta}(r(s,a),r(s,a)-\hat{r}(s,a))-r\_{\theta}(r(s,a),r(s,a)-r'(s,a))|=|[r(s,a)+\theta(s,a)(r(s,a)-\hat{r}(s,a))]-[r(s,a)+\theta(s,a)(r(s,a)-r'(s,a))]|=|\theta(s,a)(\hat{r}(s,a)-r'(s,a))|$ $\leq ||\theta(s,a)||\_{\infty}||\hat{r}(s,a)-r'(s,a)||\_{\infty}=O(\epsilon)$. Note that $||\theta(s,a)||\_\infty$ is finite because the state-action space is finite. Therefore $||Q\_{r\_{\theta}(r,r-\hat{r}),\pi\_{\hat{r}}}^{soft}(s,a)-Q\_{r\_{\theta}(r,r-r'),\pi\_{\hat{r}}}^{soft}(s,a)||\_{\infty}\leq \frac{1}{1-\gamma}||r\_{\theta}(r(s,a),r(s,a)-\hat{r}(s,a))-r\_{\theta}(r(s,a),r(s,a)-r'(s,a))||\_\infty = O(\epsilon)$. Similarly, we can get that $||Q\_{r\_{\theta}(r,r-\hat{r}),\pi\_{r'}}^{soft}(s,a)-Q\_{r\_{\theta}(r,r-r'),\pi\_{r'}}^{soft}(s,a)||\_{\infty}\leq O(\epsilon)$. Therefore, $J\_r(\pi\_{\hat{r}})-J\_r(\pi\_{r'})\leq O(\epsilon)$. This results hold for the case that the shaping parameter $\theta$ is same for $\hat{r}$ and $r'$, we will explore the case where $\theta$ is different.

---

> > ### Author Response · Authors · 2024-11-20
> >
> > **For the errors of the Q value and behavior policy**, these two errors have the same impact, i.e., identifying a wrong set of misleading state-action pairs. In this case, it is hard to quantitatively analyze how the errors impact the final performance because the relation between the misleading set and the Q-value and behavior policy is not continuous and hard to measure. However, our method can mitigate the impact caused by these errors. In specific, these errors will cause false positive and false negative. The false negative means that some misleading state-action pairs are not identified. The false positive means that some identified state-action pairs are not actually misleading and it is possible that the learning policy needs to visit these false positives to achieve high cumulative reward. We can mitigate the false positive ones by setting the budget $b>0$. In this case, we impose a soft constraint on the misleading state-action pairs, so that we do not totally avoid the misleading state-action pairs. The false positive ones can still be visited for the policy to achieve high cumulative reward.
> >
> > **Question 1**: "The key idea in this paper of learning an implied reward function of a learned RL agent and also identifying the state-action pairs that cause the agent to be non-optimal to be used in learning a new policy seems novel and interesting. However, wouldn’t this require re-learning? Wouldn’t generating the explanations require either access to the environment for generating additional data or knowledge of the environment parameters such as transition function, maximum attainable reward, goal state etc.? Can the authors add a discussion on this topic detailing what extra data and computation are needed by their method?"
> > **Answer**: Generating explanations does not require extra knowledge of the environment parameters, but requires extra environment interactions. Compared to standard SAC, the explanation part requires 6\% additional environment interactions. Please refer to our answer to Weakness 2 for details.
> > Generating explanations also requires additional computation. We use the number of gradient computation of neural networks to measure the computation. We have three neural networks: reward network, Q-network, and policy network. The number of the gradient computation of reward network is 1 for each sampled episode, the number of the gradient computation of Q-network is the batch size, and the number of the gradient computation of policy network is the batch size. For ML-IRL, at each iteration, the number of reward gradient computation is 1 since we sample 1 episode, and the number of gradient computation of Q-network and policy network are both 10 since the batch size is 10. Since we run ML-IRL for 500 iterations, the gradient computation is $500\times(1+10+10)=10500$. To learn the Q-value, the batch size is 100 at each iteration. Note that the batch size is larger than that of ML-IRL because we have 100 sampled episodes to learn the Q value while ML-IRL samples one episode per iteration. We run 50 iterations, and thus the total number of gradient computation is $100\times 50=5000$. The total number of gradient computation for the explanation part us $10500+5000=15500$. For the standard SAC, at each iteration, the batch size is also 100 because SAC also sampled more episodes than ML-IRL at each iteration. At each iteration of SAC, we have Q update and policy update, and thus the number of gradient computation is 200. The total iteration number is 1000, therefore, the total number of gradient computation of SAC is 200000. In conclusion, the additional computation of the explanation part is 15500/200000=7.75% of the computation of SAC.

---

> > > ### Author Response · Authors · 2024-11-20
> > >
> > > **Question 2**: Can the authors also compare their approach with other RL algorithms from literature where the other algorithms have access to the same total quantity of data used by the authors’ approach? The current comparison in Fig 2, does have total environment interactions, can the authors confirm if these interactions also include the additional data mentioned in Point 1 above?
> > > **Answer**: As discussed in the answer to Weakness 2, the improvement part (solving problem (1)) uses same number of sampled episodes as the standard SAC. Figure 2 only includes the data for the improvement part. The explanation part indeed requires extra 6\% (i.e., 600) episodes. For a completely fair comparison, we rerun the baselines where we increase the iteration number of the baselines so that the baselines use the same amount of sampled episodes as our method. For example, we now run SAC for 1060 iterations so that SAC uses $1060\times 10=10600$ episodes, which is same as the total episodes used in our method (explanation part and improvement part combined). We provide the results below:
> > >
> > > |                  | SAC            | UTILITY       | RICE          | SIL           | LIR           |
> > > |------------------|----------------|---------------|---------------|---------------|---------------|
> > > | Delayed HalfCheetah | $391.04\pm34.63$ | $715.96\pm42.78$ | $455.63\pm39.85$ | $527.79\pm27.66$ | $553.01\pm42.14$ |
> > > | Delayed Hopper    | $198.35\pm28.02$ | $317.99\pm19.62$ | $233.12\pm15.53$ | $261.05\pm23.11$ | $246.03\pm24.11$ |
> > > | Delayed Walker2d  | $137.22\pm26.52$ | $242.63\pm14.11$ | $162.14\pm14.78$ | $181.41\pm29.21$ | $298.53\pm20.15$ |
> > > | Delayed Ant       | $71.52\pm14.67$  | $105.80\pm14.38$ | $76.59\pm12.15$  | $85.14\pm11.56$  | $82.01\pm11.38$  |
> > >
> > > From the results above, we can see that our method UTILITY still significantly outperforms the baselines when the baselines use the same amount of sampled episodes. In fact, the performance of the baselines does not change much because (1) the baselines already converge and running more iterations will not significantly improve the performance and (2) the extra data is only 6\% of the total data.
> > >
> > > **Question 3**: "It is known from the literature on inverse RL that learning a reward function from data is a problem that can have multiple solutions. Can the authors describe how they address the non-uniqueness problem and how do they choose a single reward function?"
> > > **Answer**: It is very insightful that you mention the non-uniqueness of the reward function, i.e., there are multiple feasible reward functions. In this paper, we use ML-IRL and ML-IRL can already find a single reward function using the heuristic of maximum likelihood. In specific, ML-IRL aims to learn a reward function whose corresponding policy makes the demonstration data most likely. In other words, among all the feasible reward functions, ML-IRL picks the reward function with maximum likelihood as the solution.
> > >
> > > **Question 4**: "Can the authors clarify if different plausible high-level explanations (learned reward functions) can lead to different low-level explanations (misleading states)? If so, how do the authors propose to select the best high-level explanation? What is the authors’ opinion on using feedback from the RL agent’s performance post using their approach as input to improve their explanation generation. i.e., can their approach benefit by making the process of explanation generation and policy improvement iterative?"
> > > **Answer**: Different learned reward $\hat{r}$ will not affect the low-level explanations if the learned policy $\hat{\pi}\_A$ remains the same. In specific, the low-level explanation finds misleading state-action pairs using the learned policy $\hat{\pi}\_A$, and the learned policy $\hat{\pi}\_A$ can be optimal w.r.t. multiple different $\hat{r}$. Note that $\hat{\pi}\_A$ is learned such that $\hat{\pi}\_A$ can generate similar trajectories with the demonstrations. Therefore, different IRL algorithms may learn different $\hat{r}$, however, their corresponding $\hat{\pi}\_A$ should be similar or same because $\hat{\pi}\_A$ needs to generate similar behaviors with the demonstrations. In this sense, different $\hat{r}$ will not affect the low-level explanations as they will lead to similar $\hat{\pi}\_A$.
> > > It is a very interesting idea that we iteratively run the loop of generating explanations and solving problem (1). This method can further improve the performance. In specific, after improvement, the final policy may still be sub-optimal so that we generate explanations for this improved policy again and solve the problem (1) again. This iterative method can further improve the performance, however, we should also note that this iterative method requires a lot of environment interactions and computation and the improvement effect will becomes smaller and smaller.

---

> > > > ### Author Response · Authors · 2024-11-20
> > > >
> > > > **Question 5**: Can the authors confirm if misleading states are states where the agent does not act optimally as per its own Q-function? Is this definition valid even if the Q-function learned by the RL agent is not accurate, which is often the case in several practical RL problems?
> > > > *Answer**: Your understanding is correct. If the learned Q-function is not accurate, the identified misleading state-action pairs may not be accurate. Please refer to our answer to Weakness 4 for mitigation method. Empirically, we conduct an ablation study in table 2, and the results show that only discouraging visiting the misleading state-action pairs can also improve performance. This means that the identified misleading state-action pairs are primarily correct. To better see whether the identified state-action pairs are accurate, we use the mitigation method introduced in our answer to Weakness 4. In specific, we choose different values of $b$ and see the corresponding improvement results. Our method (UTILITY) chooses $b=0$, and we also choose $b=5$ and $b=20$ for comparisons. The values in the table below are the cumulative reward improvement.
> > > >
> > > > |        | Delayed HalfCheetah | Delayed Hopper | Delayed Walker | Delayed Ant |
> > > > |--------|---------------------|----------------|----------------|-------------|
> > > > | $b=0$  | $38.70$             | $17.22$        | $40.75$        | $7.05$      |
> > > > | $b=5$  | $38.14$             | $21.63$        | $42.19$        | $7.76$      |
> > > > | $b=20$ | $12.66$             | $5.29$         | $3.29$         | $1.48$      |
> > > >
> > > > The results above show that when $b=5$, the cumulative reward improvement will become larger. This shows that we have false positive misleading state-action pairs and a small positive $b$ can reduce the negative impact of those false positive misleading state-action pairs. When $b=20$, the cumulative improvement becomes much smaller because at this time, many misleading state-action pairs that should be avoided are not avoided. The results above indicate that while our misleading state-action pairs have some false positives, the majority is accurate.
> > > >
> > > > **Question 6**: "The authors state in Line 222 that: "However, our case is different because we only need to learn one precise Q-function, which corresponds to the specific policy $\hat{\pi}\_A$”. But won’t the authors need this Q-function to be accurate for also state-action pairs that are not visited or rarely visited under this policy? This is because in defining the $l(s,a)$ metric, the authors use a max over all Q values of a given state s. How do the authors propose to learn these values accurately or adjust the definition of their proposed metric to ensure that this does not impact their approach?"
> > > > **Answer**: Thanks for mentioning that inaccurate $Q$ may affect our approach. We agree that it is not possible to learn a 100\% correct Q-function and thus the identified misleading state-action pairs may not be 100\% accurate. In our answer to weakness 2, we propose a mitigation method. In our answer to Question 5, we empirically show that our identified misleading state-action pairs are not 100\% accurate, but the majority is accurate.
> > > >
> > > > **Question 7**: "Can the authors clarify if the shaping reward function requires access to the ground-truth reward function? If so, how do the authors propose to obtain this?"
> > > > **Answer**: The shaping reward does not require ground truth reward function, but requires ground truth reward value $r(s,a)$ given a state-action $(s,a)$. This value can be obtained by interacting with the environment.
> > > >
> > > > **Question 8**: "Can the authors specify the loss used for learning the proposed shaping reward used in equation (1), i.e., can the authors provide a discussion on the gradients proposed in Lemma 2 and its relation to this shaping reward function?"
> > > >
> > > > **Answer**: The parameter of the shaping reward is $\theta$. Given the current shaping reward $r\_\theta$, the lower level in (1) computes the policy $\pi\_{\lambda^{\ast}(\theta);\theta}$ corresponding to $r\_\theta$. Therefore, the shaping reward parameter $\theta$ is a hyper-parameter of the policy $\pi\_{\lambda^{\ast}(\theta);\theta}$. Once the shaping reward parameter $\theta$ changes, the corresponding policy $\pi\_{\lambda^{\ast}(\theta);\theta}$ also changes and thus the cumulative reward $J\_r(\pi\_{\lambda^{\ast}(\theta);\theta})$ changes. We want to optimize $\theta$ such that $\pi\_{\lambda^{\ast}(\theta);\theta}$ can maximize $J\_r(\pi\_{\lambda^{\ast}(\theta);\theta})$. We use gradient ascent to optimize $\theta$. In specific, the gradient in Lemma 2 is the gradient of $J\_r(\pi\_{\lambda^{\ast}(\theta);\theta})$ w.r.t. $\theta$. Once we get this gradient, we can use gradient ascent to optimize $\theta$ to achieve maximum cumulative reward. In conclusion, the loss function for the shaping reward is $J\_r(\pi\_{\lambda^{\ast}(\theta);\theta})$. We aim to optimize $\theta$ such that $J\_r(\pi\_{\lambda^{\ast}(\theta);\theta})$ is maximized.

---

> > > > > ### Author Response · Authors · 2024-11-20
> > > > >
> > > > > **Question 9**: "When the authors claim uniqueness of solution in Theorems 1 and 2, it would be helpful if they can clarify whether only J value is unique or $\pi$ is also unique?"
> > > > > **Answer**: As mentioned in Theorem 1, the optimal policy $\pi\_{r\_\theta}$ is unique.
> > > > >
> > > > > **Question 10**: "Minor: There is a typo in "RICE+constaint" which occurs twice in the paper (Lines 513 and 514)."
> > > > > **Answer**: We believe that this is not a typo. In line 513, we introduce the baseline “RICE+constaint”. In line 514, we mention that this baseline “RICE+constaint” is expected to have low fidelity. That is why “RICE+constaint” occurs twice.
> > > > >
> > > > > [A1] Cen, Shicong, et al. "Fast global convergence of natural policy gradient methods with entropy regularization." Operations Research 70.4 (2022): 2563-2578.
> > > > >
> > > > > [A2] Livni, Roi, Shai Shalev-Shwartz, and Ohad Shamir. "On the computational efficiency of training neural networks." Advances in Neural Information Processing Systems 27 (2014).
> > > > >
> > > > > [A3] Xu, Tengyu, Zhe Wang, and Yingbin Liang. "Improving sample complexity bounds for (natural) actor-critic algorithms." Advances in Neural Information Processing Systems 33 (2020): 4358-4369.
> > > > >
> > > > > [A4] Zeng, Siliang, et al. "Maximum-likelihood inverse reinforcement learning with finite-time guarantees." Advances in Neural Information Processing Systems 35 (2022): 10122-10135.

---

> > > > > > ### Author Response · Authors · 2024-11-24
> > > > > >
> > > > > > Dear Reviewer,
> > > > > >
> > > > > > We hope that you've had a chance to read our responses and clarification. As the end of the discussion period is approaching, we would greatly appreciate it if you could confirm that our updates have addressed your concerns.
> > > > > >
> > > > > > Best,
> > > > > > Authors.

---

> > > > > > > ### Comment · Reviewer_euxG · 2024-11-25
> > > > > > >
> > > > > > > I thank the authors for their detailed response and the derivations. While I have not been able to completely verify the derivation, I could not find any error. The additional experiments conducted by the authors were also very helpful. After going through the response in detail, I have increased my score.

---

> > > > > > > > ### Author Response · Authors · 2024-11-25
> > > > > > > >
> > > > > > > > We sincerely appreciate your time and effort in reviewing our paper and thank you for recognizing the contributions of this work.

---

### Meta-Review · Area_Chair_p6aC · 2024-12-19

**Metareview:**

Motivated by explainable RL, this paper introduces a policy iteration-like algorithm that uses a bi-level optimization procedure to estimate which state-action pairs are most responsible for the policy not being optimal. The policy is then optimized to correct for the suboptimal actions. The paper demonstrates the effectiveness of the method with both theoretical and empirical results.


After a discussion with the reviewers, they all agree that the paper is above the threshold for acceptance, and I agree with their recommendation. I encourage the authors to consider the reviewer’s points when revising the paper, particularly regarding the presentation for a more general RL audience.

**Additional Comments On Reviewer Discussion:**

Much of the discussion with the authors was centered on asking clarifying questions about the method, its framing, and possible limitations. After the discussion, the reviewers better understood the paper and agreed it warrants acceptance. The lack of understanding from the reviews is indicative that the paper needs to be presented differently to maximize the reader's understanding.

---

### Decision · Program_Chairs · 2025-01-22

Accept (Poster)